# Fourier Learning Machines: Nonharmonic Fourier–Based Neural Networks for Scientific Machine Learning

**Mominul Rubel**\**Department of Engineering Management and Systems Engineering*
*Missouri University of Science and Technology, Rolla, MO*

**Adam Meyers** *Department of Industrial and Systems Engineering*
*University of Miami, Coral Gables, FL*

**Gabriel Nicolosi**\*†*Department of Engineering Management and Systems Engineering*
*Missouri University of Science and Technology, Rolla, MO*

**Reviewed on OpenReview:** *https://openreview.net/forum?id=LPKt5vd7yz*

## Abstract

We introduce the Fourier Learning Machine (FLM), a neural network (NN) architecture designed to represent a multidimensional nonharmonic Fourier series. The FLM uses a simple feedforward structure with cosine activation functions to learn the frequencies, amplitudes, and phase shifts of the series as trainable parameters. This design allows the model to create a problem–specific spectral basis adaptable to both periodic and nonperiodic functions. Unlike previous Fourier–inspired NN models, the FLM is the first architecture able to represent a multidimensional Fourier series with a complete set of basis functions in separable form, doing so by using a standard Multilayer Perceptron–like architecture. A one–to–one correspondence between the Fourier coefficients and amplitudes and phase-shifts is demonstrated, allowing for the translation between a full, separable basis form and the cosine phase–shifted one. Additionally, we evaluate the performance of FLMs on several scientific computing problems, including benchmark Partial Differential Equations (PDEs) and a family of Optimal Control Problems (OCPs). Computational experiments show that the performance of FLMs is comparable, and often superior, to that of established architectures like SIREN and vanilla feedforward NNs.

## 1 Introduction

Approximating functions as a composition or weighted sum of simpler basis elements is a fundamental approach in both classical analysis and modern machine learning. Two typical examples of this idea are the Fourier series and artificial neural networks (NNs). The Fourier series decomposes a periodic function (or, for nonperiodic functions, their periodic extension) into a linear combination of a fixed and predetermined set of sinusoidal (*sines* and *cosines*) basis functions and gives an interpretable spectral representation of it. While nonperiodic functions can also be handled by defining a periodic extension over a finite interval, this often introduces discontinuities at the boundaries and leads to oscillatory behavior, known as the Gibbs phenomenon. Moreover, these Fourier bases are fixed and not adaptable to the specific features of the data. NNs, on the other hand, are computational models that approximate any function through compositions of parametrized nonlinear transformations of linear combinations of the data with (trainable) weights and biases. These parameters composing a NN are learned and adapted to the data during training (the process of optimizing the network parameters) using nonlinear optimization techniques. However, these weights and biases are numbers with little interpretation, and the network itself works as a black–box. Despite these

---

\*These authors contributed equally.
†Corresponding author. Email: `gabrielnicolosi@mst.edu`

differences, both approaches share the basic underlying principle of building complex functions from simpler components.

Motivated by Joseph Fourier's early $19^{th}$ century idea that any arbitrary function, regardless of its complexity or continuity, could be represented as a sum of *sines* and *cosines*, and by the successful application of NNs as universal function approximators since the late $20^{th}$ century, we propose in this paper a new Fourier–inspired NN model, and we shall name it the Fourier Learning Machine (FLM). Unlike typical black–box NN models, the FLM's internal structure is built to directly represent an $m-$dimensional nonharmonic Fourier series. It achieves this by relying solely on the foundational components of a classical multilayer perceptron (MLP): namely, affine transformations followed by nonlinear activation functions (in the context of this paper, these being pure *cosines*). This design choice eschews more complicated architectural features, such as multiplication nodes, recurrent relations or any other customized components, previously found in predecessor models. With hardwired integer weights in its input–to–hidden layer, the FLM represents an $m-$dimensional, full–basis classical (finite) Fourier series as a special case. In order to tackle the approximation of nonperiodic functions, we relax the integrality constraints of the frequency components, thus configuring the frequencies as learnable parameters of the FLM. This allows for the search for the sinusoidal basis functions that best represent the target map. While maintaining interpretability, FLMs build a function–specific spectral representation for a wide variety of functions, including nonperiodic ones. As such, our proposed model combines the interpretability of Fourier series with the data–driven nature of NNs (and, in particular, the simplicity of the MLP architecture), effectively representing a nonharmonic Fourier series (Young, 1981; Duffin & Schaeffer, 1952) in which the basis functions are learned and adapted to the target function it is meant to approximate.

The challenge of representing Fourier series with traditional MLP−like NNs can be understood by first considering a one−dimensional Fourier series on the interval $[-\pi, \pi]$ (without loss of generality) in the *cosine* phase–shifted form, *i.e.*,

$$\widehat{f}(t) = a_0 + \sum_{n=1}^{N} A_n \cos(nt - \phi_n),$$

where $A_n$ and $\phi_n$ denote the amplitude and the phase shift for the $n^{th}$ frequency component, respectively. For such a form of the Fourier series, it is straightforward to see that the approximation $\widehat{f}$ can be naturally represented by a single–layer NN with *cosine* activation functions. Nonetheless, generalizing this idea to higher dimensions requires extra care since, as we demonstrate in this paper, the NN structure then either departs too far from the MLP–like architecture or fails to represent an actual, truncated nonharmonic Fourier series. As a result, a certain degree of architectural ingenuity is required so that the model retains both a simple architecture and the integrity of the nonharmonic Fourier representation in higher dimensions.

In this paper, we show that the proposed FLM does not only combine an interpretable structure with adaptable bases, but also achieves approximation accuracy comparable to, and in some cases exceeding, that of established NN architectures. Although we focus on using FLMs to solve Partial Differential Equations (PDEs) and an Optimal Control Problem (OCP), this model can be applied to a wide range of problems, such as signal and image reconstruction, spectral analysis, and surrogate modeling for high–dimensional systems, where both interpretability and adaptability are desired. The main contributions of this paper are:

1. We provide a comprehensive review of all previous attempts to construct such a function approximator, typically referred in the literature as "Fourier Neural Networks", departing from foundational works (Gallant & White, 1988) and covering up to recent architectures with trigonometric activation functions (Sitzmann et al., 2020). Within this context, we demonstrate that the FLM is not only architecturally simpler than previous models but also computationally competitive, often outperforming established, traditional NN models.

2. By defining the $m$–Lexi Sign Matrix, we show how the FLM, a feedforward NN, can simultaneously represent a full basis finite Fourier series in $m$–dimensions in both separable and *cosine* phase–shifted forms, with a one–to–one correspondence between Fourier coefficients, frequencies and phase–shift and the NN weights and biases. To the best of the author's knowledge, this flexibility to represent a full–basis classical Fourier series in a MLP–like architecture, is unique in the literature. In the

nonharmonic case, it is shown that such a representation feature is directly translated to a more computationally efficient parameter optimization when compared to other traditional and Fourier–based NN models.

3. Through a series of carefully designed computational experiments, we empirically demonstrate the FLM's approximation capabilities for solving PDEs and optimization problems. In particular, we apply the FLM to the numerical solution of a multidimensional OCP arising from the evolutionary dynamics of a Rock–Paper–Scissors (odd–circulant) game, demonstrating its capability to solve complex, nonlinear problems.

The remainder of the paper is organized as follows: Section 2 reviews related literature and prior work on Fourier–related NNs. Section 3 introduces a few mathematical preliminaries and introduces the inherent challenges of representing a (nonharmonic) Fourier series with a MLP–like NN architecture. Section 4 constructs, formalizes and presents the proposed FLM architecture in detail. Section 5 reports computational experiments and compares the FLM's performance with established NN architectures. Finally, Section 6 concludes the paper and discusses potential future research directions.

## 2   Related Literature

The idea of a Fourier series–inspired NN was first proposed in the late 1980s, just before NNs were shown to possess universal function approximation properties (Hornik et al., 1989). This pioneer architecture, called "Fourier Network" (FN) was proposed by Gallant & White (1988). In their work, they carefully constructed a Fourier series representation into a NN by defining a "*cosine*–squasher" activation function, a domain–constrained, monotonic, *cosine* function. By fixing the input–to–hidden layer weights, this NN was demonstrated to mimic a finite multidimensional Fourier series. This "*cosine*–squasher" is a monotonic and piecewise differentiable function, $F : \mathbb{R} \to [0, 1]$ , defined as

$$F(net) = \begin{cases} 0 & \text{if } -\infty < net \leq -\frac{\pi}{2} \\ \left[\cos\left(net + \frac{3\pi}{2}\right) + 1\right]/2 & \text{if } -\frac{\pi}{2} \leq net \leq \frac{\pi}{2} \\ 1 & \text{if } \frac{\pi}{2} \leq net < \infty. \end{cases} \tag{1}$$

While $F(\cdot)$ is not strictly periodic, the FN constructs the Fourier *cosine* bases by constraining and summing multiple hidden units within a group of neurons called "clumps". This activation function is designed to fit the "squashing" paradigm of early NNs (Rumelhart et al., 1986) while allowing for the construction of a NN replicating the approximation properties of Fourier series. A single "clump" is depicted in Fig. 1. As such, the output of each of these "clumps" forms one of the *cosine* or *sine* basis (by adding a phase–shift) present in a Fourier series. To create a single *cosine* basis, the weighted sum of the inputs, *i.e.*, the pre–activation $net_i = \boldsymbol{w}_i^\top \boldsymbol{x} + b_i$ is preprocessed and activated using the *cosine*–squasher function. By constraining $\xi_{im} = \delta_{im} = 2\theta_i$, where $i = 1, 2, \ldots, K$ and $m = -M_i, \ldots, 0, \ldots, M_i$, the output of the FN is given by,

$$o = \sum_{i=1}^{K} \underbrace{\sum_{m=-M_i}^{M_i} \left[2\theta_i \, F(-net_i + \tfrac{\pi}{2} - m2\pi) + 2\theta_i \, F(net_i - \tfrac{3\pi}{2} + m2\pi)\right]}_{o_i}$$

$$= \sum_{i=1}^{K} \theta_i \cos(net_i) - \sum_{i=1}^{K} \left[\theta_i - \theta_i \, 2(2M_i + 1)\right],$$

which is a classical Fourier series in *cosine* phase-shifted form when an extra term $\delta_0 = \theta_0 + \sum_{i=1}^{K} \left[\theta_i - \theta_i \, 2(2M_i + 1)\right]$ is added to the output. The role of $M_i$ is to bound the range within which the output $net_i$ can vary. As the input $\boldsymbol{x}$ varies in the (perturbed) hypercube $[\epsilon, 2\pi - \epsilon]^m$, $net_i$ varies in the interval $[-2\pi M_i, 2\pi(M_i + 1)]$. Therefore, $M_i$ is inherently dependent on the "hardwired" parameters $\mathbf{w}_i$ and $b_i$ in the first layer. Furthermore, $M_i$ increases with $i$, that is, the inclusion of higher frequency modes

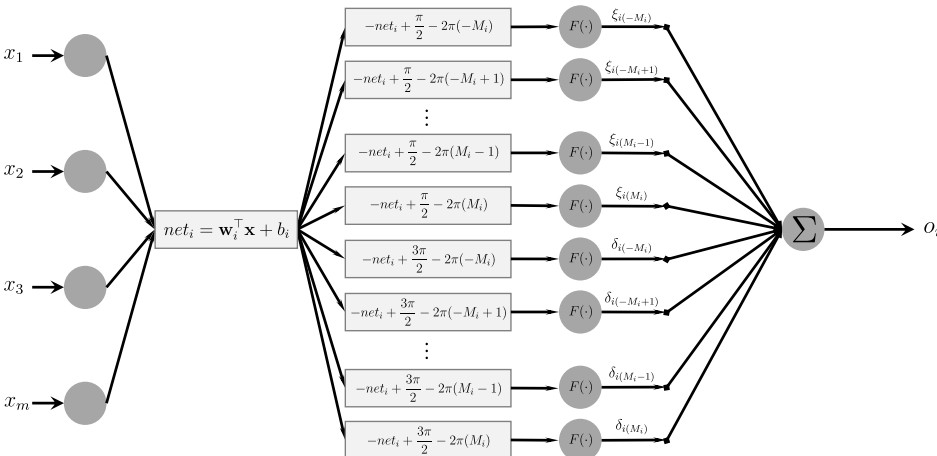

Figure 1: A "clump" of neurons that outputs the $i^{th}$ single *cosine* basis of the Fourier Network appearing in Gallant & White (1988).

requires more neurons within a "clump". For smaller network sizes, while the authors claim that by undoing the "hardwiring" (in $\mathbf{w}_i$ and $b_i$) and removing the constraint in $\xi_{im}$ and $\delta_{im}$ the FN could potentially perform better, doing so nullifies its exact correspondence with a finite Fourier series: making $\mathbf{w}_i$ and $b_i$ trainable implies that one could not determine a specific value for each $M_i$ before training the FN. Thus, it is unclear how to pick $M_i$ under the aforementioned construction. Lastly, by construction, removing the constraint $\xi_{im} = \delta_{im} = 2\theta_i$ would hinder the $i^{th}$ "clump" to create a *cosine* basis. Therefore, such an attempt would not lead the network to represent a nonharmonic Fourier series.

It is worth noticing that the requirement of a "monotonic" activation function was later dropped based on the universal approximation (and convergence) result in Leshno et al. (1993), which established that non-polynomial and bounded functions (*e.g.*, *cosine*) can be also serve as activation functions in NNs. Therefore, such an intricate design proposed in Gallant & White (1988) can be partially justified by the lack of such a result when their architecture was developed.

In the context of systems control and identification, Zhu & Paul (1995) proposed the first NN capable of mimicking a finite Fourier series by using pure *sines* and *cosines*, naming it the "Fourier Series Neural Network" (FSNN). Despite their NN capability to represent a full-basis Fourier series, given a fixed set of nontrainable "frequency weights", the architecture chosen relies on multiplicative nodes to compose each of the basis functions. The training process in this case is rather simple, given the linearity (with respect to the Fourier coefficients) of the output layer, thus yielding a linear optimization problem. More recently, Benrabah et al. (2021) introduced an adaptive controller where a FSNN is used to compute and adjust the gains of a PID controller online.

With an emphasis on similarities of its model with the brain neurophysiology, Silvescu (1999) proposed an architecture relying on multiplicative nodes that attempted to reproduce a finite Fourier series in *cosine* phase–shifted form (optionally "squeezed" through a sigmoid function). This network, named "Fourier Neural Network" (the first appearance of such a term in the literature) assumed the form

$$f(x_1, \ldots, x_m) = \sigma \left( \sum_i c_i \prod_{j=1}^m \cos(w_{ij} x_j + \varphi_{ij}) \right). \tag{2}$$

However, it is implicit in Silvescu (1999) that the trainable parameters $w$ and $\varphi$ are not fixed or constrained to be either natural numbers (including zero) or odd–multiple of $\frac{\pi}{2}$, respectively. Therefore, despite its similarities with a Fourier series, $f(\cdot)$ fails to represent either a finite, classical Fourier series or a (typical) nonharmonic one. In its current form, a classical Fourier series could be attained from Equation 2 by fixing $w$'s to a predetermined set of integers and making some $\varphi \in \{(2k+1)\frac{\pi}{2} \mid k \in \mathbb{Z}\}$. Conversely, a nonharmonic series can only be obtained in the form of a sum of *cosine*–only basis functions, which would configure a very

specific (and arguably weak) form of nonharmonic Fourier series, allowing one to approximate even functions only. For a depiction of this FN model, the reader is pointed to Uteuliyeva et al. (2020).

In Yang et al. (2003), a single–input, multiple–output NN is proposed, wherein each output corresponds to that of an individual finite Fourier series. In this architecture, all the series share the same basis functions but each have their own set of (trainable) Fourier coefficients. Besides being of limited applicability (once it is design to approximate single variable functions only), Yang et al. (2003) also fails to make any reference to previously proposed, Fourier–inspired NN already in existence in the literature at that time.

Halawa (2008) uses a Discrete Fourier Transform to find the hidden–to–output weights for the FSNN model from Zhu & Paul (1995). This neural network formulation, however, is unnecessary for the task, as the calculation provides little advantage over a direct computation performed outside of an NN framework.

In Liu (2013), another type of FNN is proposed, this version utilizing a vector form of the multidimensional, finite Fourier series. This NN assumes the form

$$f(\boldsymbol{x}) = \sum_{i=1}^{N} \left[ a_i \, \sin \left( \boldsymbol{c}_i^\top \boldsymbol{x} \right) + b_i \, \cos \left( \boldsymbol{d}_i^\top \boldsymbol{x} \right) \right], \tag{3}$$

where, the (real–valued) numbers $\boldsymbol{c}_i$, $\boldsymbol{d}_i$, $a_i$ and $b_i$ are the learnable parameters of the corresponding NN. Unless the frequency components are constrained to be equal position-wise, *i.e.*, $\boldsymbol{c}_i = \boldsymbol{d}_i$, this representation is a trigonometric expansion, but it is not strictly a nonharmonic Fourier series. This is because the use of different frequency vectors inside the *sine* and *cosine* prevents Equation 3 to be written as its equivalent exponential or separable basis forms. Therefore, in its original form presented, Equation 3 represents neither a classical Fourier series nor a nonharmonic one.

A comparative study on the performance of the FN model of Gallant & White (1988), the FNN of Silvescu (1999) and the FNN of Liu (2013) is presented in Uteuliyeva et al. (2020), which concluded that the top performing model among these three is the early FN model of Gallant & White (1988). However, this FN model of Gallant & White (1988) was shown to underperform a vanilla feedforward NN with sigmoid activation function. The reason behind this result, as hypothesized by the authors, is that, outside the interval $[-\frac{\pi}{2}, \frac{\pi}{2}]$, the "*cosine*–squasher" as written in Equation 1 is constant, whereas the sigmoid employed in the vanilla feedforward NN is asymptotically constant outside of this same interval.

A comprehensive and critical investigation on the use of *sine* functions as activation functions in NNs has appeared in Parascandolo et al. (2017). In this work, the authors claim that the wave–like behavior introduced in the loss function by the presence of *sines* introduces many local minima, making such models harder to train. However, the output of the NNs chosen for the experiments shown in Parascandolo et al. (2017) are not meant to represent a Fourier series in any form. Furthermore, the performance of such activation function is not studied for a simple feedforward NN structure, but in the context of encoder–decoder architectures based on recurrent and long short–term memory NNs. Related to this idea, Ozmermer (2020) designs a "Sinusoidal Neural Network (SNN)", with a newly defined *sine*–based node operations inspired by information propagation in the brain. Even though empirical results presented are promising, no theoretical background is provided to guarantee the convergence of this NN.

A (deep) NN with *sine* activation functions, called SIREN (Sinusoidal Representation Network) is introduced in Sitzmann et al. (2020), tailored for applications in digital media processing. The authors customize the first layer of this NN in order to extract high–frequency components of the target data (image, video or audio), doing so by multiplying the weighted sum of the input by a relatively large constant term $\omega_0$ (keeping the bias untouched), while all the other activation functions assume the form $y \longmapsto \sin y$. We note that a SIREN–1, that is, a SIREN with one single layer, for $\omega_0 = 1$ can be shown, under certain trigonometric identities, to assume the mathematical form of

$$f(\mathbf{x}) = \sum_{i=1}^{N} \left[ a_i \, \sin \left( \boldsymbol{w}_i^\top \boldsymbol{x} \right) + b_i \, \cos \left( \boldsymbol{w}_i^\top \boldsymbol{x} \right) \right]. \tag{4}$$

Furthermore, we note that Equation 4 is identical to Equation 3 for when $\mathbf{c}_i = \mathbf{d}_i$. Thus, similar to the case of Liu (2013), a SIREN–1 represents a Fourier series by fixing the components of $\mathbf{w}$ as integers (noted only

later on by Benbarka et al. (2022)), and a nonharmonic series otherwise. It is worth noting that in Sitzmann et al. (2020) there is no mention to any potential relations between a SIREN NN and any form of a Fourier series. However, as it is also the case of the FNN of Liu (2013), this Fourier series cannot be represented in a full–basis, separable form.

In the context of scientific machine learning, Ngom & Marin (2021) introduced a one–dimensional FNN as function approximators. Their FNN model assumes the form of a nonharmonic (phase–shifted *cosine*) Fourier series (with trainable frequency components). Nonetheless, all the numerical experiments reported considered only the approximation of periodic functions that can be fully decomposed as a finite Fourier series, thus not fully exploring the potential of their model in the approximation of nonperiodic functions.

In order to avoid confusion by the use of similar terms and concepts, other scientific contributions that somehow involve the use of Fourier analysis (but do not attempt to construct a NN that replicates a Fourier series) must be mentioned in order to avoid confusion. Recently, the Fourier Neural Operator (FNO) gained tremendous attention in the scientific machine learning community (Li et al., 2020). The FNO primary goal is to approximate a operator, that is, a mapping between function spaces. Thus, it is suitable to learn operators in the context of approximating the solution of entire parametric families of PDEs. In this regard, the "Fourier" comes from the fact that the integral kernel is parameterized in Fourier space. The FLM model presented in this paper, instead, aims to approximate a mapping from a higher–dimension Euclidean space to a real number. In Tancik et al. (2020), preprocessing input vectors with a Fourier feature mapping allows this NN to overcome its inherent spectral bias, *i.e.*, the tendency to learn low–frequency components and therefore facilitating learning of high–frequency functions. Lastly, by automatically determining the optimal sampling frequency and number of neurons, Fourier–based extreme learning machines are applied in Ma et al. (2022) for the solution of PDEs.

Some of the models described above move toward learnable spectral bases, while others are constrained by the integrality of their frequency components. Nonetheless, in either case, they often lack a systematic architecture allowing them to represent a complete, separable, and interpretable Fourier series of the target function (either classical or nonharmonic). Even though SIREN–1 and the FNN in Ngom & Marin (2021) fulfill the aforementioned criteria for a single input, these properties don't hold true when considering the general, multidimensional case. As we will show later in this paper, FLMs fill this gap, being the first Fourier–replicating NN that is capable of representing both finite and nonharmonic Fourier series in a full, separable basis form in $m$ dimensions by using only MLP–like, feedforward NN operations.

Lastly, even though explicitly outside of the NN domain (at least in the sense in which NNs are used in this work, *i.e.*, for deterministic function approximation), it is worth mentioning that FLMs share a fundamental similarity with Sparse Spectral Gaussian Processes (SSGP), introduced in Lázaro-Gredilla et al. (2010). This relationship draws from established results bridging NNs and Gaussian Processes (GPs) (Neal (1996); Murphy (2023)). Both FLMs and SSGP rely on learning frequencies of their underlying trigonometric basis functions. However, FLMs rely on the usual gradient descent in the space of frequencies (and amplitudes) to find the set of frequencies that "best" describe the target function. As it will be shown in Section 5, this is not necessarily a data-driven approach, but instead based on the customized loss function that enforces the desired system dynamics. On the other hand, in SSGP, the frequencies are treated as hyperparameters and optimized by maximizing the marginal likelihood (thus data-driven) to approximate the spectral density of the covariance kernel. Therefore, SSGP falls into the framework of (efficient) probabilistic regression for large-scale, discrete data, not undermining the FLM's spectral adaptability, but rather reinforcing its importance and applicability.

Next, we present a few mathematical preliminaries before introducing the proposed FLM architecture. Unless the reader is well–versed in Fourier analysis, reading the next section is strictly necessary in order to understand the architecture introduced later in Section 4.

## 3  Mathematical Preliminaries

In this section, we establish the mathematical foundation for our approach, using the classical Fourier series as a departing point. Our goal is to generalize this representation by treating the frequencies as arbitrary

real–valued parameters, instead of fixed integer multiples of the fundamental frequency. In doing so, we transform the classical Fourier series into a nonharmonic one.

Let $f : [-\pi, \pi]^m \to \mathbb{R}$ be a square–integrable function and $\widehat{f}_H$ be the Fourier series expansion of its $2\pi$–periodic extension in each variable. We consider the domain $[-\pi, \pi]^m$ for simplicity, however, this can be generalized to any domain $[-L_1, L_1] \times [-L_2, L_2] \times \cdots \times [-L_m, L_m]$ by scaling the input variables accordingly. The Fourier series of $f$ in complex exponential form is given by

$$\widehat{f}_H(\boldsymbol{x}) = \sum_{\boldsymbol{n} \in \mathbb{Z}^m} c_{\boldsymbol{n}}\, e^{i\boldsymbol{n} \cdot \boldsymbol{x}}, \tag{5}$$

where $\boldsymbol{x} = (x_1, x_2, \ldots, x_m) \in [-\pi, \pi]^m$ , $\boldsymbol{n} = (n_1, n_2, \ldots, n_m) \in \mathbb{Z}^m$ and $\boldsymbol{n} \cdot \boldsymbol{x}$ represents their dot product. The analytical expression for calculating the Fourier coefficients is

$$c_{\boldsymbol{n}} = \frac{1}{(2\pi)^m} \int_{[-\pi,\pi]^m} f(\boldsymbol{x})\, e^{-i\boldsymbol{n} \cdot \boldsymbol{x}}\, d\boldsymbol{x},$$

although in this paper we are interested in approximating them numerically. Alternatively, the same series $\widehat{f}_H$ can be expressed using real separable basis functions (products of the fundamental *sine* and *cosine* functions) as

$$
\begin{aligned}
\widehat{f}_H(\boldsymbol{x}) = \sum_{\boldsymbol{n} \in \mathbb{N}_0^m} \Big[ & a_1^{(\boldsymbol{n})}\, \cos(n_1 x_1) \cos(n_2 x_2) \ldots \cos(n_m x_m) \\
& + a_2^{(\boldsymbol{n})}\, \cos(n_1 x_1) \cos(n_2 x_2) \ldots \sin(n_m x_m) + \ldots \\
& + a_{2^m}^{(\boldsymbol{n})}\, \sin(n_1 x_1) \sin(n_2 x_2) \ldots \sin(n_m x_m) \Big],
\end{aligned}
\tag{6}
$$

for input $\boldsymbol{x} = (x_1, x_2, \ldots, x_m) \in [-\pi, \pi]^m$ and frequency components $\boldsymbol{n} = (n_1, n_2, \ldots, n_m) \in \mathbb{N}_0 = \{0, 1, 2, \ldots\}$. This trigonometric separable form has the advantage that the set of basis functions, associated with $\boldsymbol{n}$, simultaneously accounts for all possible $2^m$ symmetric frequency components. Here, symmetric components refer to the sign–flipped versions of a frequency vector, *i.e.*, $(\pm n_1, \pm n_2, \ldots, \pm n_m)$, which are always grouped together in the spectrum of a real–valued function. As a result, redundant terms that would otherwise appear in the complex exponential form in Equation 5 are implicitly bundled together, allowing real–valued functions to be expressed more efficiently while still spanning the complete set of available basis functions.

The classical Fourier series Equation 5 and Equation 6, which are defined by a rigid, evenly–spaced grid of frequencies, can be generalized into the nonharmonic Fourier series (Duffin & Schaeffer, 1952; Young, 1981). This concept provides a mathematical bridge to the Fourier Transform (FT), which uses a continuous spectrum of all possible frequencies. The nonharmonic series occupies a middle ground: like a classical series, it is a discrete sum of sinusoids, but it inherits the flexibility of the FT by allowing its frequencies to be arbitrary real numbers. This transformation is formally achieved by replacing the fixed set of integer frequency vectors, $\mathbb{N}_0^m$, with a general finite (for practical implementation) set of real–valued vectors, $\mathcal{N} = \{\boldsymbol{n}^1, \boldsymbol{n}^2, \ldots, \boldsymbol{n}^N\}$. This flexibility allows for the construction of an adaptive, task-specific Fourier basis capable of efficiently capturing the dominant spectral features of both periodic and nonperiodic functions. From this point forward, unless specified otherwise, we will use the term "Fourier series" to refer to this more general nonharmonic form, where the frequencies are from the set $\mathcal{N}$. The classical harmonic series will be considered a special case for when $\mathcal{N} \subseteq \mathbb{N}_0^m$.

With this change of frequency set just described, the classical series defined in Equation 5 and Equation 6 are generalized to their nonharmonic forms. The complex exponential form Equation 5 is generalized using a frequency set $\mathcal{N} \subseteq \mathbb{R}^m$ as

$$\widehat{f}_{NH}(\boldsymbol{x}) = \sum_{\boldsymbol{n} \in \mathcal{N}} c_{\boldsymbol{n}} e^{i\boldsymbol{n} \cdot \boldsymbol{x}}, \tag{7}$$

and the real trigonometric separable form Equation 6 is generalized using a frequency set of nonnegative real vectors, $\mathcal{N} \subseteq \mathbb{R}_{\geq 0}^m$ as

$$
\begin{aligned}
\widehat{f}_{NH}(\boldsymbol{x}) = \sum_{\boldsymbol{n} \in \mathcal{N}} \Big[ & a_1^{(\boldsymbol{n})} \cos(n_1 x_1) \cos(n_2 x_2) \ldots \cos(n_m x_m) \\
& + a_2^{(\boldsymbol{n})} \cos(n_1 x_1) \cos(n_2 x_2) \ldots \sin(n_m x_m) + \ldots \\
& + a_{2^m}^{(\boldsymbol{n})} \sin(n_1 x_1) \sin(n_2 x_2) \ldots \sin(n_m x_m) \Big].
\end{aligned}
\tag{8}
$$

To begin with, we set the frequencies to nonnegative vectors ($\mathcal{N} \subseteq \mathbb{R}_{\geq 0}^m$), for two reasons: (i) in practice, this constraint reduces the search space for the optimal frequencies from $2^m$ possible orthants to a single one, which leads to a more efficient and stable search process. This is justified since each frequency vector comes implicitly with its symmetrical equivalent in all the other $2^m - 1$ remaining orthants, and; (ii) in theory, this is justified because any sign flip in the components of a frequency vector only affects the sign of the coefficients, not the basis functions themselves, thus keeping the representation intact.

While both Equation 7 and Equation 8 represent standard forms of the Fourier series, they are not readily compatible with NN architectures: in the exponential form Equation 7, the basis functions are complex exponentials, which require complex–valued weights, activations, and gradient updates. Since traditional NNs are designed for real–valued computation, this would necessitate specialized handling or an explicit separation into real and imaginary parts, increasing implementation complexity. Additionally, the trigonometric form Equation 8 expresses each multidimensional basis as a product of one–dimensional *sine* and *cosine* factors. Thus, these bases become difficult to be represented by traditional NNs, which combine inputs additively through weighted sums before applying nonlinear activations.

To resolve the representation issues above, we opt for the *cosine* phase–shifted form, where, by phase shift we mean the translation of a *cosine* wave along its direction of oscillation. By converting the Fourier basis into a sum of *cosines* with phase shifts, we eliminate the need for complex numbers in Equation 7 and avoid the multiplicative interactions present in the trigonometric form of Equation 8. This makes the resulting representation more compatible with the additive structure of traditional, feedforward NNs. Next, we rewrite the standard separable trigonometric Fourier series in a *cosine* phase–shifted form, beginning with the two–dimensional case before generalizing to $m$ dimensions. We also show the relationship between the Fourier coefficients in the separable trigonometric form and the amplitudes and phase shifts in the *cosine* phase–shifted form.

### 3.1 Cosine Phase–Shifted Basis for Two–dimensional Input

Consider the two−dimensional Fourier series

$$
\begin{aligned}
\widehat{f}_{NH}(x_1, x_2) = \sum_{\boldsymbol{n} \in \mathcal{N}} \Big[ & a_1^{(\boldsymbol{n})} \cos(n_1 x_1) \cos(n_2 x_2) + a_2^{(\boldsymbol{n})} \cos(n_1 x_1) \sin(n_2 x_2) \\
& + a_3^{(\boldsymbol{n})} \sin(n_1 x_1) \cos(n_2 x_2) + a_4^{(\boldsymbol{n})} \sin(n_1 x_1) \sin(n_2 x_2) \Big],
\end{aligned}
\tag{9}
$$

where $\boldsymbol{n} = (n_1, n_2)$. Alternatively, Equation 9 can be expressed in the *cosine* phase-shift form

$$
\begin{aligned}
\widehat{f}_{NH}(x_1, x_2) &= \sum_{\boldsymbol{n} \in \mathcal{N}} \left[ A_1^{(\boldsymbol{n})} \cos\left(n_1 x_1 + n_2 x_2 - \phi_1^{(\boldsymbol{n})}\right) + A_2^{(\boldsymbol{n})} \cos\left(n_1 x_1 - n_2 x_2 - \phi_2^{(\boldsymbol{n})}\right) \right] \\
&= \sum_{\boldsymbol{n} \in \mathcal{N}} H_{\boldsymbol{n}},
\end{aligned}
\tag{10}
$$

where $H_{\boldsymbol{n}} = A_1^{(\boldsymbol{n})} \cos\left(n_1 x_1 + n_2 x_2 - \phi_1^{(\boldsymbol{n})}\right) + A_2^{(\boldsymbol{n})} \cos\left(n_1 x_1 - n_2 x_2 - \phi_2^{(\boldsymbol{n})}\right)$ can be viewed as an implicit linear combination of the four separable Fourier basis functions associated with $\boldsymbol{n}$. $A_1^{(\boldsymbol{n})}$ and $A_2^{(\boldsymbol{n})}$ are the amplitudes of the phase–shifted *cosines*, and $\phi_1$, $\phi_2$ are the respective phase shifts. The equivalence of Equation 9 and Equation 10 is shown in Appendix A with the detailed derivation of the relation that expresses the Fourier coefficients $a_1^{(\boldsymbol{n})}$, $a_2^{(\boldsymbol{n})}$, $a_3^{(\boldsymbol{n})}$ and $a_4^{(\boldsymbol{n})}$ in terms of the amplitudes and phase shifts as

$$a_1^{(\boldsymbol{n})} = A_1^{(\boldsymbol{n})} \cos \phi_1^{(\boldsymbol{n})} + A_2^{(\boldsymbol{n})} \cos \phi_2^{(\boldsymbol{n})}, \quad a_2^{(\boldsymbol{n})} = A_1^{(\boldsymbol{n})} \sin \phi_1^{(\boldsymbol{n})} - A_2^{(\boldsymbol{n})} \sin \phi_2^{(\boldsymbol{n})},$$
$$a_3^{(\boldsymbol{n})} = A_1^{(\boldsymbol{n})} \sin \phi_1^{(\boldsymbol{n})} + A_2^{(\boldsymbol{n})} \sin \phi_2^{(\boldsymbol{n})}, \quad a_4^{(\boldsymbol{n})} = -A_1^{(\boldsymbol{n})} \cos \phi_1^{(\boldsymbol{n})} + A_2^{(\boldsymbol{n})} \cos \phi_2^{(\boldsymbol{n})}. \tag{11}$$

Notice how the linear combination of two phase–shifted *cosine* bases in Equation 10 is implicitly generating the linear combination of the four separable trigonometric bases in Equation 9, both associated with same frequency components. The same frequency pair $(n_1, n_2)$ occurs inside both *cosine* bases with different amplitudes and phase–shifts, and with the sign of the second element flipped. To extend this idea of expressing separable trigonometric bases with phase–shifted *cosines* to $m$ dimensions, we define the following matrix.

**Definition 1 ($m$–Lexi Sign Matrix)** *Let $m \geq 2$ be a positive integer and $l = 2^{m-1}$. The $m$–Lexi Sign Matrix $S^{(m)}$ is the $l \times m$ matrix whose rows are denoted by*

$$\boldsymbol{e}^{(i)} = \left(e_1^{(i)}, e_2^{(i)}, \dots, e_m^{(i)}\right), \quad i = 1, 2, \dots, l,$$

*with $e_1^{(i)} = 1$ and $\left(e_2^{(i)}, e_3^{(i)}, \dots, e_m^{(i)}\right) \in \{1, -1\}^{m-1}$, i.e., the $(m-1)$–fold Cartesian product of the set $\{1, -1\}$. The rows $\boldsymbol{e}^{(i)}$ are arranged in lexicographic order with $1$ taken to have higher priority than $-1$. Explicitly,*

$$S^{(m)} = \begin{bmatrix} \boldsymbol{e}^{(1)} \\ \boldsymbol{e}^{(2)} \\ \boldsymbol{e}^{(3)} \\ \boldsymbol{e}^{(4)} \\ \vdots \\ \boldsymbol{e}^{(l-1)} \\ \boldsymbol{e}^{(l)} \end{bmatrix} = \begin{bmatrix} 1 & 1 & 1 & \dots & 1 & 1 \\ 1 & 1 & 1 & \dots & 1 & -1 \\ 1 & 1 & 1 & \dots & -1 & 1 \\ 1 & 1 & 1 & \dots & -1 & -1 \\ \vdots & \vdots & \vdots & \ddots & \vdots & \vdots \\ 1 & -1 & -1 & \dots & -1 & 1 \\ 1 & -1 & -1 & \dots & -1 & -1 \end{bmatrix}_{l \times m} . \tag{12}$$

For instance, for the cases of $m = 2$ and $m = 3$, we have

$$S^{(2)} = \begin{bmatrix} 1 & 1 \\ 1 & -1 \end{bmatrix}, \quad S^{(3)} = \begin{bmatrix} 1 & 1 & 1 \\ 1 & 1 & -1 \\ 1 & -1 & 1 \\ 1 & -1 & -1 \end{bmatrix} .$$

$S^{(m)}$ helps us to construct the separable bases with phase–shifted *cosines* in $m-$dimensions, which we discuss next.

### 3.2 Cosine Phase-Shifted Basis for $m-$dimensional Input

In a similar fashion as in the two–dimensional case, $H_{\boldsymbol{n}}$ represents the linear combination of $l = 2^{m-1}$ phase–shifted *cosines*, which is equivalent to a linear combination of $2^m$ separable Fourier basis in Equation 8 and can be written as

$$H_{\boldsymbol{n}} = \sum_{i=1}^{l} A_i^{(\boldsymbol{n})} \cos \left( (\boldsymbol{e}^{(i)} \odot \boldsymbol{n}) \cdot \boldsymbol{x} - \phi_i^{(\boldsymbol{n})} \right),$$

where $\boldsymbol{e}^{(i)}$ is the $i^{th}$ row of the $m-$Lexi Sign Matrix $S$ and $\boldsymbol{e}^{(i)} \odot \boldsymbol{n}$ represents its Hadamard (element wise) product with the frequency vector $\boldsymbol{n}$. The $m-$dimensional Fourier series can be expressed in a single compact summation as

$$\widehat{f}_{NH}(\boldsymbol{x}) = \sum_{\boldsymbol{n} \in \mathcal{N}} H_{\boldsymbol{n}}. \tag{13}$$

The relationship between the Fourier coefficients from Equation 8 and the amplitudes and phase shifts of this equivalent form Equation 13 is presented below. Stating this relationship for a general dimension $m$ requires

a more descriptive framework than the sequential indexing (*e.g.*, $a_1^{(n)}, a_2^{(n)}$) suitable for the two–dimensional case. We aim to calculate the coefficient $a_k^{(n)}$ corresponding to the basis function

$$\prod_{p \in \mathcal{I}_k^c} \cos(n_p x_p) \prod_{q \in \mathcal{I}_k} \sin(n_q x_q),$$

where the set $\mathcal{I}_k \subseteq \{1, \ldots, m\}$ and its complement $\mathcal{I}_k^c = \{1, \ldots, m\} \setminus \mathcal{I}_k$ specify the indices of the *sine* and cosine terms, respectively, for the basis corresponding to the index $k \in \{1, \ldots, 2^m\}$. The index set $\mathcal{I}_k$ is determined from the sequential index $k$ using the $m$–bit binary representation of $k-1$: an index $j$ is included in $\mathcal{I}_k$ if and only if the $j^{th}$ bit from the left is 1. Given the amplitudes $A_i^{(n)}$ and the phase shifts $\phi_i^{(n)}$, the separable product–basis Fourier coefficients $a_k^{(n)}$, are related by

$$a_k^{(n)} = \begin{cases} \displaystyle\sum_{i=1}^{l} A_i^{(n)} s_{i,k} \cos \phi_i^{(n)}, & |\mathcal{I}_k| \equiv 0 \pmod 4, \\ \displaystyle\sum_{i=1}^{l} A_i^{(n)} s_{i,k} \sin \phi_i^{(n)}, & |\mathcal{I}_k| \equiv 1 \pmod 4, \\ -\displaystyle\sum_{i=1}^{l} A_i^{(n)} s_{i,k} \cos \phi_i^{(n)}, & |\mathcal{I}_k| \equiv 2 \pmod 4, \\ -\displaystyle\sum_{i=1}^{l} A_i^{(n)} s_{i,k} \sin \phi_i^{(n)}, & |\mathcal{I}_k| \equiv 3 \pmod 4, \end{cases}$$

where the sign factor $s_{i,k} \in \{-1, 1\}$ is defined as

$$s_{i,k} = \prod_{j \in \mathcal{I}_k} e_j^{(i)}.$$

## 4 Model Construction and Architecture

In this section, we discuss the architectural construction of FLMs. We begin with the standard definition of a single–layer feedforward NN followed by a formal definition of the FLM.

**Definition 2 (Single-Layer Feedforward Neural Network)** *A single–hidden–layer NN with an input vector $\boldsymbol{x} \in \mathbb{R}^m$, $l$ hidden neurons, and a single output $\widehat{f}(\boldsymbol{x}) \in \mathbb{R}$ is a function defined as:*

$$\widehat{f}(\boldsymbol{x}) = g\left(\sum_{j=1}^{l} w_j^{out} \cdot \sigma(\boldsymbol{w}_j^{in} \cdot \boldsymbol{x} - b_j)\right)$$

*where $\boldsymbol{w}_j^{in}$ is the input–to–hidden weight vector, $b_j$ is the hidden neuron bias, $\sigma(\cdot)$ is the hidden layer activation function, $w_j^{out}$ is the hidden–to–output weight, and $g(\cdot)$ is the output layer activation function.*

**Definition 3 (Fourier Learning Machine (FLM))** *The FLM is composed of $N$ parallel sub–networks, each consisting of a complete single–layer NN conforming to Definition 2, with its hidden activation function $\sigma : x \mapsto \cos(x)$, and its output activation, $g : x \mapsto x$. The weight applied to the input to the top neuron in the hidden–layer of each sub–network is set to be a row vector $\boldsymbol{n}$. To each subsequent neuron in the hidden-layer, the same weight vector is used, with the respective sign of its components determined by the $m$–Lexi Sign Matrix. The final output of the entire FLM is the ordinary sum of the outputs from all these individual sub–networks.*

Next, we present three instances of possible sub–networks, first for the two–input and three–input cases and then for the general $m-$input case.

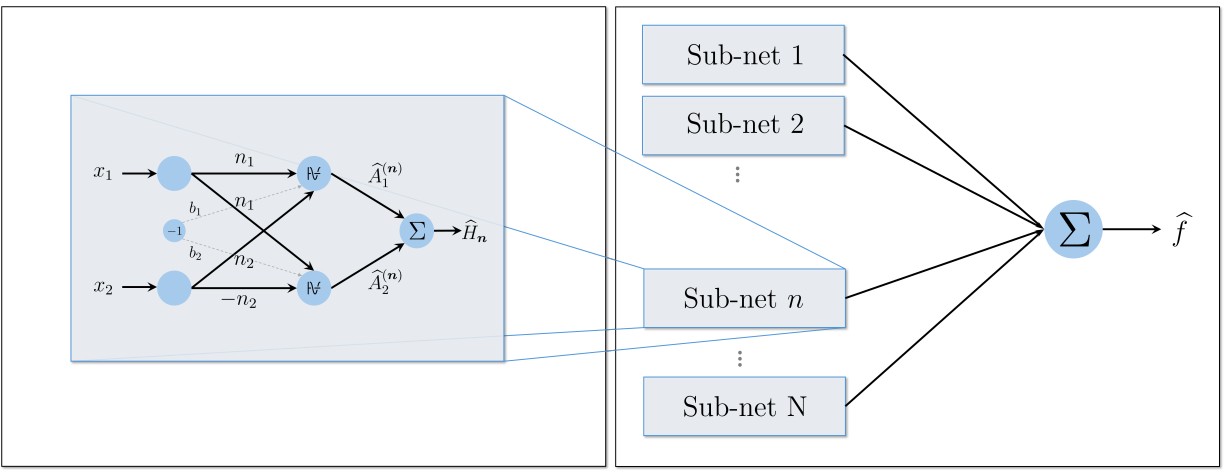

Figure 2: FLM with two inputs. The sub–network associated with frequency pair $\boldsymbol{n}$ (Left) and the main network with $N$ sub–networks (Right).

**Example 4.1 (FLM Sub–networks with Two Inputs)** *An FLM with two inputs is composed of N sub–networks, each parameterized by a learnable weight vector $\boldsymbol{n}$, with input $(x_1, x_2)$, two hidden–neurons with a cosine activation function and bias. The vector $\boldsymbol{n} = (n_1, n_2)$ contains the weights from the input–layer to the first hidden–neuron. For the second hidden–neuron, the same weights are reused except that the second element $n_2$ has opposite sign. The output of the sub–network is the weighted sum of outputs of the two cosine–activated hidden–neurons, i.e.,*

$$\widehat{H}_{\boldsymbol{n}} = \widehat{A}_1^{(\boldsymbol{n})} \cos(n_1 x_1 + n_2 x_2 - b_1^{(\boldsymbol{n})}) + \widehat{A}_2^{(\boldsymbol{n})} \cos(n_1 x_1 - n_2 x_2 - b_2^{(\boldsymbol{n})}).$$

*The output of the FLM is the ordinary sum of the sub–network outputs*

$$\widehat{f}(x_1, x_2) = \sum_{\boldsymbol{n} \in \mathcal{N}} \widehat{H}_{\boldsymbol{n}}(x_1, x_2), \tag{14}$$

*where $\mathcal{N}$ is the set of learnable frequency vectors, one from each sub–network. Equation 14 has the same form of the Fourier series in Equation 10. The hidden–to–output layer weights $\widehat{A}_i^{(\boldsymbol{n})}$ of the sub–network can be interpreted as the amplitudes $A_i^{(\boldsymbol{n})}$ and the biases $b_i^{(\boldsymbol{n})}$ as phase shifts $\phi_i^{(\boldsymbol{n})}$, respectively, for $i = 1, 2$. Fig. 2 illustrates a FLM sub–network with two inputs and the FLM itself.*

Additionally, a FLM with 3 inputs is shown in Fig. 3 alongside its underlying $S^{(3)}$ matrix. Now we discuss the structure of the FLM sub–networks in general for $m$–inputs and explain the interpretation of the multidimensional FLM.

**Example 4.2 (FLM Sub–Networks with $m$ inputs)** *An $m$–input FLM has $N$ parallel sub–networks, each of which is associated with a weight vector $\boldsymbol{n} = (n_1, n_2, \ldots, n_m)$. It has inputs $x_1, x_2, \ldots, x_m$ and $l = 2^{m-1}$ hidden nodes with cosine activation function and biases $b_i^{(\boldsymbol{n})}$ for $i = 1, \ldots, l$. The input–to–hidden layer weights for the $i^{th}$ node of the sub–network is the Hadamard (element wise) product $\boldsymbol{e}^{(i)} \odot \boldsymbol{n}$ where the row vector $\boldsymbol{e}^{(i)}$ is the $i^{th}$ row of the $m$–Lexi Sign Matrix $S^{(m)}$ (Equation 12). An illustration of the $m$–dimensional FLM sub–network is presented in Fig. 4.*

*The same node operations to the 2–dimensional case are used and the weighted sum of the activation is calculated. Then the output of the sub–network becomes*

$$\widehat{H}_{\boldsymbol{n}} = \sum_{i=1}^{l} \widehat{A}_i^{(\boldsymbol{n})} \cos \left[ \left( e_1^{(i)} n_1 x_1 + e_2^{(i)} n_2 x_2 + \cdots + e_m^{(i)} n_m x_m \right) - b_i^{(\boldsymbol{n})} \right]$$

$$= \sum_{i=1}^{l} \widehat{A}_i^{(\boldsymbol{n})} \cos \left( (\boldsymbol{e}^{(i)} \odot \boldsymbol{n}) \cdot \boldsymbol{x} - b_i^{(\boldsymbol{n})} \right).$$

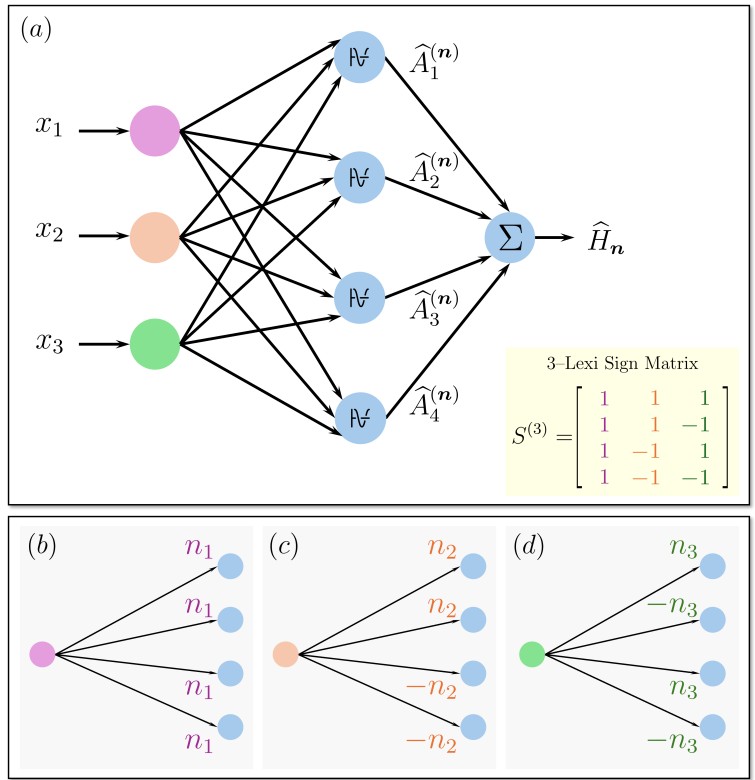

Figure 3: $(a)$ FLM with 3 inputs and the $S^{(3)}$ matrix defining the sign of the frequency components $n_1$, $n_2$ and $n_3$. These components are illustrated, respectively, in $(b) - (d)$.

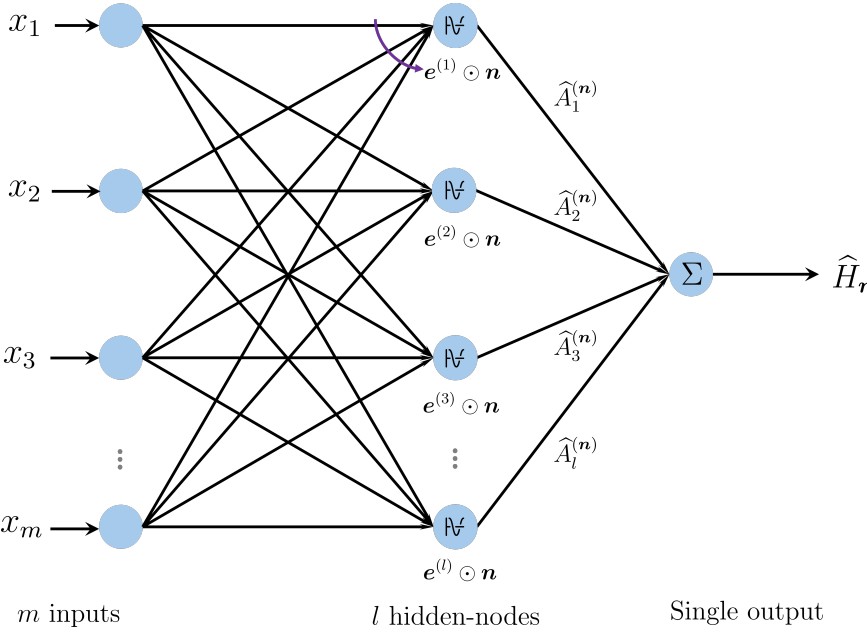

Figure 4: $m-$dimensional FLM sub–network (biases are not shown).

*Therefore, the output of the main network is given by*

$$\widehat{f}(x_1, x_2, \ldots, x_m) = \sum_{\boldsymbol{n} \in \mathcal{N}} \widehat{H}_{\boldsymbol{n}}(x_1, x_2, \ldots, x_m),$$

*where $\mathcal{N}$ is the finite set containing the learnable frequency vector from each sub–network. Notice that $\widehat{H}_{\boldsymbol{n}}$ matches the form of $H_{\boldsymbol{n}}$ and, consequently, $\widehat{f}$ with the Fourier series $\widehat{f}_{NH}$ itself in Equation 13. The hidden–to–output layer weights $\widehat{A}_i^{(\boldsymbol{n})}$ of the sub–network can be interpreted as the amplitudes $A_i^{(\boldsymbol{n})}$ and the biases $b_i^{(\boldsymbol{n})}$ as phase shifts $\phi_i^{(\boldsymbol{n})}$, respectively for $i = 1, \ldots, l$. We note that the $2^{m-1}$ neurons per sub–network grow exponentially with m, which is an inherent consequence of representing an m–dimensional Fourier basis in separable form. Thus, each sub–network with $2^{m-1}$ neurons explicitly generates a composition of $2^{m-1}$ m–dimensional phase–shifted cosines or implicitly $2^m$ separable Fourier basis with a specific frequency vector. The relative contributions of each individual Fourier component are learned through the sub–network biases and output layer weights. Although the network does not perform explicit projection onto the orthogonal bases, it learns both the amplitude and phase of each learned basis function, enabling a structured and interpretable approximation of the target map.*

We have demonstrated in this section that a MLP–like, feedforward NN with traditional node operations can represent a *m*–dimensional nonharmonic Fourier series. In the next section, we compare the performance of our FLM model against other NN models.

## 5 Computational Results

In this section we test and compare the performance of our proposed FLM model in the computation of numerical solutions of four PDEs and an OCP arising from an odd–circulant evolutionary game. Additionally, we design a series of computational experiments aimed at comparing the performance of our proposed model against other traditionally employed NN architectures.

The PDEs considered herein are: the one–dimensional heat equation, the two–dimensional Poisson equation, the generalized Black–Scholes model, and the one–dimensional inviscid Burgers' equation. Besides the emerging interest in machine learning techniques for the solution of PDEs, demonstrated, for example, in Ngom & Marin (2021); Ma et al. (2022), the solution of OCPs by such methods was also recently brought to attention (see, *e.g.,* Nicolosi et al. (2022); Nicolosi & Griffin (2023); Nicolosi et al. (2025); Rubel & Nicolosi (2025); Lundqvist & Oliveira (2025)). Thus, we consider in this paper an OCP arising from the evolutionary dynamics of a Rock–Paper–Scissors game, as studied in Griffin & Fan (2022). Its nonlinear dynamics alongside its state space configuration make this OCP particularly challenging to be solved analytically, so our solution shall be tested against an (indirect) numerical solution based on the Pontryagin's Minimum Principle (PMP). While traditional techniques like Finite Elements and Finite Differences are well–established methods, the recent rise in interest on machine learning-based techniques has introduced a new class of powerful, mesh–free solvers based on NNs (Raissi et al., 2019). In this section, we demonstrate the efficacy of our proposed FLMs in solving several benchmark PDEs and a class of OCPs.

In all experiments, the frequency vectors of the FLMs are initialized at a finite subset of integer grid points in $\mathbb{N}_0^m$, chosen sequentially from the lattice: for example, $(0,0), (0,1), (1,0), (1,1)$ in the two–dimensional case with four sub–networks. However, we do not constrain them to remain within $\mathbb{R}_{\geq 0}^m$ during the training. The reason for this lies in the inclusion of sign–flipped versions of the frequency vectors within each sub–network, together with the even symmetry of the *cosine* function. If any component of the frequency vector becomes negative, the FLM automatically accounts for its symmetrical equivalent, thus adjusting the corresponding coefficients to mitigate this effect. In each sub–network, biases (phase–shifts) are randomly sampled from the normal distribution $N(0, \frac{\pi}{3})$ and hidden–to–output layer weights (amplitudes) are initialized to 0. This initialization strategy is primarily motivated by empirical observations from our experiments and we keep it as an open question whether there is any mathematical way of determining a better set of initial weights. Next, we introduce each PDE considered, as well as the customized loss function.

## 5.1 Numerical Experiments for Partial Differential Equations

We consider a general nonlinear PDE defined on the spatio–temporal domain $\Omega \times [0, T]$, where $\Omega \subset \mathbb{R}^{m-1}$, and is given by

$$
\begin{aligned}
\mathcal{F}[u](\boldsymbol{x}, t) &= 0, & (\boldsymbol{x}, t) &\in \Omega \times (0, T], \\
u(\boldsymbol{x}, 0) &= u_0(\boldsymbol{x}), & \boldsymbol{x} &\in \Omega, \\
\mathcal{B}[u](\boldsymbol{x}, t) &= g(\boldsymbol{x}, t), & (\boldsymbol{x}, t) &\in \partial\Omega \times (0, T].
\end{aligned}
$$

Here, $\partial\Omega$ is the spatial boundary of $\Omega$, $\mathcal{F}[\cdot]$ and $\mathcal{B}[\cdot]$ denote general (possibly nonlinear) differential operators, $u(\boldsymbol{x}, t)$ is the unknown solution, and $u_0(\boldsymbol{x})$ and $g(\boldsymbol{x}, t)$ define the initial and boundary conditions, respectively.

We approximate the solution $u(\boldsymbol{x}, t)$ using a FLM $u_\Theta(\boldsymbol{x}, t)$ parameterized by its weights and biases, collectively denoted as $\Theta$. The network is trained by minimizing a composite loss function $\mathcal{L}(\Theta)$ (such as in Cuomo et al. (2022)), that penalizes deviations from the governing PDE and its associated initial and boundary conditions

$$
\mathcal{L}(\Theta) = \mathcal{L}_{\text{IC}} + \mathcal{L}_{\text{BC}} + \mathcal{L}_{\text{PDE}}, \tag{15}
$$

where

$$
\mathcal{L}_{\text{IC}} = \frac{1}{N_{\text{IC}}} \sum_{i=1}^{N_{\text{IC}}} \left| u_\Theta(\boldsymbol{x}_i, 0) - u_0(\boldsymbol{x}_i) \right|^2,
$$

$$
\mathcal{L}_{\text{BC}} = \frac{1}{N_{\text{BC}}} \sum_{j=1}^{N_{\text{BC}}} \left| \mathcal{B}[u_\Theta](\boldsymbol{x}_j, t_j) - g(\boldsymbol{x}_j, t_j) \right|^2,
$$

$$
\mathcal{L}_{\text{PDE}} = \frac{1}{N_{\text{PDE}}} \sum_{k=1}^{N_{\text{PDE}}} \left| \mathcal{F}[u_\Theta](\boldsymbol{x}_k, t_k) \right|^2.
$$

Each component of the loss function Equation 15 is a mean square error (MSE) computed over a distinct set of collocation points, defining the training set. These points are randomly sampled from their corresponding domains: $\{\boldsymbol{x}_i\}$ from the spatial domain $\Omega$ at $t = 0$, $\{(\boldsymbol{x}_j, t_j)\}$ from the boundary $\partial\Omega \times [0, T]$, and $\{(\boldsymbol{x}_k, t_k)\}$ from the interior $\Omega \times (0, T]$. The integers $N_{\text{IC}}$, $N_{\text{BC}}$, and $N_{\text{PDE}}$ are the number of the training points used to enforce the initial, boundary, and PDE constraints in the loss function, respectively. The differential operators $\mathcal{F}[\cdot]$ and $\mathcal{B}[\cdot]$ are evaluated on these points using automatic differentiation, and the loss function is then minimized with adaptive momentum gradient descent algorithm (ADAM) (Kingma & Ba, 2014).

To test the model's final accuracy, a separate test set is constructed from a evenly spaced grid covering the entire domain. On this test set, we compare the FLM–approximated solution to the exact solution using three standard error metrics. We have reported the MSE, which is the average of squared differences; the Mean Absolute Errors (MAE), the average of absolute differences; and the Max Error, which is the largest point–wise absolute difference. Now we examine the effectiveness of FLMs to solve PDEs and see how their performance compared to those of other popular NN architectures, listed in Table 1. Each network in the table has $N_{\text{hidden}}$ neurons per hidden–layer.

Table 1: Description of other NN architectures for performance benchmarking against FLMs.

| Model | Description |
|---|---|
| SIREN–1 | SIREN with 1 hidden layer ($\omega_0 = 1$) |
| SIREN–2 | SIREN with 2 hidden layers ($\omega_0 = 1$) |
| SIREN–3 | SIREN with 3 hidden layers ($\omega_0 = 1$) |
| V Relu | Vanilla feedforward NN with Relu activation function and 3 hidden layers |
| V LRelu | Vanilla feedforward NN with Leaky Relu activation function and 3 hidden layers |
| V Tanh | Vanilla feedforward NN with Tanh activation function and 3 hidden layers |

We trained FLMs and diverse NN models with multiple hyperparameter configurations, selected the best using a hierarchical statistical testing procedure, and further trained the top configuration for 30,000 epochs with a stricter loss tolerance. Details are shown in Appendix C.

**Example 5.1 (One–Dimensional Heat Equation)** *We consider the one–dimensional heat equation on the domain $x \in [0, 1]$, $t \in [0, 1]$, given by*

$$\frac{\partial u}{\partial t} = \alpha \frac{\partial^2 u}{\partial x^2}, \quad x \in (0, 1), \ t \in (0, 1],$$

$$u(x, 0) = \sin(\pi x), \quad x \in [0, 1].$$

$$u(0, t) = 0, \quad u(1, t) = 0, \quad t \in [0, 1],$$

*where the thermal diffusivity constant is set to be $\alpha = 0.1$. The analytical solution to this initial–boundary value problem is $u(x, t) = \sin(\pi x) e^{-\alpha \pi^2 t}$.*

Fig. 5a illustrates the comparison between the exact solution and the FLM approximated solution. The performances of all models are summarized in Table 2a.

**Example 5.2 (Two-Dimensional Poisson Equation)** *Consider the Poisson equation on the unit square with homogeneous Dirichlet boundary conditions:*

$$\frac{\partial^2 u}{\partial x^2} + \frac{\partial^2 u}{\partial y^2} = -2\pi^2 \sin(\pi x) \sin(\pi y), \quad (x, y) \in \Omega = (0, 1)^2,$$

$$u(x, y) = 0, \quad (x, y) \in \partial\Omega.$$

*The exact solution is $u(x, y) = \sin(\pi x) \sin(\pi y)$.*

Fig. 5b shows the comparison between the exact solution and the FLM approximated solution. The results are summarized in Table 2b.

**Example 5.3 (Generalized Black-Scholes Equation)** *Consider the PDE extracted from Ma et al. (2022), with initial and boundary condition*

$$\frac{\partial u}{\partial t} = a(x, t) \frac{\partial^2 u}{\partial x^2} + b(x, t) \frac{\partial u}{\partial x} + c(x, t)u + d(x, t), \quad (x, t) \in (-2, 2) \times (0, 1],$$

$$u(x, 0) = e^x, \quad x \in [-2, 2],$$

$$u(-2, t) = e^{-2-t}, \quad u(2, t) = e^{2-t}, \quad t \in [0, 1].$$

*The coefficient functions are chosen as*

$$a(x, t) = 0.08 \left(2 + (1 - t) \sin(e^x)\right)^2,$$

$$b(x, t) = 0.06(1 + t \, e^{-e^x}) - 0.02 e^{-t - e^x} - a(x, t),$$

$$c(x, t) = -0.06(1 + t \, e^{-e^x}),$$

$$d(x, t) = 0.02 e^{x - e^x - 2t} - e^{x - t}.$$

*The exact solution is $u(x, t) = e^{x - t}$.*

The comparison between the exact solution and the FLM approximated solution is presented in Fig. 5c. The results are presented in Table 2c.

**Example 5.4 (Inviscid Burgers' Equation)**

$$\frac{\partial u}{\partial t} + u \frac{\partial u}{\partial x} = 0, \qquad 0 \le x \le 1, \ 0 \le t \le 1,$$

$$u(x, 0) = 1.0 + 0.35 \sin(2\pi x),$$

$$u(0, t) = u(1, t).$$

This PDE has discontinuous solution and the boundary condition $u(0,t) = u(1,t)$ induces a discontinuity in the boundary as well. In this case, we compare the approximated solution with the numerical solution obtained by Godunov's method. The results are illustrated in Fig. 5d and summarized in Table 2d.

Table 2: Approximation Errors for Various PDEs Using Different Models

(a) Heat PDE

| Models | MSE | MAE | Max Error |
|---|---|---|---|
| FLM | $6.24 \times 10^{-8}$ | $2.01 \times 10^{-4}$ | $7.37 \times 10^{-4}$ |
| SIREN–1 | $5.07 \times 10^{-7}$ | $5.29 \times 10^{-4}$ | $1.97 \times 10^{-3}$ |
| SIREN–2 | $3.07 \times 10^{-6}$ | $1.18 \times 10^{-3}$ | $8.06 \times 10^{-3}$ |
| SIREN–3 | $5.40 \times 10^{-5}$ | $3.90 \times 10^{-3}$ | $2.00 \times 10^{-2}$ |
| V Relu | $6.39 \times 10^{-2}$ | $2.03 \times 10^{-1}$ | $6.72 \times 10^{-1}$ |
| V Tan | $9.24 \times 10^{-6}$ | $1.39 \times 10^{-3}$ | $4.08 \times 10^{-3}$ |

(b) Poisson PDE

| Models | MSE | MAE | Max Error |
|---|---|---|---|
| FLM | $1.34 \times 10^{-7}$ | $2.04 \times 10^{-4}$ | $6.74 \times 10^{-4}$ |
| SIREN–1 | $2.01 \times 10^{-7}$ | $3.95 \times 10^{-4}$ | $1.86 \times 10^{-3}$ |
| SIREN–2 | $8.07 \times 10^{-5}$ | $7.09 \times 10^{-3}$ | $3.43 \times 10^{-2}$ |
| V Relu | $2.45 \times 10^{-1}$ | $3.97 \times 10^{-1}$ | $9.99 \times 10^{-1}$ |
| V Tan | $2.64 \times 10^{-4}$ | $9.95 \times 10^{-3}$ | $6.56 \times 10^{-2}$ |

(c) Generalized Black–Scholes PDE

| Models | MSE | MAE | Max Error |
|---|---|---|---|
| FLM | $3.80 \times 10^{-7}$ | $4.44 \times 10^{-4}$ | $4.53 \times 10^{-3}$ |
| SIREN–1 | $4.09 \times 10^{-5}$ | $4.84 \times 10^{-3}$ | $3.44 \times 10^{-2}$ |
| SIREN–2 | $8.37 \times 10^{-7}$ | $7.39 \times 10^{-4}$ | $2.91 \times 10^{-3}$ |
| V LRelu | $6.53 \times 10^{-2}$ | $1.75 \times 10^{-1}$ | $9.85 \times 10^{-1}$ |
| V Tan | $6.06 \times 10^{-6}$ | $2.02 \times 10^{-3}$ | $5.54 \times 10^{-3}$ |

(d) Burgers' PDE

| Models | MSE | MAE | Max Error |
|---|---|---|---|
| FLM | $6.55 \times 10^{-3}$ | $4.07 \times 10^{-2}$ | $3.73 \times 10^{-1}$ |
| SIREN–1 | $2.13 \times 10^{-2}$ | $1.15 \times 10^{-1}$ | $3.89 \times 10^{-1}$ |
| SIREN–2 | $9.80 \times 10^{-3}$ | $5.86 \times 10^{-2}$ | $3.59 \times 10^{-1}$ |
| SIREN–3 | $6.96 \times 10^{-3}$ | $4.34 \times 10^{-2}$ | $3.68 \times 10^{-1}$ |
| V Relu | $3.09 \times 10^{-2}$ | $1.46 \times 10^{-1}$ | $3.74 \times 10^{-1}$ |
| V Tan | $4.58 \times 10^{-3}$ | $2.88 \times 10^{-2}$ | $4.44 \times 10^{-1}$ |

The numerical experiments demonstrate that FLMs consistently outperform other NN models across a set of multidimensional PDEs. While SIREN and vanilla feedforward NNs perform well when the target function contains similar mathematical behavior as the activation function they use, FLMs achieve significantly lower error metrics and faster convergence (see tables in Appendix C) regardless of the PDE. In terms of the MSE and MAE metrics, FLMs were shown to perform better than all models considered for each PDE, with the exception of Burgers' PDE. This difficulty in approximating the solution of Burgers' equation was anticipated, mainly because of the presence of a discontinuity. However, the FLM was still able to attain an MSE of the same order of magnitude of the best performing model, the Vanilla Tan NN. In terms of Max Error, FLMs also showed an overall superior performance, even though not always dominating the other models. In general, out of all of the competing models, SIREN NNs are the models with the closest performance to that of FLMs. In the next section, we apply the proposed FLM model to the solution of a family of OCPs.

## 5.2 Numerical Experiments for Optimal Control Problems

Both NN and Fourier series have been proven to be very effective to solve OCPs in recent years (Effati & Pakdaman, 2013; Nicolosi et al., 2022; Nicolosi & Griffin, 2023; Nicolosi et al., 2025; Rubel & Nicolosi, 2025). To show the effectiveness of FLMs to solve OCPs, we consider the optimal control of a system governed by the dynamics of a 3–strategy odd–circulant evolutionary game, namely, a Rock–Paper–Scissors game (Griffin & Fan, 2022). The state of the system, a vector $\boldsymbol{u}(t) = (u_1, u_2, u_3)^\top$ with $u_1 + u_2 + u_3 = 1$, represents the proportion of a population playing each strategy. The goal is to push these proportions toward the desired equilibrium point $\boldsymbol{u}_{eq} = (1/3, 1/3, 1/3)^\top$ representing an equally balanced population.

The system's evolution is described by the replicator dynamics and is influenced by an external control input, $\gamma(t)$. This control actively alters the game's payoff matrix, which is defined as $\boldsymbol{A}_3(\gamma) = \boldsymbol{L}_3 + \gamma \boldsymbol{M}_3$, where $\boldsymbol{L}_3$ is the standard payoff matrix and $\boldsymbol{M}_3$ is an actuating matrix and given by

$$\boldsymbol{L}_3 = \begin{pmatrix} 0 & -1 & 1 \\ 1 & 0 & -1 \\ -1 & 1 & 0 \end{pmatrix}, \quad \boldsymbol{M}_3 = \begin{pmatrix} 0 & 0 & 1 \\ 1 & 0 & 0 \\ 0 & 1 & 0 \end{pmatrix}.$$

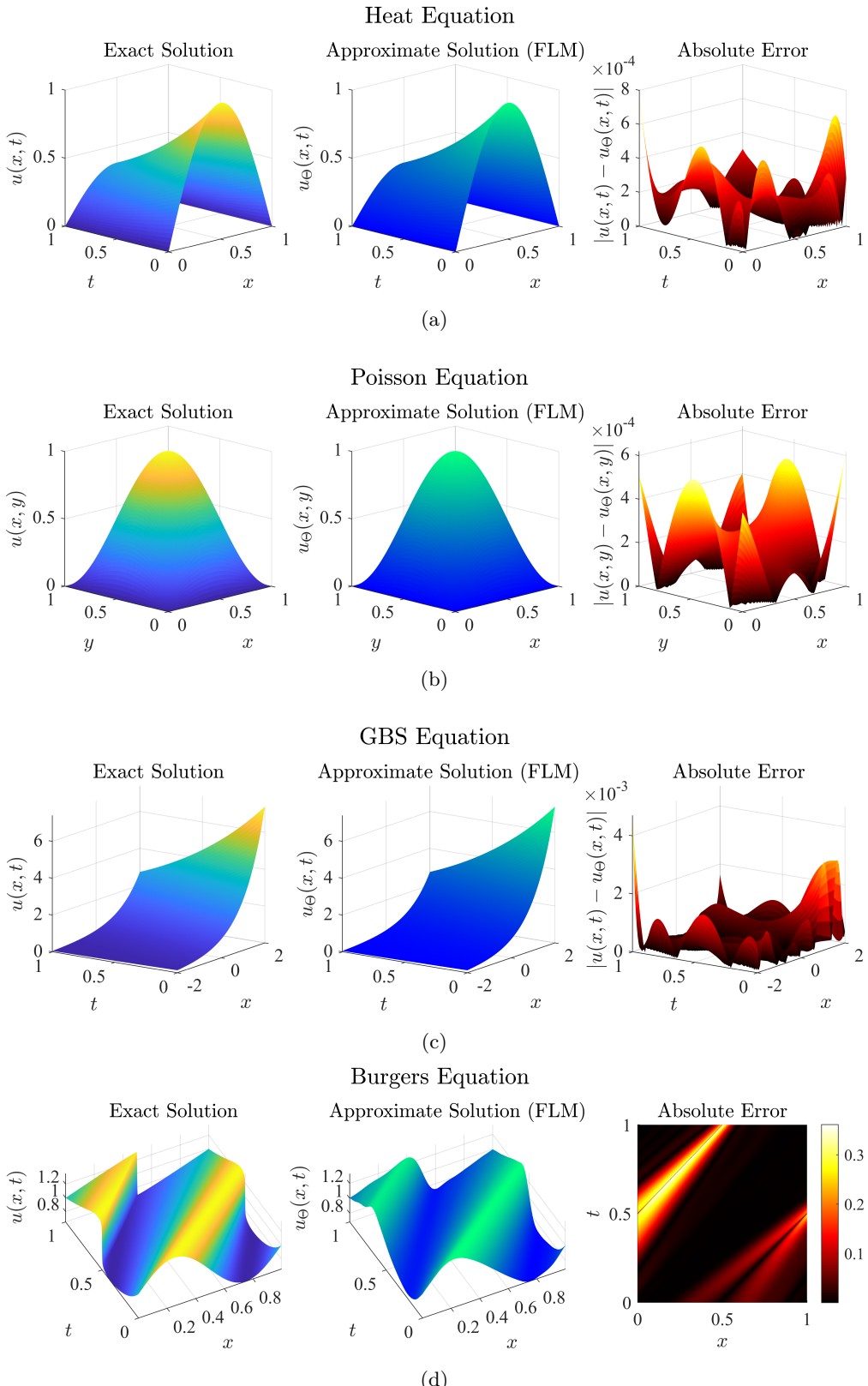

Figure 5: Solution surfaces of the Heat (a), Poisson (b), GBS (c), and Burgers' (d) equations, showing exact solution (left), FLM approximation (middle), and absolute error (right).

The objective is to find an optimal control strategy $\gamma(t)$ that minimizes a cost functional over a time horizon $[0, T]$. This cost has two components: a term that penalizes the squared distance of the state $\boldsymbol{u}(t)$ from the target equilibrium $\boldsymbol{u}_{eq}$, and a term that penalizes the squared control effort $\gamma(t)^2$.

An important point to note is that the given OCP possesses a cyclic symmetry. If $\boldsymbol{u}^*(t) = (u_1^*(t), u_2^*(t), u_3^*(t))^\top$ is the optimal state trajectory, $\gamma^*(t)$ is the optimal control and $J^*$ is the optimal objective value for an initial condition $\boldsymbol{u}_0 = (u_{01}, u_{02}, u_{03})^\top$, then any cyclic permutation of this initial state will yield the same optimal control solution and objective value. Consequently, the new optimal state trajectory will be the corresponding cyclic permutation of the original trajectory, $\boldsymbol{u}^*(t)$. For instance, if the initial state is permuted to $(u_{02}, u_{03}, u_{01})$, the optimal trajectory becomes $(u_2^*(t), u_3^*(t), u_1^*(t))$, while the optimal control and objective value remain unchanged. The numerical approximation of the solution of the OCP with our proposed model is presented in the next example.

Consider a class of the OCP described above, where the initial condition $\boldsymbol{u_0}$ varies withing a given range, written as

$$
\begin{aligned}
\min \quad & J(\boldsymbol{u}, \gamma) = \int_0^T \left( \frac{1}{2} \|\boldsymbol{u} - \boldsymbol{u}_{eq}\|_2^2 + \frac{r}{2} \gamma^2 \right) dt \\
\text{s.t.} \quad & \dot{\boldsymbol{u}} = \boldsymbol{F}(\boldsymbol{u}) + \gamma(t) \boldsymbol{G}(\boldsymbol{u}) \\
& \boldsymbol{u}(0) = \boldsymbol{u}_0 \in U_0,
\end{aligned}
\tag{16}
$$

where $\boldsymbol{u}(t) \in \Delta_2$ lies on the 2–simplex. The vector fields, for $i = 1, 2, 3$, are given by

$$
\begin{aligned}
\boldsymbol{F}_i(\boldsymbol{u}) &= u_i \left( (\boldsymbol{e}_i - \boldsymbol{u})^T \boldsymbol{L}_3 \boldsymbol{u} \right), \\
\boldsymbol{G}_i(\boldsymbol{u}) &= u_i \left( (\boldsymbol{e}_i - \boldsymbol{u})^T \boldsymbol{M}_3 \boldsymbol{u} \right).
\end{aligned}
\tag{17}
$$

Let $\widehat{u}_i(t, \boldsymbol{u_0}; \Theta_{u_i})$ be the FLM that learns the approximation of state variable $u_i(t, u_0)$, *i.e.*, $\boldsymbol{u} \approx \widehat{\boldsymbol{u}} = (\widehat{u}_1, \widehat{u}_2, \widehat{u}_3)^\top$ and $\widehat{\gamma}(t, \boldsymbol{u_0}; \Theta_\gamma)$ be the FLM that learns the control variable $\gamma(t, u_0)$ for any trajectory departing from $U_0$, where $\Theta = \{\Theta_{u_1}, \Theta_{u_2}, \Theta_{u_3}, \Theta_\gamma\}$ is the collection of all network parameters of the respective networks. Given the complex nature of the OCP above, the training process is different than it was for the PDEs and so is the construction of the training loss function. We train all the FLM simultaneously by minimizing the training loss using a penalty method (Ch. 9 in Bazaraa et al. (2006)). We define an unconstrained loss function that incorporates the objective functional and penalties for dynamics and initial condition violations as

$$
\mathcal{L}(\Theta) = \frac{1}{N_{\text{IC}}} \sum_{\boldsymbol{u}_0 \in U_0} \left[ \mathcal{J} + \frac{1}{2} \boldsymbol{\mu_1}^\top \boldsymbol{\mathcal{V}}_{\text{dyn}} + \frac{1}{2} \boldsymbol{\mu_2}^\top \boldsymbol{\mathcal{V}}_{\text{init}} \right],
\tag{18}
$$

with

$$
\begin{aligned}
\mathcal{J}(\Theta) &= \int_0^T \left( \frac{1}{2} \|\widehat{\boldsymbol{u}} - \boldsymbol{u}_{eq}\|^2 + \frac{r}{2} \widehat{\gamma}^2 \right) dt, \\
\boldsymbol{\mathcal{V}}_{\text{dyn}}(\Theta) &= \int_0^T \left( \dot{\widehat{\boldsymbol{u}}} - \boldsymbol{F}(\widehat{\boldsymbol{u}}) - \widehat{\gamma} \boldsymbol{G}(\widehat{\boldsymbol{u}}) \right) \odot \left( \dot{\widehat{\boldsymbol{u}}} - \boldsymbol{F}(\widehat{\boldsymbol{u}}) - \widehat{\gamma} \boldsymbol{G}(\widehat{\boldsymbol{u}}) \right) dt, \\
\boldsymbol{\mathcal{V}}_{\text{init}}(\Theta) &= (\widehat{\boldsymbol{u}}(0) - \boldsymbol{u}_0) \odot (\widehat{\boldsymbol{u}}(0) - \boldsymbol{u}_0),
\end{aligned}
$$

where $\boldsymbol{\mu_1}, \boldsymbol{\mu_2}$ are vectors of positive penalty coefficients. Minimizing the total loss $\mathcal{L}$ with respect to $\Theta_u$ and $\Theta_\gamma$ forces the FLM to approximate the optimal state trajectory and control signal that solve the original OCP. Next, we consider two cases: a fixed initial condition case, for a given $\boldsymbol{u_0}$, and a for a range of initial conditions $U_0 \subset \Delta_2$.

### 5.2.1 Fixed Initial Condition OCP

First we solve the OCP for fixed initial condition with a one–dimensional FLM, which is a NN with one hidden–layer and *cosine* activation function. We consider three such FLMs, two for the state variables $u_1$ and $u_2$ and one for the control $\gamma$. The third state variable is calculated using the relation $u_3 = 1 - u_1 - u_2$. The training is performed with default learning rate 0.001 and betas $\beta_1 = 0.9$ and $\beta_2 = 0.999$ with the ADAM

optimizer. To test the accuracy of our model, we compare its performance against a numerical solution obtained via PMP to formulate a two–point boundary value problem, which was then solved numerically to a high precision (tolerance of $10^{-8}$) using the `solve_bvp` function from Python's SciPy library. The FLM approximated solutions and the PMP–based numerical solutions for three different, fixed initial conditions are shown in Fig. 6. The approximated optimal objective function values have absolute percentage errors of 0.47%, 0.37%, and 0.34% respectively. Fig. 7 shows the state trajectory for the initial point $\boldsymbol{u}_0 = (0.20, 0.20, 0.60)^\top$. The parameters of the OCP, as in Griffin & Fan (2022), are chosen as $T = 6.0$, $r = 0.2$.

We note that the PMP–based method we compare our results against is an indirect method, thus it is contingent upon the existence of a Hamiltonian function associated to that particular system. On the other hand, our FLM–based method is a direct one (*i.e.*, it translates the OCP to a discrete, nonlinear optimization problem), neither based on the existence of a Hamiltonian nor on any convexity assumptions, thus applicable to solve both convex and nonconvex OCPs.

Figure 6: The PMP-based numerical solution vs the FLM approximation for different initial conditions with $N = 5$ sub–networks.

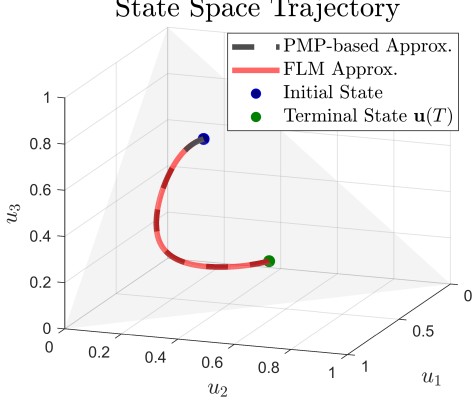

Figure 7: Optimal state trajectories for $u_0 = (0.20, 0.20, 0.60)$, $T = 6$ and $r = 0.2$.

### 5.2.2 Varying Initial Condition OCP

We solve the same OCP considered thus far, but now varying the initial condition $\boldsymbol{u_0}$ in a set $U_0$, defined as a disk on the 2–simplex, centered at $(0.20, 0.20, 0.60)$, with radius 0.15. Three FLMs with with $N = 27$ and three inputs $(t, u_{01}, u_{02})$ (we recall that the third state variable and its initial value is dependent on the other two), are used to approximate the two state and one control variables. The three FLMs are trained on 250 random points inside the disk, and then tested in 100 newly generated random points on the disk with center at $(0.20, 0.20, 0.60)$. Given the cyclic symmetry of the 2–simplex and the underlying payoff matrix, two other disks, centered at $(0.20, 0.60, 0.20)$ and $(0.60, 0.20, 0.20)$, provided a sampling of 200 extra points

used as initial conditions for testing. This choice of testing points is a mere illustration on how one could leverage the inherent cyclic symmetry of this OCP in order to reduce the training space required to learn the approximated optimal control law. This is attained by informing the controller on what disk it is actuating on, allowing it to use the appropriate coordinate control learned at the disk used for training. Nonetheless, such a training choice does not undermine the results reported herein had we chosen to focus on a single disk in the 2–simplex.

The hyperparameters are tuned and a learning rate of $5 \times 10^{-4}$ and beta values of 0.95 and 0.97, respectively, are selected for training. To quantify the approximation accuracy, we define the Mean Absolute Percentage Error (MAPE) on the objective function as follows:

$$\text{MAPE} = \frac{1}{100} \sum_{j=1}^{100} \underbrace{\left| \frac{J_j^{FLM} - J_j^{PMP}}{J_j^{PMP}} \right| \times 100\%}_{\text{Percentage Error in the } j^{th} \text{ test point}} \quad , \tag{19}$$

where $J_j^{FLM}$ and $J_j^{PMP}$ are the objective function values obtained from the FLM–approximated solution and the PMP–based solution, respectively, for the $j^{th}$ test point. Over the test points on each of the three disks, the MAPEs are calculated to be 0.81%, 0.85%, and 0.72% respectively, while the respective MAEs are calculated to be $2.54 \times 10^{-3}$, $2.74 \times 10^{-3}$ and $1.81 \times 10^{-3}$. Fig. 8 demonstrates the percentage errors and the absolute errors (point–wise) in the test points in the three disks. These numerical results represent a substantial improvement from the ones initially reported in Nicolosi (2023), where a finite Fourier series failed to approximate the optimal control law of this OCP in this case of varying initial condition.

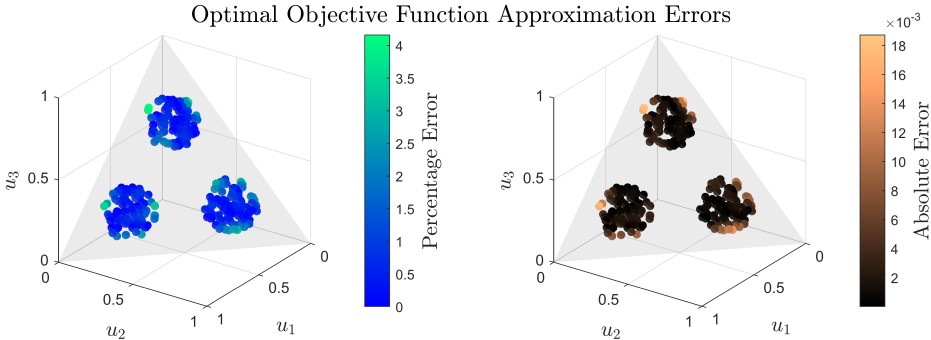

Figure 8: (Left) Absolute percentage errors and (right) absolute errors in the objective function value on test points within the circular disks on the simplex plane centered at $(0.2, 0.2, 0.6)$ and its cyclic permutations with a radius of 0.15.

### 5.2.3 Varying Initial Condition 5–dimensional OCP

To demonstrate the scalability of FLMs, we solve the 5–strategy extension of RPS, known as RPSSL (Rock–Paper–Scissors–Spock–Lizard). We extend the OCP in Equation 16 by replacing the matrices $L_3$ and $M_3$, respectively, in Equation 17 with

$$\boldsymbol{L}_5 = \begin{pmatrix} 0 & -1 & 1 & -1 & 1 \\ 1 & 0 & -1 & 1 & -1 \\ -1 & 1 & 0 & -1 & 1 \\ 1 & -1 & 1 & 0 & -1 \\ -1 & 1 & -1 & 1 & 0 \end{pmatrix}, \quad \boldsymbol{M}_5 = \begin{pmatrix} 0 & 0 & 1 & 0 & 1 \\ 1 & 0 & 0 & 1 & 0 \\ 0 & 1 & 0 & 0 & 1 \\ 1 & 0 & 1 & 0 & 0 \\ 0 & 1 & 0 & 1 & 0 \end{pmatrix}.$$

For this experiment, 4 FLMs with 5 inputs and $N = 16$ sub–networks were used to approximate 4 states and an FLM with 5 inputs and $N = 32$ sub–networks was employed to approximate the optimal control. In this

case, the equilibrium point is $\boldsymbol{u}_{eq} = (0.2, 0.2, 0.2, 0.2, 0.2)$. All of the FLMs were trained and tested following the same method discussed before with a learning rate of $10^{-4}$ and beta values of $0.95$ and $0.97$. Training the models with initial conditions (uniformly) sampled from a hyperdisk in the $4-$simplex $\Delta_4$ of radius $0.045^1$ centered at $(0.11, 0.11, 0.11, 0.11, 0.56)$, we have achieved an MAPE of $0.67\%$ with MAE of $2.10 \times 10^{-3}$.

Our results show that FLMs are capable of numerically solving PDEs with the same level of accuracy of other traditional NN models, often outperforming them in terms of error metrics considered. The OCP example also illustrated how FLMs can approximate the state trajectory and the optimal control of a high–dimensional family of OCPs.

## 6 Conclusions and Future Work

We have introduced the FLM, a neural architecture that learns optimal frequencies directly while systematically representing a multidimensional nonharmonic Fourier series, thereby bridging the gap between classical spectral methods and modern NN architectures. By jointly training frequencies, amplitudes, and phase shifts, FLMs overcome the restrictions of fixed harmonic Fourier series and adapts naturally to both periodic and nonperiodic problems. Its implementation as a NN allows it to leverage scientific computational and optimization tools, such as backpropagation, automatic differentiation and state–of–the–art solvers. Computational experiments on benchmark PDEs demonstrate that FLMs can match or outperform established architectures such as SIREN and vanilla feedforward NNs. Additionally, FLMs solve a family of OCPs with high accuracy, demonstrating its generalization capability and its potential for high–dimensional, nonlinear problems.

Future work will investigate an improved initialization scheme for the FLM weights (frequencies and amplitudes), potentially important for when the target function is dominated by high–frequency components. Additionally, one can consider applying FLMs to other domains, such as image and audio processing, medical signal processing, and time series analysis (a brief time series imputation study is presented in Appendix B). Finally, deeper theoretical relations, in particular with SSGP (Lázaro-Gredilla et al. (2010)) can be further investigated, potentially strengthening and better understanding the underlying principles of the spectral adaptability of both FLMs and SSGP models.

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

## Appendix

## A    Detailed Derivation of The Relation Between Separable Coefficients and Phase–Shifted *Cosine* Parameters in Two–Input Case

$$
\begin{aligned}
H_{\boldsymbol{n}} &= A_1^{(\boldsymbol{n})} \cos\left(n_1 x_1 + n_2 x_2 - \phi_1^{(\boldsymbol{n})}\right) + A_2^{(\boldsymbol{n})} \cos\left(n_1 x_1 - n_2 x_2 - \phi_2^{(\boldsymbol{n})}\right) \\
&= A_1^{(\boldsymbol{n})} \left[\cos\phi_1^{(\boldsymbol{n})} \cos(n_1 x_1 + n_2 x_2) + \sin\phi_1^{(\boldsymbol{n})} \sin(n_1 x_1 + n_2 x_2)\right] \\
&\quad + A_2^{(\boldsymbol{n})} \left[\cos\phi_2^{(\boldsymbol{n})} \cos(n_1 x_1 - n_2 x_2) + \sin\phi_2^{(\boldsymbol{n})} \sin(n_1 x_1 - n_2 x_2)\right] \\
&= A_1^{(\boldsymbol{n})} \Big[\cos\phi_1^{(\boldsymbol{n})} \Big(\cos(n_1 x_1)\cos(n_2 x_2) - \sin(n_1 x_1)\sin(n_2 x_2)\Big) \\
&\quad + \sin\phi_1^{(\boldsymbol{n})} \Big(\sin(n_1 x_1)\cos(n_2 x_2) + \cos(n_1 x_1)\sin(n_2 x_2)\Big)\Big] \\
&\quad + A_2^{(\boldsymbol{n})} \Big[\cos\phi_2^{(\boldsymbol{n})} \Big(\cos(n_1 x_1)\cos(n_2 x_2) + \sin(n_1 x_1)\sin(n_2 x_2)\Big) \\
&\quad + \sin\phi_2^{(\boldsymbol{n})} \Big(\sin(n_1 x_1)\cos(n_2 x_2) - \cos(n_1 x_1)\sin(n_2 x_2)\Big)\Big] \\
&= \left(A_1^{(\boldsymbol{n})} \cos\phi_1^{(\boldsymbol{n})} + A_2^{(\boldsymbol{n})} \cos\phi_2^{(\boldsymbol{n})}\right) \cos(n_1 x_1)\cos(n_2 x_2) \\
&\quad + \left(A_1^{(\boldsymbol{n})} \sin\phi_1^{(\boldsymbol{n})} - A_2^{(\boldsymbol{n})} \sin\phi_2^{(\boldsymbol{n})}\right) \cos(n_1 x_1)\sin(n_2 x_2) \\
&\quad + \left(A_1^{(\boldsymbol{n})} \sin\phi_1^{(\boldsymbol{n})} + A_2^{(\boldsymbol{n})} \sin\phi_2^{(\boldsymbol{n})}\right) \sin(n_1 x_1)\cos(n_2 x_2) \\
&\quad + \left(-A_1^{(\boldsymbol{n})} \cos\phi_1^{(\boldsymbol{n})} + A_2^{(\boldsymbol{n})} \cos\phi_2^{(\boldsymbol{n})}\right) \sin(n_1 x_1)\sin(n_2 x_2).
\end{aligned}
$$

Comparing with Equation 9 we get,

$$
\begin{aligned}
a_1^{(\boldsymbol{n})} &= A_1^{(\boldsymbol{n})} \cos\phi_1^{(\boldsymbol{n})} + A_2^{(\boldsymbol{n})} \cos\phi_2^{(\boldsymbol{n})}, \quad a_2^{(\boldsymbol{n})} = A_1^{(\boldsymbol{n})} \sin\phi_1^{(\boldsymbol{n})} - A_2^{(\boldsymbol{n})} \sin\phi_2^{(\boldsymbol{n})}, \\
a_3^{(\boldsymbol{n})} &= A_1^{(\boldsymbol{n})} \sin\phi_1^{(\boldsymbol{n})} + A_2^{(\boldsymbol{n})} \sin\phi_2^{(\boldsymbol{n})}, \quad a_4^{(\boldsymbol{n})} = -A_1^{(\boldsymbol{n})} \cos\phi_1^{(\boldsymbol{n})} + A_2^{(\boldsymbol{n})} \cos\phi_2^{(\boldsymbol{n})}.
\end{aligned}
$$

## B    FLM For Time Series Imputation

To demonstrate the potential of FLMs for time series imputation tasks, we present in this Appendix some numerical results derived from a brief set of computational experiments using data from the Electricity Transformer Temperature (ETT) dataset Zhou et al. (2021). This dataset is widely used in time series forecasting and imputation benchmarks due to its strong quasi–periodic patterns and significant fluctuations, making it a good testbed candidate to evaluate the frequency–learning capabilities of our model. We compare the performance of FLMs to the models presented in Du et al. (2024), also evaluated for the ETT data set.

### B.1    Dataset and Experimental Setup

We utilize the ETTh1 (Electricity Transformer Temperature – 1 hour) subset, which contains hourly records of oil temperature (OT) and other load features from electricity transformers over a period of two years. For this experiment, we focus on the univariate imputation of the OT channel only, which exhibits both daily and yearly seasonality alongside high–frequency oscillations.

We replicate the data preprocessing and experimental protocol present in Du et al. (2024):

- **Preprocessing:** The data is normalized using a standard scaler (zero mean, unit variance). The time index $t$ is mapped to the domain $[-\pi, \pi]$.

- **Splitting:** We employ a chronological split of 60% for training, 20% for validation, and 20% for testing.

- **Masking:** To simulate missing values, we apply random point-wise masking at two different missingness rates: 10% and 50%.

- **Training:** The model is trained to minimize the MSE on the observed (non-masked) data points of the training set. After hyperparameter tuning, we utilize the Adam optimizer with a learning rate of $1 \times 10^{-4}$ and a hidden layer size of $N = 8$. The training runs for 10,000 epochs with early stopping based on validation loss.

## B.2 Results

We report the FLM approximation error on the held-out masked values of the test set and the performance is evaluated using MAE. Table 3 summarizes the results averaged over 10 independent runs with different random seeds. For comparative baselines, we refer to the comprehensive results established in Du et al. (2024), noting that the FLM achieved a competitive performance (although not superior) with a significantly fewer number of parameters and much simpler architecture than most of the models considered in Du et al. (2024) (mostly Deep NN models). In particular, referring to the performance of models in Du et al. (2024) for the dataset considered, FLMs would rank 26 and 24 out of 29 for the cases of 10% and 50% of point–wise missing data, respectively. We emphasize that such performance was attained with a single–input FLM only. These preliminary results indicate that FLMs have great potential to be explored in this domain, and perhaps can be further studied or adapted for the specific task of time series imputation.

Table 3: FLM imputation results on ETTh1 (Mean $\pm$ SD over 10 seeds) against some of the performance of state-of-the-art models considered in Du et al. (2024) (** refers to the top performing and * to the lowest ranked one).

| | MAE |
|---|---|
| Model | Mean (Standard Deviation) |
| 10% Missing Rate | |
| FLM | 0.57 (0.15) |
| MRNN* (Yoon et al. (2018)) | 0.79 (0.02) |
| Mean (Du et al. (2024)) | 0.74 (−) |
| Median (Du et al. (2024)) | 0.71 (−) |
| SAITS** (Du et al. (2023)) | 0.14 (0.01) |
| 50% Missing Rate | |
| FLM | 0.66 (0.18) |
| AutoFormer* (Wu et al. (2021)) | 0.98 (0.01) |
| Mean (Du et al. (2024)) | 0.74 (−) |
| Median (Du et al. (2024)) | 0.71 (−) |
| SAITS** | 0.22 (0.01) |

# C Numerical Results for Hyperparameter Selection

When numerically solving each PDE, we trained FLMs and other NN models for 18–27 different hyperparameter configurations, with a maximum of 10,000 epochs and a training loss tolerance of $10^{-4}$. All error metrics (MSE, MAE, Max Error) and the number of epochs required to reach the specified training loss tolerance were averaged over 10 runs with different random seeds. The performances of all tested models applied to each PDE are reported in Tables 5–26. Table 4 shows the number of trainable parameters in each model for different number of hidden units. The effect of the number of sub–networks in approximating the solutions of the PDEs is shown in figure Fig. 9.

To select the best hyperparameter combination for each model applied to each PDE, we performed a hierarchical statistical testing procedure. For each metric (MSE, MAE, Max Error, and number of epochs, in that order), we first tested for normality using the Shapiro-Wilk test. If all hyperparameter sets were normally distributed ($p > 0.05$) in their respective metric, we applied one–way ANOVA followed by Tukey's HSD test for pairwise comparisons; otherwise, we used the nonparametric Friedman test followed by the Nemenyi test.

Table 4: Number of trainable parameters for different models. For FLM, $N$ denotes the number of sub–networks. For all other models, $N_{\text{hidden}}$ denotes the number of neurons in each hidden layer. For each row, parameter counts are reported in the same order as the listed $N$ or $N_{\text{hidden}}$ values.

| Model | Hidden Units | Formula | Trainable Parameters |
|---|---|---|---|
| FLM | $N \quad = \{04, 16, 25, 49, 64\}$ | $6N$ | $\{24, 96, 150, 294, 384\}$ |
| SIREN–1 | $N_{\text{hidden}} = \{06, 24, 37, 73, 93\}$ | $4N_{\text{hidden}} + 1$ | $\{25, 97, 149, 293, 373\}$ |
| SIREN–2 | $N_{\text{hidden}} = \{03, 08, 10, 15, 17\}$ | $N_{\text{hidden}}^2 + 5N_{\text{hidden}} + 1$ | $\{25, 105, 151, 301, 375\}$ |
| SIREN–3, V-ReLU, V-LReLU, V-Tanh | $N_{\text{hidden}} = \{02, 06, 07, 11, 12\}$ | $2N_{\text{hidden}}^2 + 6N_{\text{hidden}} + 1$ | $\{21, 109, 141, 309, 361\}$ |

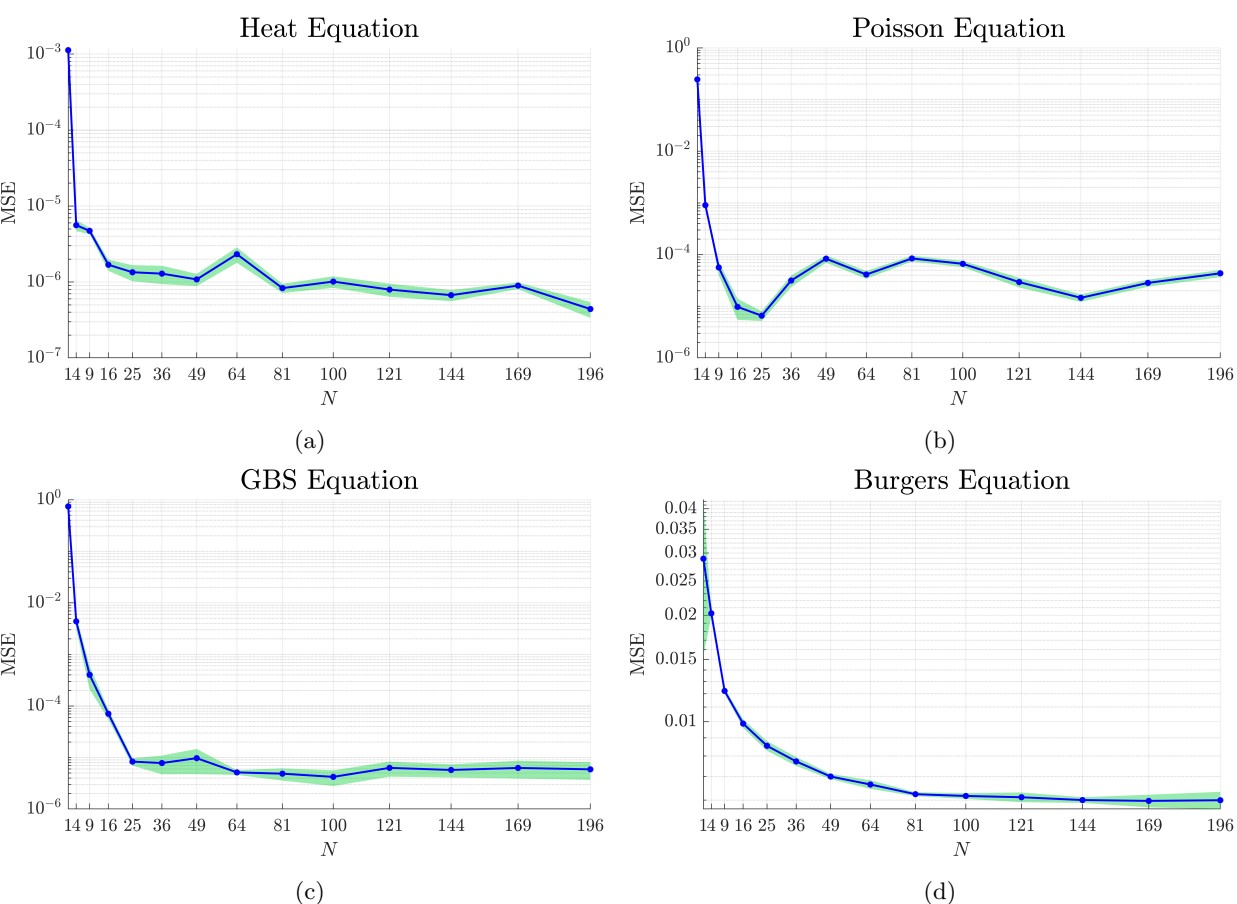

Figure 9: Effect on approximation MSE for the PDEs considered in Section 5 (shown in vertical axis in logarithmic scale) as $N$ increases. All FLMs were trained using default learning rate and beta values of ADAM optimizer in PyTorch with maximum epochs of $10,000$ and loss tolerance of $10^{-4}$.

We identified the set with the lowest respective metric mean (or average rank in the nonparametric case) for the current metric and grouped hyperparameter configurations not significantly different ($p > 0.05$ or within the critical difference) from it. This top group was carried forward to the next metric, repeating the process sequentially. If multiple groups remained statistically equivalent after all metrics had been evaluated, we selected the one with the smallest network size as the tiebreaker.

With the best performing hyperparameter configuration (highlighted in each table below) selected by following the aforementioned procedure, we train the NN models for $30,000$ new additional epochs with a training loss tolerance of $10^{-8}$, and report the error metrics for each PDE in their respective Tables 2a–2d.

Table 5: Heat Equation with FLM

| $N$ | LR | Betas | MSE $\pm$ SD | MAE $\pm$ SD | Max Error $\pm$ SD | Epochs $\pm$ SD |
|---|---|---|---|---|---|---|
| | | (0.9, 0.999) | (5.59 $\pm$ 1.51)E-06 | (1.88 $\pm$ 0.23)E-03 | (8.54 $\pm$ 1.58)E-03 | 8611 $\pm$ 1087 |
| | 0.001 | (0.95, 0.97) | (1.05 $\pm$ 0.26)E-05 | (2.54 $\pm$ 0.38)E-03 | (1.39 $\pm$ 0.32)E-02 | 2941 $\pm$ 627 |
| | | (0.97, 0.95) | (1.05 $\pm$ 0.23)E-05 | (2.57 $\pm$ 0.28)E-03 | (1.34 $\pm$ 0.23)E-02 | 2802 $\pm$ 275 |
| | | (0.9, 0.999) | (5.03 $\pm$ 1.36)E-06 | (1.76 $\pm$ 0.21)E-03 | (8.70 $\pm$ 2.08)E-03 | 3876 $\pm$ 1254 |
| 4 | 0.005 | (0.95, 0.97) | (8.71 $\pm$ 2.08)E-06 | (2.36 $\pm$ 0.29)E-03 | (1.13 $\pm$ 0.30)E-02 | 1173 $\pm$ 649 |
| | | (0.97, 0.95) | (7.42 $\pm$ 2.88)E-06 | (2.12 $\pm$ 0.39)E-03 | (1.13 $\pm$ 0.38)E-02 | 1129 $\pm$ 334 |
| | | (0.9, 0.999) | (4.89 $\pm$ 1.31)E-06 | (1.75 $\pm$ 0.22)E-03 | (8.41 $\pm$ 2.09)E-03 | 3233 $\pm$ 1807 |
| | 0.01 | (0.95, 0.97) | (7.28 $\pm$ 2.43)E-06 | (2.13 $\pm$ 0.44)E-03 | (1.03 $\pm$ 0.33)E-02 | 927 $\pm$ 609 |
| | | (0.97, 0.95) | (7.98 $\pm$ 3.49)E-06 | (2.23 $\pm$ 0.59)E-03 | (9.66 $\pm$ 2.29)E-03 | 1097 $\pm$ 502 |
| | | (0.9, 0.999) | (1.69 $\pm$ 0.46)E-06 | (9.84 $\pm$ 0.80)E-04 | (6.72 $\pm$ 2.97)E-03 | 2793 $\pm$ 592 |
| | **0.001** | **(0.95, 0.97)** | (2.22 $\pm$ 0.72)E-06 | (1.13 $\pm$ 0.18)E-03 | (6.34 $\pm$ 1.34)E-03 | 594 $\pm$ 115 |
| | | (0.97, 0.95) | (2.86 $\pm$ 1.10)E-06 | (1.28 $\pm$ 0.27)E-03 | (7.57 $\pm$ 2.19)E-03 | 598 $\pm$ 68 |
| | | (0.9, 0.999) | (1.83 $\pm$ 0.63)E-06 | (1.04 $\pm$ 0.11)E-03 | (6.67 $\pm$ 2.14)E-03 | 1005 $\pm$ 253 |
| **16** | 0.005 | (0.95, 0.97) | (2.81 $\pm$ 1.25)E-06 | (1.31 $\pm$ 0.33)E-03 | (6.29 $\pm$ 1.43)E-03 | 348 $\pm$ 41 |
| | | (0.97, 0.95) | (4.16 $\pm$ 2.32)E-06 | (1.60 $\pm$ 0.49)E-03 | (8.01 $\pm$ 2.07)E-03 | 381 $\pm$ 37 |
| | | (0.9, 0.999) | (1.99 $\pm$ 0.88)E-06 | (1.09 $\pm$ 0.17)E-03 | (6.87 $\pm$ 1.71)E-03 | 643 $\pm$ 139 |
| | 0.01 | (0.95, 0.97) | (5.55 $\pm$ 3.13)E-06 | (1.93 $\pm$ 0.62)E-03 | (6.87 $\pm$ 1.58)E-03 | 295 $\pm$ 29 |
| | | (0.97, 0.95) | (5.71 $\pm$ 2.31)E-06 | (1.87 $\pm$ 0.43)E-03 | (8.35 $\pm$ 1.94)E-03 | 338 $\pm$ 31 |
| | | (0.9, 0.999) | (1.34 $\pm$ 0.52)E-06 | (8.75 $\pm$ 1.23)E-04 | (8.09 $\pm$ 4.70)E-03 | 2509 $\pm$ 428 |
| | 0.001 | (0.95, 0.97) | (1.45 $\pm$ 0.67)E-06 | (9.32 $\pm$ 2.02)E-04 | (6.94 $\pm$ 2.65)E-03 | 491 $\pm$ 45 |
| | | (0.97, 0.95) | (1.64 $\pm$ 0.47)E-06 | (9.79 $\pm$ 1.54)E-04 | (7.79 $\pm$ 2.93)E-03 | 505 $\pm$ 25 |
| | | (0.9, 0.999) | (1.05 $\pm$ 0.37)E-06 | (7.89 $\pm$ 1.10)E-04 | (7.18 $\pm$ 3.89)E-03 | 860 $\pm$ 188 |
| 25 | 0.005 | (0.95, 0.97) | (2.49 $\pm$ 1.70)E-06 | (1.19 $\pm$ 0.41)E-03 | (7.33 $\pm$ 3.75)E-03 | 336 $\pm$ 52 |
| | | (0.97, 0.95) | (3.54 $\pm$ 2.90)E-06 | (1.48 $\pm$ 0.63)E-03 | (7.77 $\pm$ 3.07)E-03 | 353 $\pm$ 35 |
| | | (0.9, 0.999) | (1.08 $\pm$ 0.38)E-06 | (8.00 $\pm$ 1.11)E-04 | (6.80 $\pm$ 4.21)E-03 | 568 $\pm$ 158 |
| | 0.01 | (0.95, 0.97) | (6.16 $\pm$ 2.95)E-06 | (2.02 $\pm$ 0.53)E-03 | (8.27 $\pm$ 3.50)E-03 | 292 $\pm$ 60 |
| | | (0.97, 0.95) | (8.26 $\pm$ 4.61)E-06 | (2.30 $\pm$ 0.71)E-03 | (7.92 $\pm$ 1.69)E-03 | 319 $\pm$ 39 |

Table 6: Heat Equation with SIREN–1

| $N_{\text{hidden}}$ | LR | Betas | MSE $\pm$ SD | MAE $\pm$ SD | Max Error $\pm$ SD | Epochs $\pm$ SD |
|---|---|---|---|---|---|---|
| | | (0.9, 0.999) | (2.83 $\pm$ 1.58)E-05 | (4.04 $\pm$ 1.19)E-03 | (2.32 $\pm$ 0.90)E-02 | 9999 $\pm$ 0 |
| | 0.001 | (0.95, 0.97) | (1.11 $\pm$ 0.27)E-05 | (2.60 $\pm$ 0.34)E-03 | (1.59 $\pm$ 0.44)E-02 | 6269 $\pm$ 2156 |
| | | (0.97, 0.95) | (9.97 $\pm$ 2.77)E-06 | (2.49 $\pm$ 0.38)E-03 | (1.35 $\pm$ 0.24)E-02 | 5255 $\pm$ 1420 |
| | | (0.9, 0.999) | (1.14 $\pm$ 0.99)E-05 | (2.47 $\pm$ 0.92)E-03 | (1.34 $\pm$ 0.73)E-02 | 9444 $\pm$ 1148 |
| **6** | **0.005** | (0.95, 0.97) | (9.18 $\pm$ 2.22)E-06 | (2.39 $\pm$ 0.33)E-03 | (1.27 $\pm$ 0.30)E-02 | 3904 $\pm$ 2088 |
| | | **(0.97, 0.95)** | (8.88 $\pm$ 2.96)E-06 | (2.37 $\pm$ 0.43)E-03 | (1.12 $\pm$ 0.26)E-02 | 2658 $\pm$ 1019 |
| | | (0.9, 0.999) | (9.23 $\pm$ 3.38)E-06 | (2.41 $\pm$ 0.42)E-03 | (1.22 $\pm$ 0.44)E-02 | 8432 $\pm$ 1932 |
| | 0.01 | (0.95, 0.97) | (8.64 $\pm$ 4.76)E-06 | (2.27 $\pm$ 0.74)E-03 | (1.12 $\pm$ 0.32)E-02 | 3803 $\pm$ 1919 |
| | | (0.97, 0.95) | (1.06 $\pm$ 0.24)E-05 | (2.64 $\pm$ 0.35)E-03 | (1.17 $\pm$ 0.26)E-02 | 1938 $\pm$ 753 |
| | | (0.9, 0.999) | (2.08 $\pm$ 1.19)E-05 | (3.43 $\pm$ 1.01)E-03 | (2.26 $\pm$ 0.60)E-02 | 9879 $\pm$ 379 |
| | 0.001 | (0.95, 0.97) | (1.18 $\pm$ 0.10)E-05 | (2.67 $\pm$ 0.19)E-03 | (1.84 $\pm$ 0.35)E-02 | 6588 $\pm$ 1250 |
| | | (0.97, 0.95) | (1.16 $\pm$ 0.13)E-05 | (2.61 $\pm$ 0.27)E-03 | (1.95 $\pm$ 0.34)E-02 | 5990 $\pm$ 811 |
| | | (0.9, 0.999) | (9.12 $\pm$ 5.90)E-06 | (2.23 $\pm$ 0.74)E-03 | (1.52 $\pm$ 0.57)E-02 | 7008 $\pm$ 2507 |
| 24 | 0.005 | (0.95, 0.97) | (9.81 $\pm$ 2.20)E-06 | (2.33 $\pm$ 0.38)E-03 | (1.85 $\pm$ 0.34)E-02 | 5093 $\pm$ 1462 |
| | | (0.97, 0.95) | (1.01 $\pm$ 0.11)E-05 | (2.24 $\pm$ 0.20)E-03 | (2.06 $\pm$ 0.22)E-02 | 3288 $\pm$ 929 |
| | | (0.9, 0.999) | (1.20 $\pm$ 1.02)E-05 | (2.66 $\pm$ 1.30)E-03 | (1.27 $\pm$ 0.48)E-02 | 6689 $\pm$ 2824 |
| | 0.01 | (0.95, 0.97) | (1.07 $\pm$ 0.29)E-05 | (2.61 $\pm$ 0.55)E-03 | (1.54 $\pm$ 0.34)E-02 | 4371 $\pm$ 1796 |
| | | (0.97, 0.95) | (1.20 $\pm$ 0.56)E-05 | (2.63 $\pm$ 0.80)E-03 | (1.61 $\pm$ 0.45)E-02 | 2566 $\pm$ 802 |
| | | (0.9, 0.999) | (1.53 $\pm$ 0.99)E-05 | (2.96 $\pm$ 0.99)E-03 | (1.92 $\pm$ 0.69)E-02 | 9617 $\pm$ 743 |
| | 0.001 | (0.95, 0.97) | (1.27 $\pm$ 0.22)E-05 | (2.82 $\pm$ 0.36)E-03 | (1.83 $\pm$ 0.34)E-02 | 7915 $\pm$ 1354 |
| | | (0.97, 0.95) | (1.27 $\pm$ 0.12)E-05 | (2.79 $\pm$ 0.25)E-03 | (1.80 $\pm$ 0.41)E-02 | 6416 $\pm$ 746 |
| | | (0.9, 0.999) | (9.55 $\pm$ 4.03)E-06 | (2.43 $\pm$ 0.55)E-03 | (1.42 $\pm$ 0.46)E-02 | 7341 $\pm$ 2075 |
| 37 | 0.005 | (0.95, 0.97) | (1.13 $\pm$ 0.47)E-05 | (2.62 $\pm$ 0.74)E-03 | (1.65 $\pm$ 0.37)E-02 | 6120 $\pm$ 2289 |
| | | (0.97, 0.95) | (1.05 $\pm$ 0.36)E-05 | (2.45 $\pm$ 0.49)E-03 | (1.61 $\pm$ 0.52)E-02 | 4219 $\pm$ 1018 |
| | | (0.9, 0.999) | (1.00 $\pm$ 0.50)E-05 | (2.51 $\pm$ 0.75)E-03 | (1.31 $\pm$ 0.44)E-02 | 7143 $\pm$ 2049 |
| | 0.01 | (0.95, 0.97) | (1.10 $\pm$ 0.49)E-05 | (2.59 $\pm$ 0.77)E-03 | (1.45 $\pm$ 0.55)E-02 | 5468 $\pm$ 2671 |
| | | (0.97, 0.95) | (1.45 $\pm$ 0.57)E-05 | (3.01 $\pm$ 0.79)E-03 | (1.27 $\pm$ 0.29)E-02 | 2840 $\pm$ 1014 |

Table 7: Heat Equation with SIREN–2

| $N_{\text{hidden}}$ | LR | Betas | MSE ± SD | MAE ± SD | Max Error ± SD | Epochs ± SD |
|---|---|---|---|---|---|---|
| | | (0.9, 0.999) | (2.35 ± 1.70)E-05 | (3.65 ± 1.40)E-03 | (1.89 ± 0.98)E-02 | 9999 ± 0 |
| | **0.001** | (0.95, 0.97) | (5.87 ± 4.70)E-06 | (1.79 ± 0.80)E-03 | (1.15 ± 0.57)E-02 | 7297 ± 3557 |
| | | **(0.97, 0.95)** | (4.12 ± 2.24)E-06 | (1.55 ± 0.47)E-03 | (1.01 ± 0.32)E-02 | 6144 ± 2993 |
| | | (0.9, 0.999) | (1.73 ± 1.18)E-05 | (3.25 ± 1.47)E-03 | (1.43 ± 0.59)E-02 | 9218 ± 1647 |
| **3** | 0.005 | (0.95, 0.97) | (7.03 ± 6.13)E-06 | (1.93 ± 0.90)E-03 | (9.41 ± 3.58)E-03 | 5834 ± 3426 |
| | | (0.97, 0.95) | (5.84 ± 6.57)E-06 | (1.69 ± 0.96)E-03 | (1.11 ± 0.60)E-02 | 5149 ± 3420 |
| | | (0.9, 0.999) | (7.23 ± 20.80)E-05 | (3.82 ± 6.09)E-03 | (1.38 ± 1.44)E-02 | 9959 ± 126 |
| | 0.01 | (0.95, 0.97) | (9.54 ± 8.34)E-06 | (2.26 ± 0.86)E-03 | (1.24 ± 0.82)E-02 | 6205 ± 3561 |
| | | (0.97, 0.95) | (4.86 ± 1.75)E-06 | (1.74 ± 0.31)E-03 | (9.53 ± 3.95)E-03 | 5585 ± 3913 |
| | | (0.9, 0.999) | (1.90 ± 0.99)E-06 | (9.79 ± 1.89)E-04 | (8.73 ± 3.20)E-03 | 7476 ± 1894 |
| | 0.001 | (0.95, 0.97) | (2.27 ± 1.37)E-06 | (1.12 ± 0.33)E-03 | (7.86 ± 2.06)E-03 | 1432 ± 408 |
| | | (0.97, 0.95) | (1.83 ± 1.05)E-06 | (1.02 ± 0.29)E-03 | (6.76 ± 2.07)E-03 | 1506 ± 305 |
| | | (0.9, 0.999) | (1.99 ± 1.05)E-06 | (9.85 ± 2.31)E-04 | (1.02 ± 0.36)E-02 | 4380 ± 1700 |
| 8 | 0.005 | (0.95, 0.97) | (2.92 ± 1.37)E-06 | (1.32 ± 0.36)E-03 | (9.41 ± 3.78)E-03 | 891 ± 300 |
| | | (0.97, 0.95) | (2.97 ± 1.11)E-06 | (1.35 ± 0.27)E-03 | (8.62 ± 3.00)E-03 | 810 ± 206 |
| | | (0.9, 0.999) | (3.10 ± 1.87)E-06 | (1.29 ± 0.42)E-03 | (1.08 ± 0.59)E-02 | 2569 ± 1112 |
| | 0.01 | (0.95, 0.97) | (7.20 ± 2.06)E-06 | (2.25 ± 0.36)E-03 | (8.21 ± 1.82)E-03 | 853 ± 423 |
| | | (0.97, 0.95) | (5.43 ± 2.13)E-06 | (1.86 ± 0.38)E-03 | (9.76 ± 2.30)E-03 | 652 ± 172 |
| | | (0.9, 0.999) | (2.07 ± 0.92)E-06 | (1.03 ± 0.21)E-03 | (9.84 ± 2.88)E-03 | 7124 ± 1546 |
| | 0.001 | (0.95, 0.97) | (2.67 ± 1.22)E-06 | (1.24 ± 0.33)E-03 | (7.46 ± 1.30)E-03 | 1487 ± 398 |
| | | (0.97, 0.95) | (2.32 ± 0.72)E-06 | (1.15 ± 0.24)E-03 | (8.62 ± 3.05)E-03 | 1455 ± 281 |
| | | (0.9, 0.999) | (2.70 ± 2.47)E-06 | (1.17 ± 0.57)E-03 | (9.60 ± 2.58)E-03 | 4110 ± 1315 |
| 10 | 0.005 | (0.95, 0.97) | (4.48 ± 1.57)E-06 | (1.76 ± 0.41)E-03 | (9.67 ± 2.89)E-03 | 1051 ± 353 |
| | | (0.97, 0.95) | (4.16 ± 1.71)E-06 | (1.60 ± 0.44)E-03 | (1.01 ± 0.20)E-02 | 904 ± 235 |
| | | (0.9, 0.999) | (2.01 ± 0.84)E-06 | (1.05 ± 0.25)E-03 | (8.77 ± 2.49)E-03 | 2917 ± 902 |
| | 0.01 | (0.95, 0.97) | (1.05 ± 0.30)E-05 | (2.77 ± 0.56)E-03 | (1.11 ± 0.29)E-02 | 919 ± 143 |
| | | (0.97, 0.95) | (6.22 ± 2.93)E-06 | (2.06 ± 0.62)E-03 | (8.96 ± 1.66)E-03 | 761 ± 172 |

Table 8: Heat Equation with SIREN–3

| $N_{\text{hidden}}$ | LR | Betas | MSE ± SD | MAE ± SD | Max Error ± SD | Epochs ± SD |
|---|---|---|---|---|---|---|
| | | (0.9, 0.999) | (4.46 ± 3.58)E-05 | (4.92 ± 1.90)E-03 | (3.08 ± 1.12)E-02 | 9999 ± 0 |
| | 0.001 | (0.95, 0.97) | (1.19 ± 0.76)E-05 | (2.65 ± 0.81)E-03 | (1.64 ± 0.77)E-02 | 9729 ± 853 |
| | | (0.97, 0.95) | (5.48 ± 14.99)E-05 | (3.66 ± 4.79)E-03 | (1.89 ± 2.52)E-02 | 9080 ± 1963 |
| | | (0.9, 0.999) | (4.35 ± 7.13)E-05 | (4.51 ± 3.81)E-03 | (2.10 ± 1.13)E-02 | 9999 ± 0 |
| **2** | **0.005** | (0.95, 0.97) | (6.72 ± 17.28)E-05 | (4.41 ± 5.11)E-03 | (2.21 ± 2.62)E-02 | 9640 ± 1135 |
| | | **(0.97, 0.95)** | (6.41 ± 17.45)E-05 | (3.95 ± 5.26)E-03 | (2.00 ± 2.67)E-02 | 8225 ± 2878 |
| | | (0.9, 0.999) | (6.25 ± 15.12)E-05 | (4.28 ± 4.77)E-03 | (2.42 ± 2.62)E-02 | 9904 ± 301 |
| | 0.01 | (0.95, 0.97) | (8.64 ± 21.95)E-05 | (5.03 ± 5.85)E-03 | (2.43 ± 2.78)E-02 | 9576 ± 1339 |
| | | (0.97, 0.95) | (8.08 ± 16.63)E-05 | (5.29 ± 5.21)E-03 | (2.68 ± 3.14)E-02 | 9507 ± 1146 |
| | | (0.9, 0.999) | (3.67 ± 2.60)E-06 | (1.38 ± 0.47)E-03 | (1.17 ± 0.39)E-02 | 6447 ± 1848 |
| | 0.001 | (0.95, 0.97) | (3.80 ± 1.34)E-06 | (1.44 ± 0.30)E-03 | (1.11 ± 0.40)E-02 | 1233 ± 519 |
| | | (0.97, 0.95) | (2.49 ± 1.14)E-06 | (1.16 ± 0.22)E-03 | (9.21 ± 4.43)E-03 | 1199 ± 335 |
| | | (0.9, 0.999) | (5.09 ± 4.76)E-06 | (1.64 ± 0.88)E-03 | (1.04 ± 0.31)E-02 | 3164 ± 1713 |
| 6 | 0.005 | (0.95, 0.97) | (5.26 ± 2.47)E-06 | (1.75 ± 0.49)E-03 | (1.00 ± 0.29)E-02 | 922 ± 596 |
| | | (0.97, 0.95) | (5.24 ± 3.08)E-06 | (1.83 ± 0.57)E-03 | (9.33 ± 1.94)E-03 | 929 ± 550 |
| | | (0.9, 0.999) | (4.29 ± 3.05)E-06 | (1.52 ± 0.68)E-03 | (1.05 ± 0.33)E-02 | 2423 ± 1488 |
| | 0.01 | (0.95, 0.97) | (9.77 ± 4.94)E-06 | (2.52 ± 0.63)E-03 | (1.08 ± 0.35)E-02 | 884 ± 623 |
| | | (0.97, 0.95) | (1.09 ± 0.72)E-05 | (2.72 ± 1.03)E-03 | (1.11 ± 0.24)E-02 | 848 ± 627 |
| | | (0.9, 0.999) | (3.85 ± 1.36)E-06 | (1.48 ± 0.31)E-03 | (1.18 ± 0.47)E-02 | 7256 ± 1718 |
| | 0.001 | (0.95, 0.97) | (4.17 ± 1.72)E-06 | (1.50 ± 0.40)E-03 | (1.27 ± 0.24)E-02 | 1415 ± 621 |
| | | (0.97, 0.95) | (3.31 ± 0.89)E-06 | (1.34 ± 0.25)E-03 | (1.22 ± 0.27)E-02 | 1323 ± 242 |
| | | (0.9, 0.999) | (2.77 ± 1.73)E-06 | (1.21 ± 0.37)E-03 | (9.68 ± 3.06)E-03 | 3584 ± 1547 |
| 7 | 0.005 | (0.95, 0.97) | (7.23 ± 3.91)E-06 | (2.12 ± 0.75)E-03 | (1.28 ± 0.36)E-02 | 1049 ± 436 |
| | | (0.97, 0.95) | (6.76 ± 4.71)E-06 | (2.05 ± 0.92)E-03 | (1.05 ± 0.40)E-02 | 1072 ± 364 |
| | | (0.9, 0.999) | (3.94 ± 1.42)E-06 | (1.49 ± 0.32)E-03 | (1.11 ± 0.39)E-02 | 2740 ± 1005 |
| | 0.01 | (0.95, 0.97) | (1.50 ± 0.45)E-05 | (3.28 ± 0.65)E-03 | (1.38 ± 0.42)E-02 | 1201 ± 734 |
| | | (0.97, 0.95) | (1.23 ± 0.82)E-05 | (2.84 ± 1.08)E-03 | (1.16 ± 0.29)E-02 | 900 ± 694 |

Table 9: Heat Equation with Vanilla Relu

| $N_{\text{hidden}}$ | LR | Betas | MSE $\pm$ SD | MAE $\pm$ SD | Max Error $\pm$ SD | Epochs $\pm$ SD |
|---|---|---|---|---|---|---|
| **2** | 0.001 | (0.9, 0.999) | $(7.14 \pm 1.11)$E-02 | $(2.11 \pm 0.11)$E-01 | $(7.00 \pm 0.74)$E-01 | $10000 \pm 0$ |
| | | (0.95, 0.97) | $(7.17 \pm 1.14)$E-02 | $(2.13 \pm 0.13)$E-01 | $(6.84 \pm 0.61)$E-01 | $10000 \pm 0$ |
| | | (0.97, 0.95) | $(6.27 \pm 0.88)$E-02 | $(2.02 \pm 0.15)$E-01 | $(6.58 \pm 0.51)$E-01 | $10000 \pm 0$ |
| | **0.005** | (0.9, 0.999) | $(6.92 \pm 1.12)$E-02 | $(2.09 \pm 0.11)$E-01 | $(6.84 \pm 0.61)$E-01 | $10000 \pm 0$ |
| | | (0.95, 0.97) | $(6.67 \pm 0.88)$E-02 | $(2.08 \pm 0.11)$E-01 | $(6.67 \pm 0.18)$E-01 | $10000 \pm 0$ |
| | | **(0.97, 0.95)** | $(6.42 \pm 1.07)$E-02 | $(2.03 \pm 0.16)$E-01 | $(6.49 \pm 0.51)$E-01 | $10000 \pm 0$ |
| | 0.01 | (0.9, 0.999) | $(6.91 \pm 1.10)$E-02 | $(2.09 \pm 0.12)$E-01 | $(6.82 \pm 0.62)$E-01 | $10000 \pm 0$ |
| | | (0.95, 0.97) | $(6.43 \pm 1.09)$E-02 | $(2.03 \pm 0.16)$E-01 | $(6.50 \pm 0.50)$E-01 | $10000 \pm 0$ |
| | | (0.97, 0.95) | $(6.44 \pm 1.10)$E-02 | $(2.03 \pm 0.16)$E-01 | $(6.50 \pm 0.51)$E-01 | $10000 \pm 0$ |
| 6 | 0.001 | (0.9, 0.999) | $(8.03 \pm 0.65)$E-02 | $(2.26 \pm 0.09)$E-01 | $(6.51 \pm 0.16)$E-01 | $4973 \pm 3984$ |
| | | (0.95, 0.97) | $(7.80 \pm 0.83)$E-02 | $(2.23 \pm 0.10)$E-01 | $(6.52 \pm 0.18)$E-01 | $5508 \pm 4738$ |
| | | (0.97, 0.95) | $(8.17 \pm 0.15)$E-02 | $(2.25 \pm 0.07)$E-01 | $(6.60 \pm 0.53)$E-01 | $4297 \pm 4030$ |
| | 0.005 | (0.9, 0.999) | $(7.84 \pm 0.86)$E-02 | $(2.23 \pm 0.11)$E-01 | $(6.56 \pm 0.14)$E-01 | $6297 \pm 4786$ |
| | | (0.95, 0.97) | $(8.04 \pm 0.62)$E-02 | $(2.25 \pm 0.07)$E-01 | $(6.59 \pm 0.24)$E-01 | $6181 \pm 4929$ |
| | | (0.97, 0.95) | $(8.03 \pm 0.63)$E-02 | $(2.24 \pm 0.07)$E-01 | $(6.59 \pm 0.26)$E-01 | $6852 \pm 4256$ |
| | 0.01 | (0.9, 0.999) | $(7.68 \pm 0.83)$E-02 | $(2.20 \pm 0.13)$E-01 | $(6.61 \pm 0.43)$E-01 | $4994 \pm 4755$ |
| | | (0.95, 0.97) | $(8.19 \pm 0.17)$E-02 | $(2.26 \pm 0.07)$E-01 | $(6.68 \pm 0.52)$E-01 | $4958 \pm 4232$ |
| | | (0.97, 0.95) | $(8.28 \pm 0.26)$E-02 | $(2.25 \pm 0.07)$E-01 | $(6.91 \pm 0.73)$E-01 | $7183 \pm 4536$ |
| 7 | 0.001 | (0.9, 0.999) | $(8.23 \pm 0.07)$E-02 | $(2.29 \pm 0.02)$E-01 | $(6.46 \pm 0.15)$E-01 | $5614 \pm 3938$ |
| | | (0.95, 0.97) | $(8.23 \pm 0.08)$E-02 | $(2.29 \pm 0.01)$E-01 | $(6.42 \pm 0.16)$E-01 | $947 \pm 329$ |
| | | (0.97, 0.95) | $(8.20 \pm 0.09)$E-02 | $(2.28 \pm 0.01)$E-01 | $(6.43 \pm 0.16)$E-01 | $2130 \pm 2790$ |
| | 0.005 | (0.9, 0.999) | $(8.21 \pm 0.08)$E-02 | $(2.28 \pm 0.02)$E-01 | $(6.44 \pm 0.19)$E-01 | $3628 \pm 3880$ |
| | | (0.95, 0.97) | $(8.24 \pm 0.14)$E-02 | $(2.29 \pm 0.02)$E-01 | $(6.42 \pm 0.18)$E-01 | $1921 \pm 3169$ |
| | | (0.97, 0.95) | $(7.97 \pm 0.72)$E-02 | $(2.24 \pm 0.11)$E-01 | $(6.63 \pm 0.49)$E-01 | $4461 \pm 4786$ |
| | 0.01 | (0.9, 0.999) | $(7.98 \pm 0.70)$E-02 | $(2.25 \pm 0.08)$E-01 | $(6.39 \pm 0.11)$E-01 | $2833 \pm 3213$ |
| | | (0.95, 0.97) | $(8.13 \pm 0.17)$E-02 | $(2.26 \pm 0.05)$E-01 | $(6.49 \pm 0.19)$E-01 | $3401 \pm 4438$ |
| | | (0.97, 0.95) | $(8.20 \pm 0.18)$E-02 | $(2.26 \pm 0.07)$E-01 | $(6.60 \pm 0.59)$E-01 | $4811 \pm 4720$ |

Table 10: Heat Equation with Vanilla Tan

| $N_{\text{hidden}}$ | LR | Betas | MSE $\pm$ SD | MAE $\pm$ SD | Max Error $\pm$ SD | Epochs $\pm$ SD |
|---|---|---|---|---|---|---|
| 2 | 0.001 | (0.9, 0.999) | $(8.63 \pm 20.28)$E-03 | $(5.32 \pm 5.69)$E-02 | $(2.14 \pm 1.62)$E-01 | $9999 \pm 0$ |
| | | (0.95, 0.97) | $(9.75 \pm 11.43)$E-04 | $(2.06 \pm 1.41)$E-02 | $(9.93 \pm 7.48)$E-02 | $9999 \pm 0$ |
| | | (0.97, 0.95) | $(1.09 \pm 1.20)$E-03 | $(2.19 \pm 1.43)$E-02 | $(1.07 \pm 0.75)$E-01 | $9999 \pm 0$ |
| | 0.005 | (0.9, 0.999) | $(8.79 \pm 20.22)$E-03 | $(5.44 \pm 5.64)$E-02 | $(2.22 \pm 1.59)$E-01 | $9999 \pm 0$ |
| | | (0.95, 0.97) | $(1.79 \pm 1.48)$E-03 | $(3.00 \pm 1.63)$E-02 | $(1.43 \pm 0.77)$E-01 | $9999 \pm 0$ |
| | | (0.97, 0.95) | $(1.25 \pm 1.16)$E-03 | $(2.46 \pm 1.23)$E-02 | $(1.25 \pm 0.70)$E-01 | $9999 \pm 0$ |
| | 0.01 | (0.9, 0.999) | $(2.05 \pm 1.45)$E-03 | $(3.29 \pm 1.47)$E-02 | $(1.54 \pm 0.76)$E-01 | $9999 \pm 0$ |
| | | (0.95, 0.97) | $(1.22 \pm 1.49)$E-03 | $(2.33 \pm 1.69)$E-02 | $(1.12 \pm 0.73)$E-01 | $9999 \pm 0$ |
| | | (0.97, 0.95) | $(7.94 \pm 10.03)$E-04 | $(1.85 \pm 1.24)$E-02 | $(1.03 \pm 0.66)$E-01 | $9999 \pm 0$ |
| **6** | 0.001 | (0.9, 0.999) | $(5.81 \pm 2.69)$E-06 | $(1.76 \pm 0.49)$E-03 | $(1.34 \pm 0.44)$E-02 | $5322 \pm 1494$ |
| | | (0.95, 0.97) | $(5.86 \pm 3.11)$E-06 | $(1.75 \pm 0.55)$E-03 | $(1.47 \pm 0.49)$E-02 | $1628 \pm 697$ |
| | | (0.97, 0.95) | $(6.80 \pm 2.34)$E-06 | $(1.95 \pm 0.48)$E-03 | $(1.52 \pm 0.41)$E-02 | $1733 \pm 949$ |
| | 0.005 | (0.9, 0.999) | $(9.86 \pm 11.34)$E-06 | $(2.32 \pm 1.40)$E-03 | $(1.40 \pm 0.19)$E-02 | $3072 \pm 1540$ |
| | | (0.95, 0.97) | $(9.88 \pm 6.36)$E-06 | $(2.47 \pm 1.07)$E-03 | $(1.46 \pm 0.36)$E-02 | $1410 \pm 884$ |
| | | (0.97, 0.95) | $(8.96 \pm 3.56)$E-06 | $(2.18 \pm 0.64)$E-03 | $(1.66 \pm 0.37)$E-02 | $1299 \pm 1080$ |
| | **0.01** | **(0.9, 0.999)** | $(5.21 \pm 2.95)$E-06 | $(1.64 \pm 0.60)$E-03 | $(1.35 \pm 0.27)$E-02 | $2110 \pm 1195$ |
| | | (0.95, 0.97) | $(1.67 \pm 0.92)$E-05 | $(3.41 \pm 1.13)$E-03 | $(1.64 \pm 0.54)$E-02 | $1089 \pm 589$ |
| | | (0.97, 0.95) | $(2.57 \pm 1.91)$E-05 | $(4.17 \pm 1.76)$E-03 | $(1.71 \pm 0.19)$E-02 | $921 \pm 435$ |
| 7 | 0.001 | (0.9, 0.999) | $(5.70 \pm 1.72)$E-06 | $(1.80 \pm 0.31)$E-03 | $(1.38 \pm 0.36)$E-02 | $4635 \pm 2253$ |
| | | (0.95, 0.97) | $(6.35 \pm 2.42)$E-06 | $(1.82 \pm 0.44)$E-03 | $(1.53 \pm 0.35)$E-02 | $1539 \pm 726$ |
| | | (0.97, 0.95) | $(8.15 \pm 2.54)$E-06 | $(2.12 \pm 0.44)$E-03 | $(1.66 \pm 0.28)$E-02 | $1340 \pm 202$ |
| | 0.005 | (0.9, 0.999) | $(5.22 \pm 1.87)$E-06 | $(1.66 \pm 0.34)$E-03 | $(1.40 \pm 0.36)$E-02 | $1771 \pm 463$ |
| | | (0.95, 0.97) | $(1.07 \pm 0.50)$E-05 | $(2.59 \pm 0.91)$E-03 | $(1.57 \pm 0.42)$E-02 | $797 \pm 182$ |
| | | (0.97, 0.95) | $(1.31 \pm 0.47)$E-05 | $(2.93 \pm 0.87)$E-03 | $(1.70 \pm 0.24)$E-02 | $717 \pm 143$ |
| | 0.01 | (0.9, 0.999) | $(4.84 \pm 1.74)$E-06 | $(1.58 \pm 0.33)$E-03 | $(1.41 \pm 0.31)$E-02 | $1565 \pm 763$ |
| | | (0.95, 0.97) | $(1.74 \pm 0.71)$E-05 | $(3.54 \pm 1.02)$E-03 | $(1.57 \pm 0.38)$E-02 | $745 \pm 394$ |
| | | (0.97, 0.95) | $(2.13 \pm 0.59)$E-05 | $(3.85 \pm 0.77)$E-03 | $(1.81 \pm 0.38)$E-02 | $597 \pm 157$ |

Table 11: Poisson Equation with FLM

| $N$ | LR | Betas | MSE $\pm$ SD | MAE $\pm$ SD | Max Error $\pm$ SD | Epochs $\pm$ SD |
|---|---|---|---|---|---|---|
| **4** | **0.001** | (0.9, 0.999) | (9.38 $\pm$ 1.46)E-04 | (2.88 $\pm$ 0.20)E-02 | (5.59 $\pm$ 0.67)E-02 | 10000 $\pm$ 0 |
| | | **(0.95, 0.97)** | (1.71 $\pm$ 1.32)E-05 | (3.23 $\pm$ 1.45)E-03 | (1.30 $\pm$ 0.22)E-02 | 3817 $\pm$ 2113 |
| | | (0.97, 0.95) | (1.16 $\pm$ 0.35)E-05 | (2.55 $\pm$ 0.43)E-03 | (1.52 $\pm$ 0.25)E-02 | 3307 $\pm$ 619 |
| | 0.005 | (0.9, 0.999) | (6.75 $\pm$ 2.99)E-05 | (7.50 $\pm$ 1.75)E-03 | (1.67 $\pm$ 0.48)E-02 | 8981 $\pm$ 1277 |
| | | (0.95, 0.97) | (2.10 $\pm$ 0.49)E-05 | (3.71 $\pm$ 0.64)E-03 | (1.44 $\pm$ 0.30)E-02 | 814 $\pm$ 93 |
| | | (0.97, 0.95) | (2.47 $\pm$ 0.72)E-05 | (3.85 $\pm$ 0.66)E-03 | (1.50 $\pm$ 0.14)E-02 | 874 $\pm$ 49 |
| | 0.01 | (0.9, 0.999) | (4.37 $\pm$ 1.82)E-05 | (6.04 $\pm$ 1.41)E-03 | (1.28 $\pm$ 0.31)E-02 | 6494 $\pm$ 2428 |
| | | (0.95, 0.97) | (1.86 $\pm$ 0.50)E-05 | (3.34 $\pm$ 0.61)E-03 | (1.43 $\pm$ 0.17)E-02 | 630 $\pm$ 60 |
| | | (0.97, 0.95) | (2.21 $\pm$ 0.81)E-05 | (3.67 $\pm$ 0.92)E-03 | (1.33 $\pm$ 0.33)E-02 | 667 $\pm$ 24 |
| 16 | 0.001 | (0.9, 0.999) | (9.28 $\pm$ 6.97)E-06 | (2.27 $\pm$ 0.87)E-03 | (1.34 $\pm$ 0.56)E-02 | 9988 $\pm$ 32 |
| | | (0.95, 0.97) | (7.76 $\pm$ 6.19)E-06 | (2.03 $\pm$ 1.09)E-03 | (1.14 $\pm$ 0.47)E-02 | 1785 $\pm$ 528 |
| | | (0.97, 0.95) | (7.96 $\pm$ 5.61)E-06 | (2.15 $\pm$ 0.89)E-03 | (1.12 $\pm$ 0.43)E-02 | 1279 $\pm$ 229 |
| | 0.005 | (0.9, 0.999) | (4.53 $\pm$ 2.24)E-06 | (1.55 $\pm$ 0.35)E-03 | (9.67 $\pm$ 2.88)E-03 | 5865 $\pm$ 365 |
| | | (0.95, 0.97) | (3.67 $\pm$ 2.58)E-06 | (1.36 $\pm$ 0.46)E-03 | (7.35 $\pm$ 1.80)E-03 | 1069 $\pm$ 154 |
| | | (0.97, 0.95) | (4.93 $\pm$ 1.22)E-06 | (1.77 $\pm$ 0.28)E-03 | (8.12 $\pm$ 1.68)E-03 | 957 $\pm$ 181 |
| | 0.01 | (0.9, 0.999) | (4.90 $\pm$ 2.29)E-06 | (1.63 $\pm$ 0.34)E-03 | (9.72 $\pm$ 3.02)E-03 | 4471 $\pm$ 363 |
| | | (0.95, 0.97) | (6.38 $\pm$ 3.11)E-06 | (2.04 $\pm$ 0.50)E-03 | (6.95 $\pm$ 1.81)E-03 | 1129 $\pm$ 151 |
| | | (0.97, 0.95) | (8.11 $\pm$ 4.31)E-06 | (2.34 $\pm$ 0.79)E-03 | (8.72 $\pm$ 2.76)E-03 | 859 $\pm$ 127 |
| 25 | 0.001 | (0.9, 0.999) | (7.35 $\pm$ 2.01)E-06 | (2.08 $\pm$ 0.31)E-03 | (1.23 $\pm$ 0.25)E-02 | 8751 $\pm$ 672 |
| | | (0.95, 0.97) | (5.18 $\pm$ 3.52)E-06 | (1.58 $\pm$ 0.52)E-03 | (9.56 $\pm$ 3.70)E-03 | 1412 $\pm$ 314 |
| | | (0.97, 0.95) | (6.45 $\pm$ 3.02)E-06 | (1.78 $\pm$ 0.46)E-03 | (1.20 $\pm$ 0.32)E-02 | 1508 $\pm$ 429 |
| | 0.005 | (0.9, 0.999) | (5.10 $\pm$ 1.93)E-06 | (1.68 $\pm$ 0.23)E-03 | (1.02 $\pm$ 0.16)E-02 | 4796 $\pm$ 532 |
| | | (0.95, 0.97) | (5.08 $\pm$ 2.47)E-06 | (1.71 $\pm$ 0.50)E-03 | (8.31 $\pm$ 2.58)E-03 | 1491 $\pm$ 493 |
| | | (0.97, 0.95) | (5.54 $\pm$ 2.89)E-06 | (1.74 $\pm$ 0.46)E-03 | (9.85 $\pm$ 3.09)E-03 | 1632 $\pm$ 632 |
| | 0.01 | (0.9, 0.999) | (4.89 $\pm$ 2.28)E-06 | (1.65 $\pm$ 0.38)E-03 | (9.83 $\pm$ 2.67)E-03 | 3632 $\pm$ 337 |
| | | (0.95, 0.97) | (1.26 $\pm$ 0.80)E-05 | (2.91 $\pm$ 0.94)E-03 | (8.42 $\pm$ 1.68)E-03 | 2132 $\pm$ 626 |
| | | (0.97, 0.95) | (8.97 $\pm$ 5.42)E-06 | (2.40 $\pm$ 0.86)E-03 | (9.77 $\pm$ 3.26)E-03 | 2039 $\pm$ 780 |

Table 12: Poisson Equation with SIREN–1

| $N_{\text{hidden}}$ | LR | Betas | MSE $\pm$ SD | MAE $\pm$ SD | Max Error $\pm$ SD | Epochs $\pm$ SD |
|---|---|---|---|---|---|---|
| **6** | **0.001** | (0.9, 0.999) | (2.07 $\pm$ 1.77)E-03 | (3.55 $\pm$ 1.62)E-02 | (1.70 $\pm$ 0.82)E-01 | 10000 $\pm$ 0 |
| | | (0.95, 0.97) | (4.33 $\pm$ 10.44)E-05 | (3.57 $\pm$ 4.12)E-03 | (1.79 $\pm$ 1.85)E-02 | 8115 $\pm$ 2465 |
| | | **(0.97, 0.95)** | (9.23 $\pm$ 10.75)E-06 | (1.99 $\pm$ 1.26)E-03 | (1.01 $\pm$ 0.60)E-02 | 6335 $\pm$ 2371 |
| | 0.005 | (0.9, 0.999) | (7.71 $\pm$ 6.69)E-05 | (6.62 $\pm$ 3.37)E-03 | (3.30 $\pm$ 1.38)E-02 | 10000 $\pm$ 0 |
| | | (0.95, 0.97) | (2.14 $\pm$ 3.11)E-05 | (2.96 $\pm$ 2.31)E-03 | (1.49 $\pm$ 1.12)E-02 | 8946 $\pm$ 1519 |
| | | (0.97, 0.95) | (9.50 $\pm$ 4.94)E-06 | (2.37 $\pm$ 0.64)E-03 | (1.12 $\pm$ 0.34)E-02 | 2573 $\pm$ 1848 |
| | 0.01 | (0.9, 0.999) | (7.35 $\pm$ 5.51)E-05 | (6.37 $\pm$ 3.11)E-03 | (3.27 $\pm$ 1.51)E-02 | 9996 $\pm$ 14 |
| | | (0.95, 0.97) | (1.04 $\pm$ 1.14)E-05 | (2.25 $\pm$ 1.27)E-03 | (1.05 $\pm$ 0.71)E-02 | 8550 $\pm$ 2006 |
| | | (0.97, 0.95) | (1.25 $\pm$ 0.65)E-05 | (2.87 $\pm$ 0.96)E-03 | (1.09 $\pm$ 0.31)E-02 | 1553 $\pm$ 861 |
| 24 | 0.001 | (0.9, 0.999) | (2.29 $\pm$ 1.27)E-04 | (1.23 $\pm$ 0.34)E-02 | (6.26 $\pm$ 1.95)E-02 | 10000 $\pm$ 0 |
| | | (0.95, 0.97) | (9.69 $\pm$ 6.54)E-06 | (2.21 $\pm$ 1.00)E-03 | (1.16 $\pm$ 0.42)E-02 | 10000 $\pm$ 0 |
| | | (0.97, 0.95) | (1.95 $\pm$ 1.33)E-06 | (9.99 $\pm$ 4.43)E-04 | (5.38 $\pm$ 1.72)E-03 | 10000 $\pm$ 0 |
| | 0.005 | (0.9, 0.999) | (3.66 $\pm$ 3.81)E-05 | (4.53 $\pm$ 2.72)E-03 | (1.65 $\pm$ 0.60)E-02 | 10000 $\pm$ 0 |
| | | (0.95, 0.97) | (7.38 $\pm$ 3.09)E-06 | (2.18 $\pm$ 0.53)E-03 | (8.38 $\pm$ 2.07)E-03 | 10000 $\pm$ 0 |
| | | (0.97, 0.95) | (2.26 $\pm$ 2.04)E-06 | (1.09 $\pm$ 0.40)E-03 | (4.92 $\pm$ 2.25)E-03 | 10000 $\pm$ 0 |
| | 0.01 | (0.9, 0.999) | (4.15 $\pm$ 4.58)E-05 | (5.07 $\pm$ 2.80)E-03 | (1.74 $\pm$ 0.50)E-02 | 10000 $\pm$ 0 |
| | | (0.95, 0.97) | (1.00 $\pm$ 0.59)E-05 | (2.51 $\pm$ 1.02)E-03 | (9.44 $\pm$ 1.78)E-03 | 10000 $\pm$ 0 |
| | | (0.97, 0.95) | (5.09 $\pm$ 3.03)E-06 | (1.87 $\pm$ 0.68)E-03 | (5.29 $\pm$ 1.43)E-03 | 9286 $\pm$ 1507 |
| 37 | 0.001 | (0.9, 0.999) | (1.30 $\pm$ 0.45)E-04 | (9.53 $\pm$ 1.81)E-03 | (4.63 $\pm$ 1.03)E-02 | 10000 $\pm$ 0 |
| | | (0.95, 0.97) | (1.03 $\pm$ 0.92)E-05 | (2.44 $\pm$ 1.30)E-03 | (1.30 $\pm$ 0.56)E-02 | 10000 $\pm$ 0 |
| | | (0.97, 0.95) | (2.80 $\pm$ 3.14)E-06 | (1.13 $\pm$ 0.82)E-03 | (5.86 $\pm$ 4.73)E-03 | 10000 $\pm$ 0 |
| | 0.005 | (0.9, 0.999) | (2.69 $\pm$ 1.60)E-05 | (3.94 $\pm$ 1.77)E-03 | (1.89 $\pm$ 0.77)E-02 | 10000 $\pm$ 0 |
| | | (0.95, 0.97) | (6.45 $\pm$ 3.84)E-06 | (2.06 $\pm$ 0.82)E-03 | (6.91 $\pm$ 1.62)E-03 | 10000 $\pm$ 0 |
| | | (0.97, 0.95) | (2.40 $\pm$ 1.29)E-06 | (1.23 $\pm$ 0.40)E-03 | (5.01 $\pm$ 1.74)E-03 | 10000 $\pm$ 0 |
| | 0.01 | (0.9, 0.999) | (1.99 $\pm$ 1.44)E-05 | (3.44 $\pm$ 1.54)E-03 | (1.65 $\pm$ 0.48)E-02 | 10000 $\pm$ 0 |
| | | (0.95, 0.97) | (1.86 $\pm$ 0.88)E-05 | (3.67 $\pm$ 1.09)E-03 | (9.49 $\pm$ 2.68)E-03 | 10000 $\pm$ 0 |
| | | (0.97, 0.95) | (1.15 $\pm$ 1.24)E-05 | (2.77 $\pm$ 1.50)E-03 | (6.54 $\pm$ 2.96)E-03 | 9618 $\pm$ 808 |

Table 13: Poisson Equation with SIREN–2

| $N_{\text{hidden}}$ | LR | Betas | MSE $\pm$ SD | MAE $\pm$ SD | Max Error $\pm$ SD | Epochs $\pm$ SD |
|---|---|---|---|---|---|---|
| | | (0.9, 0.999) | $(2.00 \pm 1.53)$E-03 | $(3.20 \pm 1.39)$E-02 | $(1.58 \pm 0.78)$E-01 | $10000 \pm 0$ |
| | 0.001 | (0.95, 0.97) | $(2.56 \pm 2.46)$E-04 | $(1.13 \pm 0.57)$E-02 | $(5.68 \pm 2.89)$E-02 | $10000 \pm 0$ |
| | | (0.97, 0.95) | $(2.84 \pm 2.30)$E-04 | $(1.27 \pm 0.45)$E-02 | $(5.78 \pm 1.92)$E-02 | $10000 \pm 0$ |
| | | (0.9, 0.999) | $(6.95 \pm 6.00)$E-04 | $(1.94 \pm 0.72)$E-02 | $(8.87 \pm 3.51)$E-02 | $10000 \pm 0$ |
| **3** | **0.005** | (0.95, 0.97) | $(2.48 \pm 2.81)$E-04 | $(1.10 \pm 0.60)$E-02 | $(5.64 \pm 2.68)$E-02 | $10000 \pm 0$ |
| | | **(0.97, 0.95)** | $(1.63 \pm 0.66)$E-04 | $(9.78 \pm 2.40)$E-03 | $(4.92 \pm 1.53)$E-02 | $10000 \pm 0$ |
| | | (0.9, 0.999) | $(6.84 \pm 11.74)$E-04 | $(1.74 \pm 1.39)$E-02 | $(7.27 \pm 4.92)$E-02 | $10000 \pm 0$ |
| | 0.01 | (0.95, 0.97) | $(1.78 \pm 0.81)$E-04 | $(1.05 \pm 0.28)$E-02 | $(5.04 \pm 1.76)$E-02 | $10000 \pm 0$ |
| | | (0.97, 0.95) | $(2.89 \pm 2.75)$E-04 | $(1.20 \pm 0.70)$E-02 | $(5.46 \pm 3.08)$E-02 | $10000 \pm 0$ |
| | | (0.9, 0.999) | $(1.61 \pm 0.90)$E-04 | $(9.70 \pm 3.16)$E-03 | $(5.49 \pm 2.05)$E-02 | $10000 \pm 0$ |
| | 0.001 | (0.95, 0.97) | $(9.87 \pm 6.17)$E-05 | $(8.14 \pm 2.69)$E-03 | $(4.29 \pm 1.01)$E-02 | $10000 \pm 0$ |
| | | (0.97, 0.95) | $(9.33 \pm 5.82)$E-05 | $(7.86 \pm 2.92)$E-03 | $(3.94 \pm 1.28)$E-02 | $9818 \pm 575$ |
| | | (0.9, 0.999) | $(1.44 \pm 0.83)$E-04 | $(9.61 \pm 2.97)$E-03 | $(4.97 \pm 1.49)$E-02 | $10000 \pm 0$ |
| 8 | 0.005 | (0.95, 0.97) | $(1.08 \pm 0.51)$E-04 | $(8.58 \pm 2.22)$E-03 | $(4.51 \pm 1.11)$E-02 | $10000 \pm 0$ |
| | | (0.97, 0.95) | $(1.04 \pm 0.50)$E-04 | $(8.50 \pm 2.44)$E-03 | $(4.16 \pm 1.32)$E-02 | $10000 \pm 0$ |
| | | (0.9, 0.999) | $(1.19 \pm 1.22)$E-04 | $(8.21 \pm 4.91)$E-03 | $(3.63 \pm 1.46)$E-02 | $10000 \pm 0$ |
| | 0.01 | (0.95, 0.97) | $(1.09 \pm 0.52)$E-04 | $(8.50 \pm 2.69)$E-03 | $(4.28 \pm 1.59)$E-02 | $10000 \pm 0$ |
| | | (0.97, 0.95) | $(9.18 \pm 6.45)$E-05 | $(7.62 \pm 3.64)$E-03 | $(3.56 \pm 1.49)$E-02 | $10000 \pm 0$ |
| | | (0.9, 0.999) | $(1.46 \pm 0.73)$E-04 | $(9.75 \pm 2.77)$E-03 | $(5.00 \pm 0.97)$E-02 | $10000 \pm 0$ |
| | 0.001 | (0.95, 0.97) | $(1.12 \pm 0.68)$E-04 | $(8.61 \pm 3.04)$E-03 | $(4.55 \pm 1.04)$E-02 | $10000 \pm 0$ |
| | | (0.97, 0.95) | $(8.47 \pm 4.48)$E-05 | $(7.48 \pm 2.70)$E-03 | $(4.09 \pm 0.74)$E-02 | $10000 \pm 0$ |
| | | (0.9, 0.999) | $(1.77 \pm 0.98)$E-04 | $(1.07 \pm 0.40)$E-02 | $(5.54 \pm 1.35)$E-02 | $10000 \pm 0$ |
| 10 | 0.005 | (0.95, 0.97) | $(9.95 \pm 6.78)$E-05 | $(7.94 \pm 3.44)$E-03 | $(4.33 \pm 1.20)$E-02 | $10000 \pm 0$ |
| | | (0.97, 0.95) | $(7.46 \pm 5.48)$E-05 | $(6.91 \pm 3.05)$E-03 | $(3.59 \pm 0.91)$E-02 | $10000 \pm 0$ |
| | | (0.9, 0.999) | $(1.29 \pm 1.49)$E-04 | $(7.75 \pm 4.85)$E-03 | $(3.92 \pm 1.91)$E-02 | $10000 \pm 0$ |
| | 0.01 | (0.95, 0.97) | $(8.04 \pm 5.90)$E-05 | $(7.00 \pm 2.77)$E-03 | $(3.65 \pm 1.40)$E-02 | $10000 \pm 0$ |
| | | (0.97, 0.95) | $(5.53 \pm 5.36)$E-05 | $(5.65 \pm 3.26)$E-03 | $(2.91 \pm 1.22)$E-02 | $9718 \pm 891$ |

Table 14: Poisson Equation with Vanilla Relu

| $N_{\text{hidden}}$ | LR | Betas | MSE $\pm$ SD | MAE $\pm$ SD | Max Error $\pm$ SD | Epochs $\pm$ SD |
|---|---|---|---|---|---|---|
| | | **(0.9, 0.999)** | $(2.45 \pm 0.00)$E-01 | $(3.97 \pm 0.00)$E-01 | $(1.00 \pm 0.00)$E+00 | $10000 \pm 0$ |
| | **0.001** | (0.95, 0.97) | $(2.45 \pm 0.00)$E-01 | $(3.97 \pm 0.00)$E-01 | $(1.00 \pm 0.00)$E+00 | $10000 \pm 0$ |
| | | (0.97, 0.95) | $(2.45 \pm 0.00)$E-01 | $(3.97 \pm 0.00)$E-01 | $(1.00 \pm 0.00)$E+00 | $10000 \pm 0$ |
| | | (0.9, 0.999) | $(2.45 \pm 0.00)$E-01 | $(3.97 \pm 0.00)$E-01 | $(1.00 \pm 0.00)$E+00 | $10000 \pm 0$ |
| **2** | 0.005 | (0.95, 0.97) | $(2.45 \pm 0.00)$E-01 | $(3.97 \pm 0.00)$E-01 | $(1.00 \pm 0.00)$E+00 | $10000 \pm 0$ |
| | | (0.97, 0.95) | $(2.45 \pm 0.00)$E-01 | $(3.97 \pm 0.00)$E-01 | $(1.00 \pm 0.00)$E+00 | $10000 \pm 0$ |
| | | (0.9, 0.999) | $(2.45 \pm 0.00)$E-01 | $(3.97 \pm 0.00)$E-01 | $(1.00 \pm 0.00)$E+00 | $10000 \pm 0$ |
| | 0.01 | (0.95, 0.97) | $(2.45 \pm 0.00)$E-01 | $(3.97 \pm 0.00)$E-01 | $(1.00 \pm 0.00)$E+00 | $10000 \pm 0$ |
| | | (0.97, 0.95) | $(2.45 \pm 0.00)$E-01 | $(3.97 \pm 0.00)$E-01 | $(1.00 \pm 0.00)$E+00 | $10000 \pm 0$ |
| | | (0.9, 0.999) | $(2.45 \pm 0.01)$E-01 | $(3.97 \pm 0.00)$E-01 | $(1.00 \pm 0.01)$E+00 | $10000 \pm 0$ |
| | 0.001 | (0.95, 0.97) | $(2.45 \pm 0.06)$E-01 | $(3.97 \pm 0.05)$E-01 | $(1.00 \pm 0.01)$E+00 | $10000 \pm 0$ |
| | | (0.97, 0.95) | $(2.45 \pm 0.06)$E-01 | $(3.97 \pm 0.05)$E-01 | $(9.99 \pm 0.11)$E-01 | $10000 \pm 0$ |
| | | (0.9, 0.999) | $(2.45 \pm 0.06)$E-01 | $(3.97 \pm 0.05)$E-01 | $(9.99 \pm 0.13)$E-01 | $10000 \pm 0$ |
| 6 | 0.005 | (0.95, 0.97) | $(2.45 \pm 0.06)$E-01 | $(3.97 \pm 0.05)$E-01 | $(9.99 \pm 0.09)$E-01 | $10000 \pm 0$ |
| | | (0.97, 0.95) | $(2.45 \pm 0.14)$E-01 | $(3.97 \pm 0.12)$E-01 | $(9.99 \pm 0.36)$E-01 | $10000 \pm 0$ |
| | | (0.9, 0.999) | $(2.45 \pm 0.06)$E-01 | $(3.97 \pm 0.07)$E-01 | $(1.00 \pm 0.07)$E+00 | $10000 \pm 0$ |
| | 0.01 | (0.95, 0.97) | $(2.45 \pm 0.04)$E-01 | $(3.97 \pm 0.04)$E-01 | $(1.00 \pm 0.08)$E+00 | $10000 \pm 0$ |
| | | (0.97, 0.95) | $(2.45 \pm 0.13)$E-01 | $(3.97 \pm 0.11)$E-01 | $(9.99 \pm 0.32)$E-01 | $10000 \pm 0$ |
| | | (0.9, 0.999) | $(2.45 \pm 0.02)$E-01 | $(3.97 \pm 0.01)$E-01 | $(1.00 \pm 0.03)$E+00 | $10000 \pm 0$ |
| | 0.001 | (0.95, 0.97) | $(2.45 \pm 0.05)$E-01 | $(3.97 \pm 0.04)$E-01 | $(9.99 \pm 0.12)$E-01 | $10000 \pm 0$ |
| | | (0.97, 0.95) | $(2.45 \pm 0.07)$E-01 | $(3.97 \pm 0.05)$E-01 | $(9.99 \pm 0.19)$E-01 | $10000 \pm 0$ |
| | | (0.9, 0.999) | $(2.45 \pm 0.06)$E-01 | $(3.97 \pm 0.05)$E-01 | $(9.99 \pm 0.12)$E-01 | $10000 \pm 0$ |
| 7 | 0.005 | (0.95, 0.97) | $(2.45 \pm 0.01)$E-01 | $(3.97 \pm 0.02)$E-01 | $(1.00 \pm 0.02)$E+00 | $10000 \pm 0$ |
| | | (0.97, 0.95) | $(2.45 \pm 0.01)$E-01 | $(3.97 \pm 0.01)$E-01 | $(1.00 \pm 0.01)$E+00 | $10000 \pm 0$ |
| | | (0.9, 0.999) | $(2.45 \pm 0.01)$E-01 | $(3.97 \pm 0.01)$E-01 | $(1.00 \pm 0.01)$E+00 | $10000 \pm 0$ |
| | 0.01 | (0.95, 0.97) | $(2.45 \pm 0.02)$E-01 | $(3.97 \pm 0.02)$E-01 | $(1.00 \pm 0.02)$E+00 | $10000 \pm 0$ |
| | | (0.97, 0.95) | $(2.45 \pm 0.01)$E-01 | $(3.97 \pm 0.02)$E-01 | $(1.00 \pm 0.02)$E+00 | $10000 \pm 0$ |

Table 15: Poisson Equation with Vanilla Tan

| $N_{\text{hidden}}$ | LR | Betas | MSE ± SD | MAE ± SD | Max Error ± SD | Epochs ± SD |
|---|---|---|---|---|---|---|
| 2 | 0.001 | (0.9, 0.999) | (8.64 ± 2.17)E-01 | (7.79 ± 0.89)E-01 | (2.15 ± 0.67)E+00 | 10000 ± 0 |
| | | (0.95, 0.97) | (9.87 ± 2.68)E-01 | (8.33 ± 1.24)E-01 | (2.30 ± 0.60)E+00 | 10000 ± 0 |
| | | (0.97, 0.95) | (8.91 ± 2.41)E-01 | (8.00 ± 1.09)E-01 | (2.02 ± 0.36)E+00 | 10000 ± 0 |
| | 0.005 | (0.9, 0.999) | (8.14 ± 1.01)E-01 | (7.65 ± 0.30)E-01 | (2.00 ± 0.43)E+00 | 10000 ± 0 |
| | | (0.95, 0.97) | (8.85 ± 2.13)E-01 | (7.91 ± 0.87)E-01 | (2.21 ± 0.69)E+00 | 10000 ± 0 |
| | | (0.97, 0.95) | (9.01 ± 2.08)E-01 | (7.94 ± 0.84)E-01 | (2.35 ± 0.76)E+00 | 10000 ± 0 |
| | 0.01 | (0.9, 0.999) | (8.11 ± 1.21)E-01 | (7.63 ± 0.38)E-01 | (2.07 ± 0.56)E+00 | 10000 ± 0 |
| | | (0.95, 0.97) | (8.12 ± 2.17)E-01 | (7.61 ± 0.99)E-01 | (2.01 ± 0.45)E+00 | 10000 ± 0 |
| | | (0.97, 0.95) | (7.89 ± 2.21)E-01 | (7.58 ± 1.03)E-01 | (1.87 ± 0.19)E+00 | 10000 ± 0 |
| **6** | **0.001** | (0.9, 0.999) | (3.29 ± 0.67)E-03 | (4.93 ± 0.61)E-02 | (2.39 ± 0.22)E-01 | 10000 ± 0 |
| | | **(0.95, 0.97)** | (1.14 ± 0.83)E-03 | (2.68 ± 1.08)E-02 | (1.50 ± 0.58)E-01 | 10000 ± 0 |
| | | (0.97, 0.95) | (6.43 ± 6.85)E-04 | (1.81 ± 1.05)E-02 | (1.14 ± 0.69)E-01 | 10000 ± 0 |
| | 0.005 | (0.9, 0.999) | (1.91 ± 0.95)E-03 | (3.58 ± 1.06)E-02 | (1.93 ± 0.46)E-01 | 10000 ± 0 |
| | | (0.95, 0.97) | (1.24 ± 0.85)E-03 | (2.74 ± 1.31)E-02 | (1.45 ± 0.64)E-01 | 10000 ± 0 |
| | | (0.97, 0.95) | (6.72 ± 4.79)E-04 | (2.00 ± 0.70)E-02 | (1.26 ± 0.51)E-01 | 10000 ± 0 |
| | 0.01 | (0.9, 0.999) | (2.02 ± 1.13)E-03 | (3.69 ± 1.26)E-02 | (1.90 ± 0.57)E-01 | 10000 ± 0 |
| | | (0.95, 0.97) | (1.27 ± 0.81)E-03 | (2.85 ± 1.08)E-02 | (1.63 ± 0.39)E-01 | 10000 ± 0 |
| | | (0.97, 0.95) | (6.35 ± 5.62)E-04 | (1.93 ± 0.85)E-02 | (1.08 ± 0.68)E-01 | 10000 ± 0 |
| 7 | 0.001 | (0.9, 0.999) | (2.94 ± 0.77)E-03 | (4.66 ± 0.71)E-02 | (2.19 ± 0.22)E-01 | 10000 ± 0 |
| | | (0.95, 0.97) | (1.18 ± 0.72)E-03 | (2.76 ± 1.01)E-02 | (1.46 ± 0.59)E-01 | 10000 ± 0 |
| | | (0.97, 0.95) | (4.18 ± 5.46)E-04 | (1.45 ± 0.96)E-02 | (8.28 ± 4.99)E-02 | 10000 ± 0 |
| | 0.005 | (0.9, 0.999) | (2.47 ± 0.64)E-03 | (4.18 ± 0.67)E-02 | (2.19 ± 0.27)E-01 | 10000 ± 0 |
| | | (0.95, 0.97) | (9.37 ± 9.26)E-04 | (2.26 ± 1.34)E-02 | (1.23 ± 0.74)E-01 | 10000 ± 0 |
| | | (0.97, 0.95) | (5.58 ± 7.99)E-04 | (1.59 ± 1.28)E-02 | (1.00 ± 0.56)E-01 | 10000 ± 0 |
| | 0.01 | (0.9, 0.999) | (1.86 ± 0.85)E-03 | (3.52 ± 0.95)E-02 | (1.92 ± 0.59)E-01 | 10000 ± 0 |
| | | (0.95, 0.97) | (9.58 ± 8.35)E-04 | (2.40 ± 1.20)E-02 | (1.30 ± 0.53)E-01 | 10000 ± 0 |
| | | (0.97, 0.95) | (4.72 ± 4.84)E-04 | (1.57 ± 0.97)E-02 | (9.99 ± 5.85)E-02 | 10000 ± 0 |

Table 16: GBS Equation with FLM

| $N$ | LR | Betas | MSE ± SD | MAE ± SD | Max Error ± SD | Epochs ± SD |
|---|---|---|---|---|---|---|
| **25** | **0.001** | (0.9, 0.999) | (8.34 ± 2.24)E-06 | (1.86 ± 0.27)E-03 | (2.41 ± 1.02)E-02 | 10000 ± 0 |
| | | (0.95, 0.97) | (1.89 ± 0.64)E-06 | (9.28 ± 1.57)E-04 | (1.28 ± 0.54)E-02 | 4781 ± 1334 |
| | | **(0.97, 0.95)** | (1.86 ± 0.50)E-06 | (8.92 ± 1.27)E-04 | (1.19 ± 0.38)E-02 | 3208 ± 619 |
| | 0.005 | (0.9, 0.999) | (6.39 ± 5.95)E-06 | (1.66 ± 0.93)E-03 | (1.36 ± 0.57)E-02 | 9965 ± 111 |
| | | (0.95, 0.97) | (5.13 ± 2.47)E-06 | (1.75 ± 0.46)E-03 | (1.24 ± 0.48)E-02 | 5116 ± 2025 |
| | | (0.97, 0.95) | (4.57 ± 1.32)E-06 | (1.72 ± 0.28)E-03 | (1.11 ± 0.36)E-02 | 3985 ± 1380 |
| | 0.01 | (0.9, 0.999) | (2.01 ± 3.26)E-05 | (2.52 ± 2.40)E-03 | (1.56 ± 0.49)E-02 | 9790 ± 432 |
| | | (0.95, 0.97) | (2.31 ± 2.21)E-05 | (3.62 ± 1.58)E-03 | (1.25 ± 0.40)E-02 | 6768 ± 2400 |
| | | (0.97, 0.95) | (1.46 ± 0.87)E-05 | (3.03 ± 0.98)E-03 | (1.11 ± 0.31)E-02 | 5732 ± 1903 |
| 49 | 0.001 | (0.9, 0.999) | (9.73 ± 8.01)E-06 | (1.87 ± 0.68)E-03 | (2.05 ± 0.57)E-02 | 10000 ± 0 |
| | | (0.95, 0.97) | (2.59 ± 1.51)E-06 | (1.01 ± 0.24)E-03 | (1.22 ± 0.30)E-02 | 4739 ± 833 |
| | | (0.97, 0.95) | (2.19 ± 1.42)E-06 | (9.50 ± 2.28)E-04 | (1.03 ± 0.24)E-02 | 3252 ± 433 |
| | 0.005 | (0.9, 0.999) | (1.20 ± 1.51)E-05 | (2.20 ± 1.80)E-03 | (1.26 ± 0.34)E-02 | 9473 ± 960 |
| | | (0.95, 0.97) | (6.91 ± 2.49)E-06 | (2.09 ± 0.40)E-03 | (1.04 ± 0.30)E-02 | 5493 ± 1427 |
| | | (0.97, 0.95) | (6.00 ± 2.50)E-06 | (1.92 ± 0.48)E-03 | (1.16 ± 0.49)E-02 | 3947 ± 1313 |
| | 0.01 | (0.9, 0.999) | (2.27 ± 1.11)E-06 | (9.54 ± 2.37)E-04 | (1.06 ± 0.40)E-02 | 7291 ± 935 |
| | | (0.95, 0.97) | (1.64 ± 0.91)E-05 | (3.14 ± 0.94)E-03 | (1.14 ± 0.27)E-02 | 8876 ± 1423 |
| | | (0.97, 0.95) | (2.48 ± 1.45)E-05 | (3.98 ± 1.27)E-03 | (1.16 ± 0.21)E-02 | 7146 ± 1947 |

Table 17: GBS Equation with SIREN–1

| $N_{\text{hidden}}$ | LR | Betas | MSE ± SD | MAE ± SD | Max Error ± SD | Epochs ± SD |
|---|---|---|---|---|---|---|
| **37** | **0.001** | (0.9, 0.999) | (2.38 ± 1.53)E-04 | (1.17 ± 0.43)E-02 | (8.47 ± 2.29)E-02 | 10000 ± 0 |
| | | **(0.95, 0.97)** | (5.76 ± 1.92)E-05 | (5.65 ± 0.86)E-03 | (5.07 ± 0.76)E-02 | 10000 ± 0 |
| | | (0.97, 0.95) | (8.13 ± 4.64)E-05 | (6.78 ± 2.09)E-03 | (5.22 ± 1.32)E-02 | 10000 ± 0 |
| | 0.005 | (0.9, 0.999) | (2.24 ± 2.36)E-04 | (1.14 ± 0.66)E-02 | (6.02 ± 2.22)E-02 | 10000 ± 0 |
| | | (0.95, 0.97) | (9.91 ± 4.64)E-05 | (7.63 ± 1.64)E-03 | (5.13 ± 1.18)E-02 | 10000 ± 0 |
| | | (0.97, 0.95) | (7.80 ± 2.61)E-05 | (6.82 ± 1.25)E-03 | (4.91 ± 0.79)E-02 | 10000 ± 0 |
| | 0.01 | (0.9, 0.999) | (1.55 ± 2.62)E-04 | (8.36 ± 6.27)E-03 | (5.36 ± 1.83)E-02 | 10000 ± 0 |
| | | (0.95, 0.97) | (1.86 ± 1.10)E-04 | (1.08 ± 0.37)E-02 | (5.37 ± 1.97)E-02 | 10000 ± 0 |
| | | (0.97, 0.95) | (1.12 ± 0.48)E-04 | (8.61 ± 1.93)E-03 | (4.81 ± 1.70)E-02 | 10000 ± 0 |
| 73 | 0.001 | (0.9, 0.999) | (1.74 ± 1.04)E-04 | (9.92 ± 3.05)E-03 | (6.11 ± 1.68)E-02 | 10000 ± 0 |
| | | (0.95, 0.97) | (6.89 ± 1.74)E-05 | (6.25 ± 0.79)E-03 | (4.35 ± 0.52)E-02 | 10000 ± 0 |
| | | (0.97, 0.95) | (6.24 ± 1.53)E-05 | (5.92 ± 0.71)E-03 | (4.11 ± 0.35)E-02 | 10000 ± 0 |
| | 0.005 | (0.9, 0.999) | (2.48 ± 3.87)E-04 | (1.06 ± 0.98)E-02 | (4.06 ± 1.47)E-02 | 10000 ± 0 |
| | | (0.95, 0.97) | (1.20 ± 0.46)E-04 | (8.85 ± 1.86)E-03 | (4.61 ± 1.14)E-02 | 10000 ± 0 |
| | | (0.97, 0.95) | (8.68 ± 4.94)E-05 | (7.10 ± 2.21)E-03 | (3.91 ± 0.63)E-02 | 10000 ± 0 |
| | 0.01 | (0.9, 0.999) | (8.81 ± 17.50)E-04 | (1.98 ± 2.09)E-02 | (5.62 ± 2.97)E-02 | 10000 ± 0 |
| | | (0.95, 0.97) | (2.58 ± 1.16)E-04 | (1.32 ± 0.28)E-02 | (5.93 ± 1.06)E-02 | 10000 ± 0 |
| | | (0.97, 0.95) | (1.59 ± 1.08)E-04 | (9.51 ± 3.68)E-03 | (4.32 ± 0.59)E-02 | 10000 ± 0 |

Table 18: GBS Equation with SIREN–2

| $N_{\text{hidden}}$ | LR | Betas | MSE ± SD | MAE ± SD | Max Error ± SD | Epochs ± SD |
|---|---|---|---|---|---|---|
| **10** | **0.001** | (0.9, 0.999) | (9.49 ± 19.00)E-06 | (1.74 ± 1.76)E-03 | (1.47 ± 1.51)E-02 | 9866 ± 302 |
| | | (0.95, 0.97) | (2.05 ± 0.80)E-06 | (1.08 ± 0.25)E-03 | (1.05 ± 0.62)E-02 | 3179 ± 1260 |
| | | **(0.97, 0.95)** | (2.56 ± 0.68)E-06 | (1.21 ± 0.11)E-03 | (1.02 ± 0.59)E-02 | 3176 ± 718 |
| | 0.005 | (0.9, 0.999) | (2.04 ± 0.92)E-06 | (1.04 ± 0.24)E-03 | (8.63 ± 4.52)E-03 | 7790 ± 1604 |
| | | (0.95, 0.97) | (2.35 ± 1.10)E-05 | (4.09 ± 1.28)E-03 | (1.13 ± 0.30)E-02 | 3765 ± 1829 |
| | | (0.97, 0.95) | (2.07 ± 0.89)E-05 | (3.67 ± 1.15)E-03 | (1.59 ± 0.48)E-02 | 3491 ± 1829 |
| | 0.01 | (0.9, 0.999) | (4.09 ± 5.13)E-06 | (1.44 ± 0.94)E-03 | (8.26 ± 3.51)E-03 | 7468 ± 2652 |
| | | (0.95, 0.97) | (7.58 ± 4.12)E-05 | (7.01 ± 2.52)E-03 | (2.04 ± 0.65)E-02 | 5043 ± 2902 |
| | | (0.97, 0.95) | (7.96 ± 3.28)E-05 | (7.63 ± 1.60)E-03 | (2.10 ± 0.51)E-02 | 3415 ± 2026 |
| 15 | 0.001 | (0.9, 0.999) | (4.30 ± 7.94)E-06 | (1.29 ± 1.16)E-03 | (8.47 ± 6.21)E-03 | 9856 ± 295 |
| | | (0.95, 0.97) | (3.21 ± 1.51)E-06 | (1.38 ± 0.28)E-03 | (9.06 ± 5.31)E-03 | 3452 ± 2239 |
| | | (0.97, 0.95) | (2.91 ± 1.62)E-06 | (1.28 ± 0.32)E-03 | (7.94 ± 3.21)E-03 | 2576 ± 883 |
| | 0.005 | (0.9, 0.999) | (2.29 ± 1.68)E-06 | (1.09 ± 0.44)E-03 | (7.43 ± 2.79)E-03 | 7266 ± 1114 |
| | | (0.95, 0.97) | (3.14 ± 2.61)E-05 | (4.49 ± 1.89)E-03 | (1.43 ± 0.35)E-02 | 3254 ± 1314 |
| | | (0.97, 0.95) | (4.05 ± 1.33)E-05 | (5.70 ± 1.17)E-03 | (1.55 ± 0.53)E-02 | 3061 ± 1707 |
| | 0.01 | (0.9, 0.999) | (1.82 ± 5.70)E-04 | (4.38 ± 11.00)E-03 | (1.29 ± 2.08)E-02 | 7723 ± 1590 |
| | | (0.95, 0.97) | (9.16 ± 4.19)E-05 | (8.03 ± 2.07)E-03 | (2.05 ± 0.35)E-02 | 3608 ± 1457 |
| | | (0.97, 0.95) | (1.17 ± 0.46)E-04 | (9.42 ± 2.27)E-03 | (2.26 ± 0.35)E-02 | 2772 ± 1414 |

Table 19: GBS Equation with Vanilla Leaky Relu

| $N_{\text{hidden}}$ | LR | Betas | MSE ± SD | MAE ± SD | Max Error ± SD | Epochs ± SD |
|---|---|---|---|---|---|---|
| **7** | **0.001** | **(0.9, 0.999)** | (6.43 ± 0.59)E-02 | (1.73 ± 0.09)E-01 | (9.22 ± 2.16)E-01 | 10000 ± 0 |
| | | (0.95, 0.97) | (6.59 ± 0.52)E-02 | (1.73 ± 0.10)E-01 | (1.10 ± 0.24)E+00 | 10000 ± 0 |
| | | (0.97, 0.95) | (6.71 ± 0.96)E-02 | (1.80 ± 0.10)E-01 | (1.02 ± 0.28)E+00 | 10000 ± 0 |
| | 0.005 | (0.9, 0.999) | (6.51 ± 0.57)E-02 | (1.72 ± 0.06)E-01 | (9.82 ± 2.46)E-01 | 10000 ± 0 |
| | | (0.95, 0.97) | (6.23 ± 0.82)E-02 | (1.70 ± 0.11)E-01 | (9.88 ± 2.59)E-01 | 10000 ± 0 |
| | | (0.97, 0.95) | (6.31 ± 0.81)E-02 | (1.73 ± 0.10)E-01 | (8.94 ± 2.06)E-01 | 10000 ± 0 |
| | 0.01 | (0.9, 0.999) | (6.15 ± 0.81)E-02 | (1.74 ± 0.08)E-01 | (8.26 ± 1.16)E-01 | 10000 ± 0 |
| | | (0.95, 0.97) | (6.16 ± 0.54)E-02 | (1.76 ± 0.12)E-01 | (8.43 ± 0.98)E-01 | 10000 ± 0 |
| | | (0.97, 0.95) | (6.51 ± 0.69)E-02 | (1.72 ± 0.11)E-01 | (1.03 ± 0.31)E+00 | 10000 ± 0 |
| 11 | 0.001 | (0.9, 0.999) | (6.47 ± 1.11)E-02 | (1.67 ± 0.11)E-01 | (1.03 ± 0.26)E+00 | 10000 ± 0 |
| | | (0.95, 0.97) | (6.18 ± 0.60)E-02 | (1.61 ± 0.06)E-01 | (1.13 ± 0.26)E+00 | 10000 ± 0 |
| | | (0.97, 0.95) | (7.06 ± 1.05)E-02 | (1.73 ± 0.11)E-01 | (1.23 ± 0.14)E+00 | 10000 ± 0 |
| | 0.005 | (0.9, 0.999) | (6.64 ± 0.70)E-02 | (1.74 ± 0.12)E-01 | (1.01 ± 0.23)E+00 | 10000 ± 0 |
| | | (0.95, 0.97) | (6.65 ± 0.84)E-02 | (1.72 ± 0.12)E-01 | (1.05 ± 0.29)E+00 | 10000 ± 0 |
| | | (0.97, 0.95) | (6.69 ± 0.88)E-02 | (1.70 ± 0.14)E-01 | (1.12 ± 0.29)E+00 | 10000 ± 0 |
| | 0.01 | (0.9, 0.999) | (6.56 ± 0.78)E-02 | (1.78 ± 0.14)E-01 | (9.39 ± 1.69)E-01 | 10000 ± 0 |
| | | (0.95, 0.97) | (6.87 ± 1.48)E-02 | (1.78 ± 0.18)E-01 | (1.03 ± 0.31)E+00 | 10000 ± 0 |
| | | (0.97, 0.95) | (6.45 ± 1.02)E-02 | (1.73 ± 0.12)E-01 | (2.07 ± 3.33)E+00 | 10000 ± 0 |

Table 20: GBS Equation with Vanilla Tan

| $N_{\text{hidden}}$ | LR | Betas | MSE ± SD | MAE ± SD | Max Error ± SD | Epochs ± SD |
|---|---|---|---|---|---|---|
| **7** | 0.001 | (0.9, 0.999) | (3.76 ± 2.32)E-06 | (1.44 ± 0.57)E-03 | (2.30 ± 2.17)E-02 | 7482 ± 446 |
| | | (0.95, 0.97) | (4.24 ± 2.81)E-06 | (1.50 ± 0.58)E-03 | (2.14 ± 1.97)E-02 | 1402 ± 136 |
| | | (0.97, 0.95) | (7.57 ± 3.89)E-06 | (2.13 ± 0.58)E-03 | (1.14 ± 0.67)E-02 | 1184 ± 67 |
| | **0.005** | (0.9, 0.999) | (1.10 ± 1.63)E-05 | (2.42 ± 2.04)E-03 | (2.37 ± 1.97)E-02 | 3324 ± 419 |
| | | (0.95, 0.97) | (1.46 ± 1.08)E-05 | (3.11 ± 1.27)E-03 | (1.54 ± 0.97)E-02 | 618 ± 98 |
| | | **(0.97, 0.95)** | (1.20 ± 0.56)E-05 | (2.85 ± 0.89)E-03 | (1.30 ± 0.54)E-02 | 626 ± 61 |
| | 0.01 | (0.9, 0.999) | (3.42 ± 2.03)E-06 | (1.38 ± 0.40)E-03 | (1.92 ± 1.88)E-02 | 2252 ± 463 |
| | | (0.95, 0.97) | (3.68 ± 2.43)E-05 | (4.96 ± 2.24)E-03 | (1.77 ± 0.66)E-02 | 540 ± 126 |
| | | (0.97, 0.95) | (3.79 ± 1.34)E-05 | (5.33 ± 1.30)E-03 | (1.50 ± 0.34)E-02 | 536 ± 36 |
| 11 | 0.001 | (0.9, 0.999) | (2.85 ± 1.87)E-06 | (1.26 ± 0.42)E-03 | (1.80 ± 1.05)E-02 | 5829 ± 656 |
| | | (0.95, 0.97) | (5.37 ± 1.69)E-06 | (1.81 ± 0.32)E-03 | (1.37 ± 1.02)E-02 | 934 ± 45 |
| | | (0.97, 0.95) | (1.03 ± 0.47)E-05 | (2.56 ± 0.61)E-03 | (8.88 ± 1.61)E-03 | 926 ± 46 |
| | 0.005 | (0.9, 0.999) | (3.15 ± 1.73)E-06 | (1.35 ± 0.37)E-03 | (1.56 ± 0.96)E-02 | 1926 ± 393 |
| | | (0.95, 0.97) | (1.73 ± 1.35)E-05 | (3.41 ± 1.45)E-03 | (1.11 ± 0.22)E-02 | 436 ± 45 |
| | | (0.97, 0.95) | (2.31 ± 1.19)E-05 | (4.05 ± 1.25)E-03 | (1.21 ± 0.25)E-02 | 551 ± 34 |
| | 0.01 | (0.9, 0.999) | (4.80 ± 3.32)E-06 | (1.67 ± 0.62)E-03 | (1.30 ± 0.72)E-02 | 1163 ± 352 |
| | | (0.95, 0.97) | (4.08 ± 2.35)E-05 | (5.49 ± 2.08)E-03 | (1.81 ± 0.44)E-02 | 396 ± 51 |
| | | (0.97, 0.95) | (6.58 ± 4.83)E-05 | (6.86 ± 3.20)E-03 | (1.80 ± 0.38)E-02 | 513 ± 58 |

Table 21: Burgers' Equation with FLM

| N | LR | Betas | MSE ± SD | MAE ± SD | Max Error ± SD | Epochs ± SD |
|---|---|---|---|---|---|---|
| | | (0.9, 0.999) | (8.62 ± 0.45)E-03 | (5.33 ± 0.25)E-02 | (3.54 ± 0.13)E-01 | 10000 ± 0 |
| | 0.0005 | (0.95, 0.97) | (8.58 ± 0.36)E-03 | (5.33 ± 0.23)E-02 | (3.49 ± 0.10)E-01 | 10000 ± 0 |
| | | (0.97, 0.95) | (8.51 ± 0.30)E-03 | (5.25 ± 0.16)E-02 | (3.54 ± 0.10)E-01 | 10000 ± 0 |
| | | (0.9, 0.999) | (8.60 ± 0.39)E-03 | (5.26 ± 0.17)E-02 | (3.59 ± 0.15)E-01 | 10000 ± 0 |
| 25 | 0.001 | (0.95, 0.97) | (8.43 ± 0.33)E-03 | (5.10 ± 0.23)E-02 | (3.58 ± 0.10)E-01 | 10000 ± 0 |
| | | (0.97, 0.95) | (8.56 ± 0.36)E-03 | (5.17 ± 0.23)E-02 | (3.61 ± 0.12)E-01 | 10000 ± 0 |
| | | (0.9, 0.999) | (8.65 ± 0.44)E-03 | (5.30 ± 0.31)E-02 | (3.55 ± 0.15)E-01 | 10000 ± 0 |
| | 0.005 | (0.95, 0.97) | (8.39 ± 0.24)E-03 | (5.11 ± 0.21)E-02 | (3.57 ± 0.10)E-01 | 10000 ± 0 |
| | | (0.97, 0.95) | (8.52 ± 0.38)E-03 | (5.19 ± 0.23)E-02 | (3.62 ± 0.12)E-01 | 10000 ± 0 |
| | | **(0.9, 0.999)** | (6.85 ± 0.35)E-03 | (4.38 ± 0.23)E-02 | (3.70 ± 0.29)E-01 | 10000 ± 0 |
| | **0.0005** | (0.95, 0.97) | (6.85 ± 0.35)E-03 | (4.38 ± 0.25)E-02 | (3.80 ± 0.37)E-01 | 10000 ± 0 |
| | | (0.97, 0.95) | (6.89 ± 0.33)E-03 | (4.48 ± 0.30)E-02 | (3.70 ± 0.23)E-01 | 10000 ± 0 |
| | | (0.9, 0.999) | (6.66 ± 0.31)E-03 | (4.25 ± 0.18)E-02 | (3.68 ± 0.21)E-01 | 10000 ± 0 |
| **64** | 0.001 | (0.95, 0.97) | (6.58 ± 0.27)E-03 | (4.19 ± 0.16)E-02 | (3.67 ± 0.21)E-01 | 10000 ± 0 |
| | | (0.97, 0.95) | (6.63 ± 0.28)E-03 | (4.23 ± 0.26)E-02 | (3.70 ± 0.27)E-01 | 10000 ± 0 |
| | | (0.9, 0.999) | (6.93 ± 0.23)E-03 | (4.46 ± 0.16)E-02 | (3.73 ± 0.23)E-01 | 10000 ± 0 |
| | 0.005 | (0.95, 0.97) | (6.59 ± 0.23)E-03 | (4.18 ± 0.11)E-02 | (3.74 ± 0.27)E-01 | 10000 ± 0 |
| | | (0.97, 0.95) | (6.52 ± 0.24)E-03 | (4.10 ± 0.13)E-02 | (3.73 ± 0.25)E-01 | 10000 ± 0 |

Table 22: Burgers' Equation with SIREN–1

| $N_{\text{hidden}}$ | LR | Betas | MSE ± SD | MAE ± SD | Max Error ± SD | Epochs ± SD |
|---|---|---|---|---|---|---|
| | | (0.9, 0.999) | (3.26 ± 1.51)E-02 | (1.51 ± 0.37)E-01 | (4.23 ± 0.32)E-01 | 10000 ± 0 |
| | **0.0005** | **(0.95, 0.97)** | (2.21 ± 0.07)E-02 | (1.24 ± 0.03)E-01 | (3.86 ± 0.13)E-01 | 10000 ± 0 |
| | | (0.97, 0.95) | (2.22 ± 0.07)E-02 | (1.24 ± 0.04)E-01 | (3.87 ± 0.14)E-01 | 10000 ± 0 |
| | | (0.9, 0.999) | (2.03 ± 0.10)E-02 | (1.16 ± 0.05)E-01 | (3.57 ± 0.08)E-01 | 10000 ± 0 |
| **37** | 0.001 | (0.95, 0.97) | (2.12 ± 0.09)E-02 | (1.20 ± 0.04)E-01 | (3.74 ± 0.16)E-01 | 10000 ± 0 |
| | | (0.97, 0.95) | (2.11 ± 0.07)E-02 | (1.20 ± 0.04)E-01 | (3.77 ± 0.15)E-01 | 10000 ± 0 |
| | | (0.9, 0.999) | (2.07 ± 0.08)E-02 | (1.14 ± 0.03)E-01 | (3.79 ± 0.20)E-01 | 10000 ± 0 |
| | 0.005 | (0.95, 0.97) | (2.10 ± 0.07)E-02 | (1.14 ± 0.03)E-01 | (3.88 ± 0.19)E-01 | 10000 ± 0 |
| | | (0.97, 0.95) | (2.10 ± 0.06)E-02 | (1.14 ± 0.02)E-01 | (3.89 ± 0.18)E-01 | 10000 ± 0 |
| | | (0.9, 0.999) | (2.33 ± 0.63)E-02 | (1.27 ± 0.18)E-01 | (3.80 ± 0.34)E-01 | 10000 ± 0 |
| | 0.0005 | (0.95, 0.97) | (2.21 ± 0.09)E-02 | (1.24 ± 0.03)E-01 | (3.80 ± 0.11)E-01 | 10000 ± 0 |
| | | (0.97, 0.95) | (2.23 ± 0.08)E-02 | (1.25 ± 0.03)E-01 | (3.83 ± 0.10)E-01 | 10000 ± 0 |
| | | (0.9, 0.999) | (2.03 ± 0.08)E-02 | (1.15 ± 0.04)E-01 | (3.67 ± 0.14)E-01 | 10000 ± 0 |
| 93 | 0.001 | (0.95, 0.97) | (2.10 ± 0.08)E-02 | (1.19 ± 0.04)E-01 | (3.60 ± 0.12)E-01 | 10000 ± 0 |
| | | (0.97, 0.95) | (2.11 ± 0.08)E-02 | (1.20 ± 0.03)E-01 | (3.64 ± 0.13)E-01 | 10000 ± 0 |
| | | (0.9, 0.999) | (2.06 ± 0.09)E-02 | (1.14 ± 0.04)E-01 | (3.83 ± 0.22)E-01 | 10000 ± 0 |
| | 0.005 | (0.95, 0.97) | (2.14 ± 0.04)E-02 | (1.15 ± 0.02)E-01 | (3.96 ± 0.13)E-01 | 10000 ± 0 |
| | | (0.97, 0.95) | (2.13 ± 0.02)E-02 | (1.14 ± 0.02)E-01 | (3.95 ± 0.12)E-01 | 10000 ± 0 |

Table 23: Burgers' Equation with SIREN–2

| $N_{\text{hidden}}$ | LR | Betas | MSE ± SD | MAE ± SD | Max Error ± SD | Epochs ± SD |
|---|---|---|---|---|---|---|
| | | (0.9, 0.999) | (2.03 ± 0.11)E-02 | (1.10 ± 0.45)E-01 | (3.71 ± 0.15)E-01 | 10000 ± 0 |
| | **0.0005** | (0.95, 0.97) | (1.42 ± 0.18)E-02 | (7.99 ± 0.86)E-02 | (3.63 ± 0.14)E-01 | 10000 ± 0 |
| | | **(0.97, 0.95)** | (1.28 ± 0.09)E-02 | (7.27 ± 0.50)E-02 | (3.63 ± 0.07)E-01 | 10000 ± 0 |
| | | (0.9, 0.999) | (1.52 ± 0.25)E-02 | (8.71 ± 1.07)E-02 | (3.57 ± 0.08)E-01 | 10000 ± 0 |
| **10** | 0.001 | (0.95, 0.97) | (1.24 ± 0.19)E-02 | (7.19 ± 0.94)E-02 | (3.59 ± 0.13)E-01 | 10000 ± 0 |
| | | (0.97, 0.95) | (1.17 ± 0.15)E-02 | (6.82 ± 0.71)E-02 | (3.57 ± 0.16)E-01 | 10000 ± 0 |
| | | (0.9, 0.999) | (1.31 ± 0.19)E-02 | (7.67 ± 0.88)E-02 | (3.57 ± 0.09)E-01 | 10000 ± 0 |
| | 0.005 | (0.95, 0.97) | (1.12 ± 0.17)E-02 | (6.60 ± 0.79)E-02 | (3.56 ± 0.09)E-01 | 10000 ± 0 |
| | | (0.97, 0.95) | (9.57 ± 0.08)E-03 | (5.77 ± 0.39)E-02 | (3.56 ± 0.13)E-01 | 10000 ± 0 |
| | | (0.9, 0.999) | (1.84 ± 0.34)E-02 | (9.76 ± 1.29)E-02 | (3.75 ± 0.24)E-01 | 10000 ± 0 |
| | 0.0005 | (0.95, 0.97) | (1.30 ± 0.09)E-02 | (7.29 ± 0.30)E-02 | (3.68 ± 0.24)E-01 | 10000 ± 0 |
| | | (0.97, 0.95) | (1.25 ± 0.08)E-02 | (6.99 ± 0.31)E-02 | (3.66 ± 0.22)E-01 | 10000 ± 0 |
| | | (0.9, 0.999) | (1.51 ± 0.27)E-02 | (8.36 ± 1.04)E-02 | (3.70 ± 0.23)E-01 | 10000 ± 0 |
| 17 | 0.001 | (0.95, 0.97) | (1.21 ± 0.12)E-02 | (6.85 ± 0.45)E-02 | (3.67 ± 0.21)E-01 | 10000 ± 0 |
| | | (0.97, 0.95) | (1.14 ± 0.12)E-02 | (6.54 ± 0.45)E-02 | (3.67 ± 0.24)E-01 | 10000 ± 0 |
| | | (0.9, 0.999) | (1.23 ± 0.20)E-02 | (7.13 ± 0.98)E-02 | (3.67 ± 0.24)E-01 | 10000 ± 0 |
| | 0.005 | (0.95, 0.97) | (1.01 ± 0.09)E-02 | (5.95 ± 0.48)E-02 | (3.69 ± 0.24)E-01 | 10000 ± 0 |
| | | (0.97, 0.95) | (9.70 ± 0.10)E-03 | (5.74 ± 0.46)E-02 | (3.67 ± 0.22)E-01 | 10000 ± 0 |

Table 24: Burgers' Equation with SIREN–3

| $N_{\text{hidden}}$ | LR | Betas | MSE $\pm$ SD | MAE $\pm$ SD | Max Error $\pm$ SD | Epochs $\pm$ SD |
|---|---|---|---|---|---|---|
| **7** | 0.0005 | (0.9, 0.999) | (1.67 $\pm$ 0.39)E-02 | (9.10 $\pm$ 1.74)E-02 | (3.65 $\pm$ 0.18)E-01 | 10000 $\pm$ 0 |
| | | (0.95, 0.97) | (9.86 $\pm$ 1.04)E-03 | (5.94 $\pm$ 0.58)E-02 | (3.61 $\pm$ 0.15)E-01 | 10000 $\pm$ 0 |
| | | (0.97, 0.95) | (9.19 $\pm$ 0.60)E-03 | (5.54 $\pm$ 0.29)E-02 | (3.60 $\pm$ 0.15)E-01 | 10000 $\pm$ 0 |
| | 0.001 | (0.9, 0.999) | (1.50 $\pm$ 0.37)E-02 | (8.33 $\pm$ 1.68)E-02 | (3.62 $\pm$ 0.13)E-01 | 10000 $\pm$ 0 |
| | | (0.95, 0.97) | (9.51 $\pm$ 0.67)E-03 | (5.73 $\pm$ 0.30)E-02 | (3.60 $\pm$ 0.12)E-01 | 10000 $\pm$ 0 |
| | | (0.97, 0.95) | (8.83 $\pm$ 0.66)E-03 | (5.35 $\pm$ 0.35)E-02 | (3.62 $\pm$ 0.10)E-01 | 10000 $\pm$ 0 |
| | **0.005** | (0.9, 0.999) | (1.05 $\pm$ 0.09)E-02 | (6.26 $\pm$ 0.48)E-02 | (3.60 $\pm$ 0.16)E-01 | 10000 $\pm$ 0 |
| | | (0.95, 0.97) | (8.86 $\pm$ 0.56)E-03 | (5.35 $\pm$ 0.24)E-02 | (3.63 $\pm$ 0.18)E-01 | 10000 $\pm$ 0 |
| | | **(0.97, 0.95)** | (7.91 $\pm$ 0.92)E-03 | (4.87 $\pm$ 0.49)E-02 | (3.67 $\pm$ 0.19)E-01 | 10000 $\pm$ 0 |
| 12 | 0.0005 | (0.9, 0.999) | (1.24 $\pm$ 0.15)E-02 | (7.09 $\pm$ 0.69)E-02 | (3.69 $\pm$ 0.16)E-01 | 10000 $\pm$ 0 |
| | | (0.95, 0.97) | (8.83 $\pm$ 0.58)E-03 | (5.30 $\pm$ 0.26)E-02 | (3.65 $\pm$ 0.18)E-01 | 10000 $\pm$ 0 |
| | | (0.97, 0.95) | (8.26 $\pm$ 0.61)E-03 | (5.01 $\pm$ 0.30)E-02 | (3.66 $\pm$ 0.16)E-01 | 10000 $\pm$ 0 |
| | 0.001 | (0.9, 0.999) | (1.07 $\pm$ 0.17)E-02 | (6.24 $\pm$ 0.68)E-02 | (3.68 $\pm$ 0.16)E-01 | 10000 $\pm$ 0 |
| | | (0.95, 0.97) | (8.77 $\pm$ 1.18)E-03 | (5.22 $\pm$ 0.56)E-02 | (3.67 $\pm$ 0.16)E-01 | 10000 $\pm$ 0 |
| | | (0.97, 0.95) | (8.01 $\pm$ 1.10)E-03 | (4.83 $\pm$ 0.49)E-02 | (3.68 $\pm$ 0.19)E-01 | 10000 $\pm$ 0 |
| | 0.005 | (0.9, 0.999) | (9.09 $\pm$ 1.82)E-03 | (5.42 $\pm$ 0.80)E-02 | (3.70 $\pm$ 0.18)E-01 | 10000 $\pm$ 0 |
| | | (0.95, 0.97) | (7.84 $\pm$ 1.31)E-03 | (4.79 $\pm$ 0.61)E-02 | (3.67 $\pm$ 0.20)E-01 | 10000 $\pm$ 0 |
| | | (0.97, 0.95) | (7.40 $\pm$ 1.08)E-03 | (4.53 $\pm$ 0.53)E-02 | (3.72 $\pm$ 0.31)E-01 | 10000 $\pm$ 0 |

Table 25: Burgers' Equation with Vanilla Relu

| $N_{\text{hidden}}$ | LR | Betas | MSE $\pm$ SD | MAE $\pm$ SD | Max Error $\pm$ SD | Epochs $\pm$ SD |
|---|---|---|---|---|---|---|
| **7** | **0.0005** | **(0.9, 0.999)** | (3.22 $\pm$ 1.06)E-02 | (1.49 $\pm$ 0.28)E-01 | (3.72 $\pm$ 0.24)E-01 | 10000 $\pm$ 0 |
| | | (0.95, 0.97) | (3.31 $\pm$ 1.17)E-02 | (1.51 $\pm$ 0.31)E-01 | (3.66 $\pm$ 0.13)E-01 | 10000 $\pm$ 0 |
| | | (0.97, 0.95) | (3.48 $\pm$ 1.28)E-02 | (1.55 $\pm$ 0.33)E-01 | (3.76 $\pm$ 0.25)E-01 | 10000 $\pm$ 0 |
| | 0.001 | (0.9, 0.999) | (3.69 $\pm$ 1.36)E-02 | (1.60 $\pm$ 0.35)E-01 | (3.80 $\pm$ 0.32)E-01 | 10000 $\pm$ 0 |
| | | (0.95, 0.97) | (3.30 $\pm$ 1.18)E-02 | (1.51 $\pm$ 0.31)E-01 | (3.64 $\pm$ 0.14)E-01 | 10000 $\pm$ 0 |
| | | (0.97, 0.95) | (3.93 $\pm$ 1.75)E-02 | (1.64 $\pm$ 0.39)E-01 | (3.93 $\pm$ 0.67)E-01 | 10000 $\pm$ 0 |
| | 0.005 | (0.9, 0.999) | (3.29 $\pm$ 1.17)E-02 | (1.50 $\pm$ 0.31)E-01 | (3.67 $\pm$ 0.17)E-01 | 10000 $\pm$ 0 |
| | | (0.95, 0.97) | (3.24 $\pm$ 1.20)E-02 | (1.48 $\pm$ 0.32)E-01 | (3.70 $\pm$ 0.17)E-01 | 10000 $\pm$ 0 |
| | | (0.97, 0.95) | (4.17 $\pm$ 1.76)E-02 | (1.70 $\pm$ 0.39)E-01 | (3.95 $\pm$ 0.66)E-01 | 10000 $\pm$ 0 |
| 12 | 0.0005 | (0.9, 0.999) | (2.82 $\pm$ 0.80)E-02 | (1.38 $\pm$ 0.20)E-01 | (3.63 $\pm$ 0.28)E-01 | 10000 $\pm$ 0 |
| | | (0.95, 0.97) | (2.59 $\pm$ 0.09)E-02 | (1.32 $\pm$ 0.04)E-01 | (3.58 $\pm$ 0.10)E-01 | 10000 $\pm$ 0 |
| | | (0.97, 0.95) | (3.04 $\pm$ 1.40)E-02 | (1.42 $\pm$ 0.29)E-01 | (3.72 $\pm$ 0.58)E-01 | 10000 $\pm$ 0 |
| | 0.001 | (0.9, 0.999) | (2.83 $\pm$ 0.73)E-02 | (1.39 $\pm$ 0.19)E-01 | (3.65 $\pm$ 0.21)E-01 | 10000 $\pm$ 0 |
| | | (0.95, 0.97) | (2.60 $\pm$ 0.13)E-02 | (1.32 $\pm$ 0.05)E-01 | (3.59 $\pm$ 0.07)E-01 | 10000 $\pm$ 0 |
| | | (0.97, 0.95) | (2.90 $\pm$ 0.69)E-02 | (1.41 $\pm$ 0.18)E-01 | (3.66 $\pm$ 0.20)E-01 | 10000 $\pm$ 0 |
| | 0.005 | (0.9, 0.999) | (2.64 $\pm$ 0.08)E-02 | (1.34 $\pm$ 0.03)E-01 | (3.58 $\pm$ 0.09)E-01 | 10000 $\pm$ 0 |
| | | (0.95, 0.97) | (2.84 $\pm$ 0.70)E-02 | (1.39 $\pm$ 0.19)E-01 | (3.63 $\pm$ 0.19)E-01 | 10000 $\pm$ 0 |
| | | (0.97, 0.95) | (3.16 $\pm$ 1.01)E-02 | (1.47 $\pm$ 0.25)E-01 | (3.74 $\pm$ 0.35)E-01 | 10000 $\pm$ 0 |

Table 26: Burgers' Equation with Vanilla Tan

| $N_{\text{hidden}}$ | LR | Betas | MSE $\pm$ SD | MAE $\pm$ SD | Max Error $\pm$ SD | Epochs $\pm$ SD |
|---|---|---|---|---|---|---|
| 7 | 0.0005 | (0.9, 0.999) | (1.16 $\pm$ 0.16)E-02 | (6.84 $\pm$ 0.76)E-02 | (3.63 $\pm$ 0.18)E-01 | 10000 $\pm$ 0 |
| | | (0.95, 0.97) | (6.82 $\pm$ 0.42)E-03 | (4.38 $\pm$ 0.26)E-02 | (3.60 $\pm$ 0.19)E-01 | 10000 $\pm$ 0 |
| | | (0.97, 0.95) | (6.02 $\pm$ 0.50)E-03 | (3.87 $\pm$ 0.29)E-02 | (3.88 $\pm$ 0.60)E-01 | 10000 $\pm$ 0 |
| | 0.001 | (0.9, 0.999) | (8.23 $\pm$ 1.07)E-03 | (5.08 $\pm$ 0.58)E-02 | (3.63 $\pm$ 0.11)E-01 | 10000 $\pm$ 0 |
| | | (0.95, 0.97) | (6.37 $\pm$ 0.72)E-03 | (4.12 $\pm$ 0.34)E-02 | (3.68 $\pm$ 0.29)E-01 | 10000 $\pm$ 0 |
| | | (0.97, 0.95) | (5.81 $\pm$ 0.57)E-03 | (3.73 $\pm$ 0.35)E-02 | (3.85 $\pm$ 0.59)E-01 | 10000 $\pm$ 0 |
| | 0.005 | (0.9, 0.999) | (7.24 $\pm$ 0.89)E-03 | (4.55 $\pm$ 0.53)E-02 | (3.71 $\pm$ 0.29)E-01 | 10000 $\pm$ 0 |
| | | (0.95, 0.97) | (6.16 $\pm$ 0.53)E-03 | (3.94 $\pm$ 0.30)E-02 | (3.73 $\pm$ 0.36)E-01 | 10000 $\pm$ 0 |
| | | (0.97, 0.95) | (5.53 $\pm$ 0.51)E-03 | (3.68 $\pm$ 0.35)E-02 | (3.86 $\pm$ 0.52)E-01 | 10000 $\pm$ 0 |
| **12** | **0.0005** | (0.9, 0.999) | (8.95 $\pm$ 2.13)E-03 | (5.40 $\pm$ 1.14)E-02 | (3.61 $\pm$ 0.18)E-01 | 10000 $\pm$ 0 |
| | | **(0.95, 0.97)** | (5.17 $\pm$ 0.61)E-03 | (3.26 $\pm$ 0.36)E-02 | (4.14 $\pm$ 0.80)E-01 | 10000 $\pm$ 0 |
| | | (0.97, 0.95) | (4.82 $\pm$ 0.42)E-03 | (3.05 $\pm$ 0.17)E-02 | (4.15 $\pm$ 0.93)E-01 | 10000 $\pm$ 0 |
| | 0.001 | (0.9, 0.999) | (7.78 $\pm$ 0.93)E-03 | (4.89 $\pm$ 0.56)E-02 | (3.64 $\pm$ 0.19)E-01 | 10000 $\pm$ 0 |
| | | (0.95, 0.97) | (5.00 $\pm$ 0.51)E-03 | (3.18 $\pm$ 0.33)E-02 | (4.19 $\pm$ 0.69)E-01 | 10000 $\pm$ 0 |
| | | (0.97, 0.95) | (4.75 $\pm$ 0.57)E-03 | (2.99 $\pm$ 0.32)E-02 | (4.57 $\pm$ 0.95)E-01 | 10000 $\pm$ 0 |
| | 0.005 | (0.9, 0.999) | (5.93 $\pm$ 0.72)E-03 | (3.84 $\pm$ 0.45)E-02 | (3.81 $\pm$ 0.45)E-01 | 10000 $\pm$ 0 |
| | | (0.95, 0.97) | (5.00 $\pm$ 0.44)E-03 | (3.22 $\pm$ 0.28)E-02 | (4.23 $\pm$ 0.88)E-01 | 10000 $\pm$ 0 |
| | | (0.97, 0.95) | (4.42 $\pm$ 0.45)E-03 | (2.82 $\pm$ 0.31)E-02 | (4.75 $\pm$ 0.92)E-01 | 10000 $\pm$ 0 |

