# OpenReview forum: "Fourier Learning Machines: Nonharmonic Fourier-Based Neural Networks for Scientific Machine Learning"
_TMLR — Accepted by TMLR_

### Review · Reviewer_6H1i · 2025-10-20

**Summary Of Contributions:**

The paper proposes Fourier learning machine, a feedforward neural network that uses cosine functions as activation layers.
The main idea is to decompose the input signal into its spectral basis and learn coefficients to combine these for the end task.
To do this, initially the Fourier decomposition of the input signal is written as a combination of cosines and sines. Secondly, the trigonometric decomposition is written in a cosine phase-shift form that can be expressed in the form of a neural network free of complex numbers. In this form, the Fourier coefficients combine the cosine and sine of the input in different linear manners.
To generalize this representation to higher dimensions, the authors define the m–Lexi Sign Matrix, a structured matrix that systematically encodes all possible sign combinations of frequency components across the input dimensions.
This matrix allows the model to efficiently generate the full set of separable Fourier basis functions (the various sine–cosine products) in a compact cosine-only representation.
Overall, by allowing these frequencies to be arbitrary and trainable, the model learns its own spectral basis that addresses the end task. This makes the model capable of capturing both periodic and nonperiodic behaviors in data using a simple multilayer perceptron–like architecture.
The FLM architecture operationalizes this spectral formulation through a collection of parallel sub-networks, each representing one frequency component of the learned Fourier basis. Each sub-network corresponds to a particular learnable frequency vector and produces a sum of phase-shifted cosine activations whose amplitudes and phases are trained from data. By summing the outputs of all sub-networks, the FLM reconstructs the target function as a weighted superposition of learned spectral components, thereby realizing a neural analogue of the multidimensional nonharmonic Fourier series.
The authors apply it to the numerical solution of several benchmark partial differential equations and an optimal control problem. In these experiments, the FLM serves as a mesh-free spectral neural solver, where the network’s learnable frequency components replace fixed basis functions traditionally used in numerical schemes.

**Audience:**

Yes

**Audience Explanation:**

It is an interesting methodology that relates directly to neural representation learning and time series forecasting.

**Broader Impact Concerns:**

There is no evident ethical concern from this work.

**Claims And Evidence:**

Yes

**Claims Explanation:**

The methodology is novel, the explanation is clear and the results are promising.
Across all PDE benchmarks, the FLM consistently achieved superior accuracy compared to standard neural architectures such as SIREN, ReLU, and Tanh networks. In both the heat and Poisson equations, the FLM attained errors one to two orders of magnitude lower than its competitors, demonstrating its ability to capture smooth and oscillatory solutions with high spectral fidelity.
In the nonlinear optimal control problem arising from the evolutionary dynamics of a Rock–Paper–Scissors game, both the state trajectories and the control policy were approximated using coupled FLMs trained via a penalty-based loss formulation. The model successfully learned cyclically symmetric solutions consistent with the problem’s odd–circulant structure and closely matched the reference trajectories
These experiments collectively illustrate that FLMs can serve as general-purpose spectral solvers for both forward PDE problems and optimal control formulations.

**Requested Changes:**

1. To demonstrate the applicability of FLM, the model should be compared on a real-world time-series forecasting problem to demonstrate its ability to learn dominant frequencies and trends from empirical data, and compare with the SOTA.

2. It would also be beneficial to add an experiment on a high-dimensional OCP. To my understanding, all current examples are ≤3-dimensional. Since the FLM architecture scales naturally with the number of inputs via the m–Lexi Sign Matrix, testing higher-dimensional dynamics (e.g., a 5-strategy evolutionary game or a 3D diffusion equation) would demonstrate scalability and generalization.

3. The paper would benefit from an experiment with time-dependent PDEs that have non-periodic boundary conditions. Current PDE examples (heat, Poisson, Black–Scholes) hase smooth periodic structure. Since FLMs are inherently spectral, testing them on non-periodic or discontinuous boundary conditions would show robustness.


Minor:

'To begin with, we set the frequencies to nonnegative vectors...' I assume the authors mean they set them to zero.

---

> ### Author Response · Authors · 2025-11-15
>
> Thank you for your review, comments and suggestions.
>
> We are now providing an update on the state of the revisions requested and anticipate that a fully revised manuscript will be uploaded by the end of day on Monday 11/17.
>
> Regarding the requested changes:
>
> **1. Performance in Time Series Prediction**
>
> - We understand the importance of testing FLMs for time series prediction tasks, and even though our focus is on PDEs and Control, we are happy to include an assessment of FLMs for (univariate) time series prediction. Results regarding this request will be presented and discussed in the updated manuscript.
>
> **2. Higher Dimensional OCP**
>
> - We agree that a higher-dimensional problem is beneficial to showcase the scalability of FLMs. To address this request, we are also including the 5-strategy extension of RPS, known as RPSSL (Rock-Paper-Scissors-Spock-Lizard), described in https://www.samkass.com/theories/RPSSL.html. This is a 5-dimensional problem (state variables with 4 degrees of freedom + time).
> - Since the submission of this review comment, we have attained the results commensurate with those presented for the 3-dimensional case.
>
> **3. PDE with Non-periodic or Discontinuous BC**
>
> - We agree that including a more sophisticated PDE, with characteristics such as the ones suggested, will benefit the performance assessment of FLMs against the other models considered. In fact, all three reviewers suggested including such a PDE. To satisfy all of these requirements, we have selected one additional PDE, namely the Inviscid Burgers’ Equation, which has a discontinuous solution at terminal time and a BC that induces a discontinuity in the boundary as well. We run experiments for this PDE like we had run for the other 3 and found that, in this case, the vanilla NN with hypertangent activation functions outperforms FLMs, however both attained the same MSE order of magnitude ($10^{-3}$).
>
> *Minor*: By saying "we set the frequencies to nonnegative vectors..." we mean that we are intentionally not initializing the frequencies at zero, but at integer values. By doing so, we allow each frequency component to initiate at their counterpart values appearing in the case of a finite, classical Fourier series, thus allowing FLMs to capture the higher frequency components of the underlying target function.

---

### Review · Reviewer_Zt2w · 2025-10-31

**Summary Of Contributions:**

This paper introduces Fourier learning machines (FLMs), a type of neural network motivated by Fourier analysis, and in particular, non-harmonic Fourier series. This essentially entails a Fourier series with a finite set of learnable (non-integer) frequencies in an FLM "subnetwork," which are then summed together. A comprehensive comparison to other neural network architectures using sinusoidal-like activations is undertaken to contextualize the FLM. Then two sets of numerical experiments are presented. The first is with PDE learning, using a PINN-like loss, where FLMs compare favorably to SIREN and MLP networks in several examples. The second set of experiments are in optimal control problems, where FLMs learn well-performing polices.

### Strengths
- The paper is relatively clear, and the literature review is extensive.
- There is strong empirical performance among the methods tested in PDE learning, and the optimal control applications are interesting.

### Weaknesses

- There are a few claims I find overly broad or inaccurate, though I think these can mostly be fixed by tempering claims. I detail these below.
- The SNN is criticized as "no theoretical background is provided to guarantee the convergence of this NN," which is fair. But the FLM also lacks such a formal convergence analysis, which is left implicit.
- While there is a hyperparameter ablation in Appendix B, further investigation of hyperparameters would strengthen the paper. The reader is left with fairly limited understanding of how many sub-networks are typically beneficial, or how performance behaves with a increasing number of frequencies.
- I have a lot of trouble following the definition of the $m$-Lexi sign matrix. Revising for clarity would be much appreciated.
- The examined PDEs are rather simple, including the heat equation, the Poisson equation, and a Black-Scholes-like PDE with a simple analytic solution, $u(x, t) = \exp(x-t)$. Examining on more difficult PDEs would greatly improve the perceived impact in scientific ML.
- The basis representation blows up with the input dimension $m$. This is not explicitly acknowledged, to the best of my knowledge, and illustrating scaling would improve the paper.

**Additional Comments:**

I think theoretical comparison to the sparse spectrum GP [1] would also be interesting, especially the "optimized" version where the frequencies are learned.

Some additional minor typos:
- Above equation (7), "in Equations Equation 5 and Equation 6"
- Below equation (17), "are the the objective"

[1] Lázaro-Gredilla, M., Quinonero-Candela, J., Rasmussen, C. E., & Figueiras-Vidal, A. R. (2010). Sparse spectrum Gaussian process regression. The Journal of Machine Learning Research, 11, 1865-1881.

**Audience:**

Yes

**Audience Explanation:**

Yes, improved architectures in scientific ML are clearly of interest to a subset of the TMLR audience.

**Claims And Evidence:**

No

**Claims Explanation:**

There are a few claims I find to be lacking in evidence, though I think they can mostly be ameliorated via tempering claims.

1. In my opinion, "an investigation on the use of sine functions as activation functions in NNs first appeared in Parascondolo et al. (2017)" can be misread as implying Parascondolo et al. were the first to attempt/benchmark use of sine activation functions. I think this should be reworded to something like "Parascondolo et al. (2017) undertook a comprehensive and critical investigation on the use of sine activation functions in deep NNs."

2. The novelty claim ("the FLM is the first architecture able to represent a complete, separable Fourier basis in multiple dimensions using a standard Multilayer Perceptron–like architecture") is rather strong, and completeness is not formally stated. Plus, the FLM is formally capable of finite-dimensional representations, which cannot be complete.

**Requested Changes:**

Critical:
- See evidence section above.
- Improve upon the definition of the $m$-Lexi matrix.

Suggestions:
- Include more difficult PDEs for learning and comparison.
- Comment on the cost with respect to the input dimension $m$, and present experiments accordingly. Compare this to something like SIREN.
- More directly experiment and comment on the affect of $N$, beyond the hyperparameter ablation. For example, a plot would be very helpful for the reader.

---

> ### Author Response · Authors · 2025-11-15
>
> Thank you for your review, comments and suggestions.
>
> We are now providing an update on the state of the revisions requested and anticipate that a fully revised manuscript will be uploaded by the end of day on Monday 11/17.
>
> Regarding the requested changes:
>
> **Critical**
> - We fully agree with the reviewer and will update the claims as suggested, removing any potential misunderstanding about the work by Parascondolo et al. (2017).
> - By "complete", we meant a "complete basis" of the Fourier series. For instance, in the two-dimensional case, the use of $sin(\cdot)sin(\cdot) \; sin(\cdot)cos(\cdot) ;\ cos(\cdot)sin(\cdot) \;cos(\cdot)cos(\cdot)$. We agree that this can create confusion with the concept of completeness employed in the context of functional analysis and will improve the text to avoid this potential misunderstanding.
> - We will improve the exposition of the concept of $m-$Lexi Sign matrix. We thank the reviewer for this suggestion and agree that having this concept as clear as possible will benefit reader's understanding about the construction of FLMs.
>
> **Suggestions**
> - We already have experimental results for a fourth PDE, namely the Inviscid Burgers’ Equation, as per suggestion of all three reviewers. The inclusion of this PDE satisfies the requests of a more complicated PDE which includes the presence of discontinuities.
> - To address the last two suggestions, we have decided to include a table relating the number of sub-networks for each FLM and their respective number of parameters. This will be also done for all the models considered. If time allows, we are going to follow the suggestion of the reviewer and include a graphical depiction of the relationship of N (number of sub-networks) and the approximation error obtained by FLMs. We thank the reviewer for this suggestion.
>
> *Minor*:
> - We thank the reviewer for this suggestion. We are currently examining how to make such a theoretical bridge and will attempt to make a note on the relationship between FLMs and GP as they appear in the citation provided.
> - Thank you for identifying the typos in Eq. 7 and 17. They will be fixed in the revised manuscript.

---

> > ### Comment · Reviewer_Zt2w · 2025-11-20
> >
> > I think the authors for these productive revisions. I have a few remaining (minor) comments:
> >
> > - Thanks for revising the abstract regarding completeness, I think the revised claims are more appropriate. I think a similar comment holds in the end of section 2, where an ambiguous claim of "complete" representations remains.
> > - Thanks for the inclusion of Table 4; I hope that the authors include the plot as mentioned in a camera ready version, though I don't deem this "critical."
> > - I look forward to any discussion regarding the sparse spectrum GP and the FLM.

---

> > > ### Author Response · Authors · 2025-11-21
> > >
> > > We thank the reviewer for their thoughtful feedback. Our responses to the comments are below.
> > >
> > > - We unintentionally overlooked the remaining use of the term “complete” at the end of Section 2. In the revised version, we will make this wording consistent with the corrected abstract and remove any implication of a formal completeness guarantee.
> > >
> > > - We appreciate the reviewer’s suggestion. Given the previous two weeks to address all the reviewers comments, we chose to prioritize the computational experiments and other critical suggestions. Moving forward, we will address this by including such a table in the camera-ready version.
> > >
> > > - We thank the reviewer for emphasizing the importance of connecting Sparse Spectrum Gaussian Process (SSGP) with the FLM.  We will add a discussion on the similarities and differences, especially in how both methods learn Fourier frequencies. In fact, we believe the SSGP work further strengthens our central idea that frequency learning is a powerful mechanism for function approximation.

---

### Review · Reviewer_Kxbw · 2025-11-05

**Summary Of Contributions:**

In this paper, the authors propose the Fourier Learning Machine, a neural network architecture designed to emulate the Fourier series representation of an unknown target function. The network employs shifted cosine functions as activation functions, and its weighting parameters are optimized to eliminate redundancies for improved efficiency. Unlike existing methods, this approach enables accurate construction of both standard Fourier series and their non-harmonic extensions. Simulation results demonstrate that the proposed architecture outperforms comparable Fourier network models and conventional multilayer perceptrons.

**Audience:**

Yes

**Audience Explanation:**

The proposed architecture is particularly interesting as it bridges the well-known Fourier series approximation with conventional neural network structures. Preliminary results indicate that there are cases where this architecture is superior to standard networks in terms of accuracy.

**Claims And Evidence:**

Yes

**Claims Explanation:**

All claims are supported with data and or proofs.

**Requested Changes:**

Major:
It seems natural that a Neural Network with this type of smooth basis functions outperforms "standard" MLPs on problems with rather smooth solutions (as shown in Fig. 4-6. It would be interesting to swap one of these examples to a PDE with a discontinuous solution (e.g. https://towardsdatascience.com/can-neural-networks-formulate-shock-wave-147bb75d0715/)

Minor:
Page 7: "in Equations Equation 5 and Equation 6"
Page 13: "These points are randomly sampled" Please specify what kind of distribution was used (assume uniform)

---

> ### Author Response · Authors · 2025-11-15
>
> Thank you for your review, comments and suggestions.
>
> We are now providing an update on the state of the revisions requested and anticipate that a fully revised manuscript will be uploaded by the end of day on Monday 11/17.
>
> Regarding the requested changes:
>
> **Major: PDE with Discontinuous Solution**
>
> We agree that including such a PDE will benefit the assessment of FLMs performance against the other models considered. We have already performed experiments for the Inviscid Burgers’ Equation, which has a discontinuous solution at terminal time and a BC that induces a discontinuity in the boundary as well. We thank the reviewer for suggesting this specific PDE. In this case, the vanilla NN with hypertangent activation functions outperforms FLMs, however both attained the same MSE order of magnitude of $10^{-3}$.
>
> *Minor*:
> - Page 7 typo will be fixed in the revised manuscript. Thank you for catching this.
> - Page 13: The reviewer is correct: we sample points uniformly in the domain of each PDE. We are going to make this clear in the revised manuscript. Thank you for the suggestion.

---

### Decision · Action_Editor_KGd1 · 2025-12-02

**Recommendation:** Accept with minor revision

**Additional Comments:**

One of the reviewers, in their recommendation (which I believe the authors cannot see) asked " I would still encourage the authors to follow through with their promised revisions to a camera ready version", in regards to the following comment made by the authors

"We appreciate the reviewer’s suggestion. Given the previous two weeks to address all the reviewers comments, we chose to prioritize the computational experiments and other critical suggestions. Moving forward, we will address this by including such a table in the camera-ready version.
        We thank the reviewer for emphasizing the importance of connecting Sparse Spectrum Gaussian Process (SSGP) with the FLM. We will add a discussion on the similarities and differences, especially in how both methods learn Fourier frequencies. In fact, we believe the SSGP work further strengthens our central idea that frequency learning is a powerful mechanism for function approximation."

Please address this in the final version of the manuscript.

**Audience:**

Yes

**Audience Explanation:**

All 3 reviewers agreed that this work would be of interest to the TMLR audience. This is especially the case given that the paper develops a new architecture for scientific machine learning.

**Claims And Evidence:**

Yes

**Claims Explanation:**

In the first round of reviews, 2 of the 3 reviewers agreed that the claims were supported. In response to the reviews, the authors provided additional experimental results and were more precise about their claims. After the edits and responses by the authors, all 3 reviewers agreed that the claims were supported.

---

> ### Author Response · Authors · 2025-12-16
>
> We thank you and the reviewers for the careful and constructive evaluation of our manuscript.
>
> We have now submitted the final camera-ready version, in which we have addressed all of the requested minor revisions. We will submit the code repository link in the future after cleaning the scripts and adding detailed comments, and will update the OpenReview page accordingly. We appreciate the time and effort of the reviewers and yourself throughout the review process.